# How Learning Dynamics Drive Adversarially Robust Generalization?

## Abstract

Despite significant progress in adversarially robust learning, the underlying mechanisms that govern robust generalization remain poorly understood. We propose a novel PAC-Bayesian framework that explicitly links adversarial robustness to the posterior covariance of model parameters and the curvature of the adversarial loss landscape. By characterizing discrete-time SGD dynamics near a local optimum under quadratic loss, we derive closed-form posterior covariances for both the stationary regime and the early phase of non-stationary transition. Our analyses reveal how key factors, such as learning rate, gradient noise, and Hessian structure, jointly shape robust generalization during training. Through empirical visualizations of these theoretical quantities, we fundamentally explain the phenomenon of robust overfitting and shed light on why flatness-promoting techniques like adversarial weight perturbation help to improve robustness.

## 1 Introduction

Adversarial robustness—the ability of deep neural networks (DNNs) to maintain performance under worst-case perturbations—remains a fundamental challenge in machine learning. Since the discovery of adversarial examples (Szegedy et al., 2013), numerous methods have been proposed aiming to improve the resilience of DNNs (Goodfellow et al., 2014; Papernot et al., 2016; Wong & Kolter, 2018; Cohen et al., 2019). Among them, PGD-based adversarial training (AT) (Madry et al., 2017) is most popular, which has become the de facto approach for building robust models. Nevertheless, the mechanisms underlying robust generalization are far from fully understood. A striking manifestation of this gap is the phenomenon of robust overfitting (Rice et al., 2020), where robust accuracy on the training set steadily improves while test-time robustness increases shortly but continuously deteriorates after learning rate decay. This paradox underscores a key open question: *what factors determine whether robustness learned during training generalizes reliably to unseen data?*

Several complementary lines of research have sought to address this question. PAC-Bayesian analyses have been adapted to derive non-vacuous generalization bounds in adversarial settings (Xiao et al., 2023; Alquier et al., 2024). However, these bounds often abstract away from the actual optimization trajectory and adopt simple isotropic Gaussian posteriors for tractability, overlooking structural properties of the learned model that are crucial for explaining generalization. Other works (Foret et al., 2020; Dziugaite et al., 2021; Wang et al., 2023) connect the KL divergence term in PAC-Bayesian bounds to curvature-related quantities of the loss landscape, suggesting that richer prior–posterior choices can yield more informative guarantees. In particular, adversarial weight perturbation (AWP) (Wu et al., 2020) studied the role of curvature and flatness in the adversarial loss landscape and designed a variant of adversarial training by iteratively perturbing the model weights, which demonstrates strong empirical support that flatter minima help robust generalization.

Despite these efforts, the existing literature remains fragmented. PAC-Bayes bounds offer general guarantees but lack fidelity to the learning dynamics, whereas curvature-based approaches provide qualitative insight without rigorous predictive guarantees. Motivated by the recent theoretical works on standard generalization by analyzing the continuous- and discrete-time dynamics of stochastic gradient descent (SGD) (Mandt et al., 2017; Liu et al., 2021; Ziyin et al., 2021; Wu & Su, 2023; Ziyin et al., 2024), we introduce a unified but principled framework that links optimization dynamics with both Hessian curvature and posterior geometry through PAC-Bayesian analysis to study robust

generalization and to account for puzzling empirical observations such as robust overfitting, while also rationalizing the effectiveness behind flatness-promoting methods such as AWP.

**Contributions.** We develop a PAC-Bayesian framework that explicitly couples Hessian curvature and posterior geometry through the finite-time dynamics of SGD in adversarial settings. Our analysis begins with a compact PAC-Bayesian inequality that preserves the posterior covariance structure (Section 3). Modeling SGD with momentum in a quadratic basin, we derive closed-form solutions for the posterior covariance in two representative regimes—(i) a stationary regime under a fixed learning rate and (ii) an early non-stationary transition triggered by a learning-rate drop (Section 4). Substituting these covariances into the bound yields tractable inequalities that quantitatively predict the evolution of robust generalization capabilities. To connect theory with practice, we conduct controlled studies under standard adversarial training and adversarial weight perturbation, tracking the Hessian spectrum, gradient-noise covariance, and the bound's dominant terms across different training phases (Section 5). The empirical observations consistently align with our theory, underscoring the central role of coupled curvature–posterior geometry in adversarially robust generalization.

## 2 RELATED WORK

**Adversarial Training.** Adversarial training (Madry et al., 2017), which optimizes the model parameters using SGD while leveraging projected gradient descent (PGD) to simultaneously search for worst-case input perturbations, is a canonical approach for robustness. Despite wide adoptions, it often exhibits *robust overfitting*, with test-time robust accuracy degrading while training robustness keeps improving (Rice et al., 2020). To mitigate this, objective-level regularization such as TRADES introduces a robustness–accuracy trade-off via a KL term (Zhang et al., 2019), and semi-supervised variants leverage unlabeled data to expand adversarial support (Carmon et al., 2019; Gowal et al., 2021). Complementarily, landscape-shaping methods—including adversarial weight perturbation (AWP) and sharpness-aware minimization (SAM)—promote flatter minima by perturbing parameters or minimizing neighborhood worst-case loss (Wu et al., 2020; Foret et al., 2020), highlighting the central role of curvature in shaping robust generalization in adversarial settings.

**Robust Generalization.** A parallel line of research studies adversarially robust generalization from a theoretical standpoint. PAC-Bayesian analyses for adversarial robustness (Viallard et al., 2021; Mustafa et al., 2023; Xiao et al., 2023) derive explicit generalization bounds but model the posterior as a static, trajectory-independent distribution, thereby overlooking how SGD dynamics shape posterior geometry and curvature-dependent behavior. Stability-based approaches (Xing et al., 2021; Xiao et al., 2022; Cullina et al., 2018; Tian & Mao, 2025) provide guarantees through uniform stability, but they abstract away the structure of the adversarial loss landscape—ignoring the curvature and the anisotropic noise induced by SGD—and their bounds do not vary meaningfully with the perturbation strength $\epsilon$, thus unable to capture how increasing $\epsilon$ fundamentally alters robust generalization. Recent studies of robust overfitting (Fu & Wang, 2023; Mustafa et al., 2024; Liu et al., 2024) offer valuable insights but still rely on fixed hypothesis classes or static posteriors that do not account for the temporal evolution of training dynamics. Collectively, these theoretical frameworks capture important aspects of robust generalization but lack a dynamic perspective that connects curvature, gradient noise, and posterior geometry throughout the course of adversarial training.

**SGD Dynamics.** The generalization ability of deep networks is strongly shaped by the dynamics of stochastic gradient descent (SGD). With a finite learning rate, SGD can be modeled as a stochastic process where deterministic gradient flow is perturbed by minibatch noise (Mandt et al., 2017; Liu et al., 2021). This noise is anisotropic: its covariance reflects the local loss curvature and data structure, producing characteristic fluctuations in parameter trajectories. The stationary distribution acts as an implicit posterior whose variance is shaped by the learning rate, batch size, and Hessian spectrum (Ziyin et al., 2021; 2024). These noise–curvature interactions govern how SGD explores flat versus sharp regions, thereby influencing stability and generalization. In adversarial settings, where loss landscapes are typically sharper and more anisotropic, such interactions are amplified, motivating a dynamic view of robust generalization beyond static capacity-based bounds. In this work, we connect the robust generalization performance of adversarially trained models with the SGD learning dynamics through a PAC-Bayesian analytical framework, aiming to explain phenomena such as robust overfitting more fundamentally.

## 3 BOUNDING ROBUST GENERALIZATION VIA PAC-BAYESIAN FRAMEWORK

### 3.1 PRELIMINARIES

**Notation.** We use lowercase boldface letters such as $\boldsymbol{x}$ to denote vectors and uppercase boldface letters such as $\mathbf{X}$ for matrices. For any vector $\boldsymbol{x} \in \mathbb{R}^d$, denote by $\|\boldsymbol{x}\|_p$ with $p \geq 1$ the $L_p$-norm of $\boldsymbol{x}$. For any matrix $\mathbf{X} \in \mathbb{R}^{m \times m}$, $\mathrm{Tr}(\mathbf{X})$ denotes its trace, $\|\mathbf{X}\|_{\mathrm{op}}$ its operator norm, and $\det(\mathbf{X})$ its determinant. For any $\mathcal{S}$, $|\mathcal{S}|$ denotes its cardinality. We write $\mathbf{X} \succ 0$ to indicate that $\mathbf{X}$ is a positive definite matrix, and $\mathbf{X} \succeq 0$ for semi-positive definite. Let $\mathbf{I}$ be the identity matrix and $\mathcal{N}(\boldsymbol{\mu}, \boldsymbol{\Sigma})$ be the Gaussian distribution with mean $\boldsymbol{\mu}$ and covariance $\boldsymbol{\Sigma}$. Given sequences $\{a_n\}$ and $\{b_n\}$, we write $a_n = O(b_n)$ if there exist constants $n_0 \in \mathbb{Z}_+$ and $C > 0$ such that $a_n \leq C \cdot b_n$ for all $n \geq n_0$.

In particular, we work with the following notion of adversarial risk, which closely relates to adversarially robust generalization and often serves as the theoretical basis for robustness evaluation in previous literature on adversarial ML (Madry et al., 2017; Zhang et al., 2019; Foret et al., 2020).

**Definition 3.1** (Adversarial risk). *Let $\mathcal{X} \subseteq \mathbb{R}^d$ be the input space, $\mathcal{Y}$ be the output label space, $f_{\boldsymbol{w}} : \mathcal{X} \to \mathcal{Y}$ be a model with $\boldsymbol{w} \in \mathbb{R}^m$ being its parameters, and $\mathcal{D}$ be a data distribution over $\mathcal{X} \times \mathcal{Y}$. For any distance metric $\Delta : \mathcal{X} \times \mathcal{X} \to \mathbb{R}_{\geq 0}$ and $\epsilon \geq 0$, define $\mathcal{B}_\epsilon(\mathbf{0}) = \{\boldsymbol{\delta} \in \mathcal{X} : \Delta(\boldsymbol{\delta}, \mathbf{0}) \leq \epsilon\}$ as the ball centered at $\mathbf{0}$ with radius $\epsilon$ measured in $\Delta$. Then, the adversarial risk of $f_{\boldsymbol{w}}$ is defined as:*

$$\mathcal{R}_{\mathrm{adv}}(\boldsymbol{w}) := \mathbb{E}_{(\boldsymbol{x},y) \sim \mathcal{D}}\big[\ell_{\mathrm{adv}}(\boldsymbol{w}, \boldsymbol{x}, y)\big], \quad where \ \ell_{\mathrm{adv}}(\boldsymbol{w}, \boldsymbol{x}, y) = \max_{\boldsymbol{\delta} \in \mathcal{B}_\epsilon(\mathbf{0})} \ell(\boldsymbol{w}, \boldsymbol{x} + \boldsymbol{\delta}, y). \quad (1)$$

*Here, $\ell(\boldsymbol{w}, \boldsymbol{x}, y)$ denotes the standard loss function (e.g., cross-entropy loss) that measures the discrepancy between the model prediction $\hat{y} = f_{\boldsymbol{w}}(\boldsymbol{x})$ and the ground-truth class label $y$.*

Small adversarial risk indicates that the model $f_{\boldsymbol{w}}$ is resilient to worst-case perturbations in $\mathcal{B}_\epsilon(\mathbf{0})$, while a larger value of $\mathcal{R}_{\mathrm{adv}}(\boldsymbol{w})$ means higher vulnerability to adversarial perturbations. Note that when $\epsilon = 0$, adversarial risk $\mathcal{R}_{\mathrm{adv}}(\boldsymbol{w})$ is equivalent to the standard notion of risk. Aligned with prior work (Madry et al., 2017; Rice et al., 2020), we consider $\Delta$ as some $\ell_p$-norm bounded distance.

The following lemma, proven in Appendix A.1, establishes a generic robust generalization bound relating adversarial risk to its empirical counterpart through a PAC-Bayesian framework. PAC-Bayes bounds have been pivotal in understanding the standard generalization of ML models (McAllester, 1999; Neyshabur et al., 2017; Dziugaite & Roy, 2017; Xiao et al., 2023; Alquier et al., 2024).

**Lemma 3.2** (PAC-Bayesian Robust Generalization Bound). *Let $\mathcal{D}$ be a probability distribution over $\mathcal{X} \times \mathcal{Y}$ and $\mathcal{S}$ be a set of examples i.i.d. sampled from $\mathcal{D}$. Suppose $\mathcal{P}$ is a data-independent prior distribution defined over the model parameter space $\mathcal{W}$. For any $\beta > 0$, any $\alpha \in (0, 1)$ and any posterior distribution $\mathcal{Q}$ supported on $\mathcal{W}$, with probability at least $1 - \alpha$, we have*

$$\mathbb{E}_{\boldsymbol{w} \sim \mathcal{Q}}\big[\mathcal{R}_{\mathrm{adv}}(\boldsymbol{w})\big] \leq \mathbb{E}_{\boldsymbol{w} \sim \mathcal{Q}}\left[\frac{1}{|\mathcal{S}|} \sum_{(\boldsymbol{x},y) \in \mathcal{S}} \ell_{\mathrm{adv}}(\boldsymbol{w}, \boldsymbol{x}, y)\right] + \frac{1}{\beta}\mathrm{KL}(\mathcal{Q} \,\|\, \mathcal{P}) + \frac{\beta C^2}{8|\mathcal{S}|} - \frac{1}{\beta}\ln \alpha, \quad (2)$$

*where $\mathrm{KL}(\mathcal{Q} \,\|\, \mathcal{P})$ denotes the Kullback–Leibler (KL) divergence between the posterior $\mathcal{Q}$ and the prior $\mathcal{P}$, and $C$ is a constant derived by Hoeffding's inequality that bounds the loss range.*

In the following discussions, we write $\hat{\mathcal{R}}_{\mathrm{adv}}(\boldsymbol{w}, \mathcal{S}) = \sum_{(\boldsymbol{x},y) \in \mathcal{S}} \ell_{\mathrm{adv}}(\boldsymbol{w}, \boldsymbol{x}, y)/|\mathcal{S}|$ as the empirical adversarial loss of $f_{\boldsymbol{w}}$ with $\mathcal{S}$, and we use $d, m$ to denote the dimensions of the input space $\mathcal{X}$ and the parameter space $\mathcal{W}$, respectively for ease of presentation. Note that the robust generalization bound derived in Equation 2 characterizes a general relationship between $\mathcal{R}_{\mathrm{adv}}(\boldsymbol{w})$ and $\hat{\mathcal{R}}_{\mathrm{adv}}(\boldsymbol{w}, \mathcal{S})$, which holds for any data-independent prior $\mathcal{P}$ and any posterior distribution $\mathcal{Q}$. Assuming the prior $\mathcal{P}$ to be independent of $\mathcal{S}$ ensures the applicability of Fubini's theorem, which has been widely adopted in the literature for establishing PAC-Bayes bounds (Mbacke et al., 2023; Alquier et al., 2024). For adversarial training algorithms, the prior $\mathcal{P}$ can be understood as the weight initialization, while the posterior $\mathcal{Q}$ can be viewed as the distribution of model parameters at a certain training epoch.

### 3.2 RELATING ROBUST GENERALIZATION TO HESSIAN AND POSTERIOR STRUCTURE

So far, we've established a generic upper bound on robust generalization. To further digest Lemma 3.2, we need to set the prior $\mathcal{P}$ and the posterior $\mathcal{Q}$ properly such that the right-hand side of Equation 2 can be simplified into an analytical form while being sufficiently tight to yield useful insights.

Specifically, we first introduce the following assumption regarding the prior and the posterior.

**Assumption 3.3** (Gaussian Prior & Posterior). *We assume both $\mathcal{P}$ and $\mathcal{Q}$ follow Gaussian distributions: $\mathcal{P} = \mathcal{N}(\mathbf{0}, \sigma_{\mathcal{P}}^2 \mathbf{I})$ and $\mathcal{Q} = \mathcal{N}(\boldsymbol{\mu}_{\mathcal{Q}}, \boldsymbol{\Sigma}_{\mathcal{Q}})$, where $\sigma_{\mathcal{P}} \in \mathbb{R}_+$, $\boldsymbol{\mu}_{\mathcal{Q}} \in \mathbb{R}^m$ and $\boldsymbol{\Sigma}_{\mathcal{Q}} \succ 0$.*

For the ease of presentation, we assume the posterior is a single Gaussian distribution for the following derivations. However, we note that our theoretical results and proof techniques can be easily generalized to scenarios where $\mathcal{Q}$ is modeled as a mixture of Gaussians (Corollary 3.8).

The following lemma, proven in Appendix A.2, illustrates how the KL divergence term in Equation 2 can be simplified into an analytically tractable expression using Assumption 3.3.

**Lemma 3.4.** *Let the prior $\mathcal{P} = \mathcal{N}(0, \sigma_{\mathcal{P}}^2 \mathbf{I})$ and the posterior $\mathcal{Q} = \mathcal{N}(\boldsymbol{\mu}_{\mathcal{Q}}, \boldsymbol{\Sigma}_{\mathcal{Q}})$. Then, we have*

$$\mathrm{KL}(\mathcal{Q} \,\|\, \mathcal{P}) = \frac{\mathrm{Tr}(\boldsymbol{\Sigma}_{\mathcal{Q}})}{2\sigma_{\mathcal{P}}^2} + \frac{\|\boldsymbol{\mu}_{\mathcal{Q}}\|_2^2}{2\sigma_{\mathcal{P}}^2} - \frac{m}{2} + \frac{m}{2} \ln \sigma_{\mathcal{P}}^2 - \frac{1}{2} \ln \det \boldsymbol{\Sigma}_{\mathcal{Q}}. \tag{3}$$

Compared to the commonly-adopted assumption that both $\mathcal{P}$ and $\mathcal{Q}$ are spherical Gaussians (Grunwald et al., 2021; Jin et al., 2022; Mbacke et al., 2023), assuming an isotropic Gaussian prior while allowing a general Gaussian posterior enables an analytically tractable yet less restrictive expression of the KL divergence term. Since the robust generalization bound derived in Lemma 3.2 holds for any data-independent prior and posterior, setting $\mathcal{P} = \mathcal{N}(\mathbf{0}, \sigma_{\mathcal{P}}^2 \mathbf{I})$, $\mathcal{Q} = \mathcal{N}(\boldsymbol{\mu}_{\mathcal{Q}}, \boldsymbol{\Sigma}_{\mathcal{Q}})$ does not compromise the validity of the bound. Such choices retain the simplicity of a closed-form KL divergence while capturing anisotropic parameter variability through the full covariance matrix $\boldsymbol{\Sigma}_{\mathcal{Q}}$.

In addition, to deal with the first empirical adversarial loss term on the right-hand side of Equation 2, we introduce the following quadratic loss assumption regarding the posterior $\mathcal{Q}$, which enables us to further connect the expected empirical adversarial loss to the Hessian and posterior structure.

**Assumption 3.5** (Quadratic Loss). *We assume there exist a local optimum $\boldsymbol{w}^* \in \mathbb{R}^m$ such that for any $\boldsymbol{w} \sim \mathcal{Q}$, $\hat{\mathcal{R}}_{\mathrm{adv}}(\boldsymbol{w}, \mathcal{S})$ can be characterized by the following quadratic loss defined at $\boldsymbol{w}^*$:*

$$\hat{\mathcal{R}}_{\mathrm{adv}}(\boldsymbol{w}, \mathcal{S}) = \hat{\mathcal{R}}_{\mathrm{adv}}(\boldsymbol{w}^*, \mathcal{S}) + \frac{1}{2}(\boldsymbol{w} - \boldsymbol{w}^*)^\top \mathbf{H}^* (\boldsymbol{w} - \boldsymbol{w}^*), \tag{4}$$

*where $\mathbf{H}^*$ is the Hessian matrix with the empirical adversarial loss $\hat{\mathcal{R}}_{\mathrm{adv}}(\boldsymbol{w}, \mathcal{S})$ at the local optimum $\boldsymbol{w}^*$. For simplicity, we assume the Hessian matrix at $\boldsymbol{w}^*$ is positive definite[1], namely $\mathbf{H}^* \succ 0$.*

The quadratic loss assumption has been adopted in prior works for formalizing the learning dynamics of ML models (Bartlett et al., 2021; Liu et al., 2021; Ziyin et al., 2021; Suri et al., 2024; Ziyin et al., 2024). Imposing such an assumption not only simplifies the derivations but also can largely capture the dynamics of deep learning models used in practice. For instance, the stationary dynamics of SGD can be viewed as having a quadratic potential near a local minimum (Liu et al., 2021).

**Lemma 3.6.** *Under Assumptions 3.3 and 3.5, we can simplify the expected adversarial loss as:*

$$\mathbb{E}_{\boldsymbol{w} \sim \mathcal{Q}}\big[\hat{\mathcal{R}}_{\mathrm{adv}}(\boldsymbol{w}, \mathcal{S})\big] = \hat{\mathcal{R}}_{\mathrm{adv}}(\boldsymbol{w}^*, \mathcal{S}) + \frac{1}{2}(\boldsymbol{\mu}_{\mathcal{Q}} - \boldsymbol{w}^*)^\top \mathbf{H}^*(\boldsymbol{\mu}_{\mathcal{Q}} - \boldsymbol{w}^*) + \frac{1}{2}\mathrm{Tr}(\mathbf{H}^* \boldsymbol{\Sigma}_{\mathcal{Q}}). \tag{5}$$

Lemma 3.6, proven in Appendix A.3, suggests that close to a local minimum $\boldsymbol{w}^*$, the expected empirical adversarial loss under $\mathcal{Q}$ is primarily governed by the distance between $\boldsymbol{\mu}_{\mathcal{Q}}$ and $\boldsymbol{w}^*$ induced by $\mathbf{H}^*$, plus the trace of the multiplication of the Hessian and the posterior covariance $\mathrm{Tr}(\mathbf{H}^* \boldsymbol{\Sigma}_{\mathcal{Q}})$.

By expressing the KL divergence and the expected empirical adversarial loss terms using Equation 3 and Equation 5 in the robust generalization bound in Lemma 3.2, we obtain the following result.

**Theorem 3.7** (Robust Generalization with Gaussians & Quadratic Loss). *Under the same set of conditions as assumed in Lemmas 3.2, 3.4 and 3.6, with probability at least $1 - \alpha$, we have*

$$\mathbb{E}_{\boldsymbol{w} \sim \mathcal{Q}}[\mathcal{R}_{\mathrm{adv}}(\boldsymbol{w})] \leq \frac{1}{2}\mathrm{Tr}(\mathbf{H}^* \boldsymbol{\Sigma}_{\mathcal{Q}}) + \frac{1}{2}(\boldsymbol{\mu}_{\mathcal{Q}} - \boldsymbol{w}^*)^\top \mathbf{H}^*(\boldsymbol{\mu}_{\mathcal{Q}} - \boldsymbol{w}^*) + \hat{\mathcal{R}}_{\mathrm{adv}}(\boldsymbol{w}^*, \mathcal{S})$$

$$+ \frac{1}{2\beta}\left(\frac{\mathrm{Tr}(\boldsymbol{\Sigma}_{\mathcal{Q}})}{\sigma_{\mathcal{P}}^2} + \frac{\|\boldsymbol{\mu}_{\mathcal{Q}}\|_2^2}{\sigma_{\mathcal{P}}^2} - m + m \ln \sigma_{\mathcal{P}}^2 - \ln \det \boldsymbol{\Sigma}_{\mathcal{Q}}\right) + \frac{\beta C^2}{8|\mathcal{S}|} - \frac{1}{\beta} \ln \alpha. \tag{6}$$

---

[1]Technically, we can only ensure $\mathbf{H}^*$ is a positive semidefinite matrix; however, one can easily enforce it to be positive definite by adding a small $L_2$ weight-norm regularization term to the empirical adversarial loss, which is a typical implementation adopted in practice for training ML models.

Although the perturbation strength $\epsilon$ does not explicitly appear on the right hand side of Equation 6, it will implicitly affect the bound through the Hessian matrix $\mathbf{H}^*$ and posterior parameters ($\boldsymbol{\mu}_{\mathcal{Q}}$, $\boldsymbol{\Sigma}_{\mathcal{Q}}$), as long as we are analyzing algorithms trained to minimize the empirical adversarial loss. The hyperparameter $\beta$ balances empirical adversarial loss, KL divergence, and complexity penalty. In practice, $\beta$ is often chosen by cross-validation or proportional to $\sqrt{|\mathcal{S}|}$ so that all $\beta$-involved terms are of comparable order. Among all the terms in Equation 6, $\hat{\mathcal{R}}_{\mathrm{adv}}(\boldsymbol{w}^*, \mathcal{S})$, $C$, $\alpha$, $m$, and $\sigma_{\mathcal{P}}^2$ can be viewed as constants once the learning setup is decided. The remaining components–such as the matrix trace terms $\mathrm{Tr}(\mathbf{H}^*\boldsymbol{\Sigma}_{\mathcal{Q}})$ and $\mathrm{Tr}(\boldsymbol{\Sigma}_{\mathcal{Q}})$, the weight norm $\|\boldsymbol{\mu}_{\mathcal{Q}}\|_2^2$, and the log-determinant term $\ln\det\boldsymbol{\Sigma}_{\mathcal{Q}}$–depend on the posterior distribution $\mathcal{Q}$, which will be largely affected by optimization and learning dynamics. As we will illustrate in Section 4, these quantities will be the primary factors explaining the underlying mechanisms of adversarially robust generalization.

Assumptions 3.3 and 3.5 assume the posterior is a single Gaussian centered around a local optimum. In practice, however, the posterior learned by adversarial training may be distributed across multiple regions in the model parameter space. To accommodate this, we relax the assumptions and extend our analysis to a more general family of Gaussian mixture posterior distributions. The corresponding robust generalization bound is derived in Corollary 3.8, with the full proof deferred to Appendix A.4.

**Corollary 3.8** (Robust Generalization with Gaussian Mixtures & Locally Quadratic Loss)**.** *Let* $\mathcal{P} = \mathcal{N}(0, \sigma_{\mathcal{P}}^2\mathbf{I})$ *be the prior, and let the posterior be a mixture of Gaussians with the form:*

$$\mathcal{Q} = \sum_{\ell=1}^{L} \pi_\ell \, \mathcal{Q}_\ell, \quad \text{where } \mathcal{Q}_\ell = \mathcal{N}(\boldsymbol{\mu}_\ell, \boldsymbol{\Sigma}_\ell), \sum_{\ell=1}^{L} \pi_\ell = 1, \text{ and } \pi_\ell \geq 0. \tag{7}$$

*For each posterior component* $\mathcal{Q}_\ell$*, assume the adversarial loss is locally quadratic at a respective local optimal point* $\boldsymbol{w}_\ell^*$*. For any* $\beta > 0$ *and* $\alpha \in (0, 1)$*, with probability at least* $1 - \alpha$*, we have*

$$\mathbb{E}_{\boldsymbol{w} \sim \mathcal{Q}}[\mathcal{R}_{\mathrm{adv}}(\boldsymbol{w})] \leq \sum_{\ell=1}^{L} \pi_\ell \left[ \hat{\mathcal{R}}_{\mathrm{adv}}(\boldsymbol{w}_\ell^*, \mathcal{S}) + \frac{1}{2}(\boldsymbol{\mu}_\ell - \boldsymbol{w}_\ell^*)^\top \mathbf{H}_\ell^*(\boldsymbol{\mu}_\ell - \boldsymbol{w}_\ell^*) + \frac{1}{2}\mathrm{Tr}(\mathbf{H}_\ell^*\boldsymbol{\Sigma}_\ell) \right]$$

$$+ \sum_{\ell=1}^{L} \frac{\pi_\ell}{2\beta} \left( \frac{\mathrm{Tr}(\boldsymbol{\Sigma}_\ell)}{\sigma_{\mathcal{P}}^2} + \frac{\|\boldsymbol{\mu}_\ell\|_2^2}{\sigma_{\mathcal{P}}^2} - m + m \ln \sigma_{\mathcal{P}}^2 - \ln \det \boldsymbol{\Sigma}_\ell \right) + \frac{\beta C^2}{8|\mathcal{S}|} - \frac{1}{\beta} \ln \alpha. \tag{8}$$

## 4    How Learning Dynamics Shape Robust Generalization?

In the previous section, we've established an upper bound on robust generalization in Theorem 3.7, relating the population adversarial risk to the curvature of the empirical adversarial loss $\mathbf{H}^*$ and the covariance matrix of the posterior distribution $\boldsymbol{\Sigma}_{\mathcal{Q}}$, through a PAC-Bayesian framework. However, it is still difficult to comprehend what the posterior $\mathcal{Q}$ really means in the context of robust learning. Therefore, we propose to investigate the behavior of the posterior mean $\boldsymbol{\mu}_{\mathcal{Q}}$ and covariance $\boldsymbol{\Sigma}_{\mathcal{Q}}$ for models learned during adversarial training with *stochastic gradient descent* (SGD) optimizers.

More specifically, we analyze the learning dynamics of SGD with Polyak momentum. The iterative updates of model parameters during SGD can be cast into a dynamical system: for $t = 1, 2, \ldots$,

$$\boldsymbol{g}_t = \mathbf{H}^*(\boldsymbol{w}_{t-1} - \boldsymbol{w}^*) + \boldsymbol{\xi}_{t-1}, \quad \boldsymbol{h}_t = \mu\boldsymbol{h}_{t-1} + \boldsymbol{g}_t, \quad \boldsymbol{w}_t = \boldsymbol{w}_{t-1} - \eta\boldsymbol{h}_t, \tag{9}$$

where $\eta > 0$ is the learning rate and $\mu \in [0, 1)$ denotes the momentum hyperparameter. In Equation 9, $\mathbf{H}^*(\boldsymbol{w}_{t-1} - \boldsymbol{w}^*)$ can be understood as the expected gradient incurred by SGD with respect to the empirical adversarial loss $\hat{\mathcal{R}}_{\mathrm{adv}}(\boldsymbol{w}, \mathcal{S})$ assumed in Equation 4, while the extra $\boldsymbol{\xi}_{t-1}$ term denotes the mini-batch gradient noise with $\mathbb{E}[\boldsymbol{\xi}_{t-1}] = \mathbf{0}$. Throughout this section, we interpret the posterior $\mathcal{Q}$ as the data-dependent distribution of the SGD iterates $\boldsymbol{w}_t$ conditioned on the training set $\mathcal{S}$, either at stationarity or at a finite iteration $t$. We use $\mathbf{C}_{t-1} = \mathbb{E}[\boldsymbol{\xi}_{t-1}\boldsymbol{\xi}_{t-1}^\top]$ to denote its covariance.

Built upon the update rule defined by Equation 9, the following lemma, proven in Appendix B.1, characterizes how the posterior covariance $\boldsymbol{\Sigma}_{\mathcal{Q}}$ is updated during the optimization process of SGD.

**Lemma 4.1** (State-Space Representation & Covariance Propagation)**.** *Consider the dynamical system induced by SGD defined in Equation 9 with learning rate* $\eta > 0$ *and momentum* $\mu \in [0, 1)$*.*

*Denote by the joint state vector $\boldsymbol{u}_t = \begin{bmatrix} \boldsymbol{w}_t - \boldsymbol{w}^* \\ \boldsymbol{h}_t \end{bmatrix} \in \mathbb{R}^{2m}$, then we have*

$$\boldsymbol{u}_t = \mathbf{A}\boldsymbol{u}_{t-1} + \mathbf{G}\boldsymbol{\xi}_{t-1}, \ \ where \ \mathbf{A} = \begin{bmatrix} \mathbf{I} - \eta\mathbf{H} & -\eta\mu\mathbf{I} \\ \mathbf{H} & \mu\mathbf{I} \end{bmatrix}, \ \mathbf{G} = \begin{bmatrix} -\eta\mathbf{I} \\ \mathbf{I} \end{bmatrix}. \tag{10}$$

*Suppose at the initial state, the (posterior) distribution of the model parameters has covariance $\boldsymbol{\Sigma}_t$ (i.e., at time step $t$), then after running $k \geq 1$ steps of SGD with momentum, we have*

$$\boldsymbol{\Sigma}_{t+k} = \boldsymbol{\Pi}\mathbf{A}^k\,\boldsymbol{\Sigma}_t\,(\mathbf{A}^k)^\top\boldsymbol{\Pi}^\top \ + \ \sum_{j=0}^{k-1}(\boldsymbol{\Pi}\mathbf{A}^j\mathbf{G})\,\mathbf{C}_{t+k-1-j}\,(\boldsymbol{\Pi}\mathbf{A}^j\mathbf{G})^\top, \tag{11}$$

*where $\boldsymbol{\Pi} = \begin{bmatrix} \mathbf{I} & \mathbf{0} \end{bmatrix}$ denotes the projection operator mapping to the first state.*

Equipped with the dynamical view of SGD dynamics derived in Lemma 4.1, we now introduce two variations of the PAC-Bayesian robust generalization bound under: (i) the stationary regime (Section 4.1), where the posterior reaches a steady state, and (ii) the early transition phase from stationary to non-stationary state (Section 4.2), which is often triggered by a learning rate change. Both scenarios are highly relevant to explaining the robust overfitting phenomenon detailed in Rice et al. (2020).

### 4.1 STATIONARY REGIME

Recall that the remaining task is to analyze the posterior mean and covariance $(\boldsymbol{\mu}_\mathcal{Q}, \boldsymbol{\Sigma}_\mathcal{Q})$ such that we can understand all the technical terms in the PAC-Bayesian robust generalization bound in Theorem 3.7. The following two lemmas, proven in Appendix B.2, characterize how the mean and covariance are derived from SGD dynamics under the stationary regime.

**Lemma 4.2** (Stationary Mean). *Under Assumptions 3.3 and 3.5, suppose the posterior $\mathcal{Q}$ reaches a steady state with stationary mean $\boldsymbol{\mu} = \lim_{t\to\infty}\mathbb{E}(\boldsymbol{w}_t)$, then we have $\boldsymbol{\mu} = \boldsymbol{w}^*$.*

**Lemma 4.3** (Stationary Covariance). *Under the same conditions as in Lemma 4.2, suppose both the stationary covariance $\boldsymbol{\Sigma} = \lim_{t\to\infty}\mathrm{Cov}(\boldsymbol{w}_t)$ and the noise covariance $\mathbf{C} = \lim_{t\to\infty}\mathbf{C}_t$ exist and are finite. Then, the following equation holds:*

$$\boldsymbol{\Sigma} = \boldsymbol{\Pi}\,\boldsymbol{\Sigma}_{\mathrm{joint}}, \ \ where \ \boldsymbol{\Sigma}_{\mathrm{joint}} \ satisfies \ \boldsymbol{\Sigma}_{\mathrm{joint}} = \mathbf{A}\boldsymbol{\Sigma}_{\mathrm{joint}}\mathbf{A}^\top + \mathbf{G}\mathbf{C}\mathbf{G}^\top, \tag{12}$$

*where $\boldsymbol{\Pi}$, $\mathbf{A}$, and $\mathbf{G}$ are defined in Lemma 4.1. In addition, if the noise covariance $\mathbf{C}$ commutes with the Hessian $\mathbf{H}^*$, then the stationary covariance $\boldsymbol{\Sigma}$ has a closed-form solution:*

$$\boldsymbol{\Sigma} = \left[\mathbf{H}^*\left(2\mathbf{I} - \frac{\eta}{1+\mu}\mathbf{H}^*\right)\right]^{-1}\frac{\eta}{1-\mu}\mathbf{C}. \tag{13}$$

**Remark 4.4.** Assuming $\mathbf{C}$ commutes with $\mathbf{H}^*$ under the stationary regime aligns with empirical observations that gradient noise covariance tends to align with the Hessian eigenspectrum during representation formation in neural networks (Ziyin et al., 2025). Denote by $\{\lambda_1, \lambda_2, \ldots, \lambda_m\}$ and $\{\gamma_1, \gamma_2, \ldots, \gamma_m\}$ the two sets of eigenvalues of $\mathbf{H}^*$ and $\mathbf{C}$, respectively. As shown in the proof of Lemma 4.3, the set of eigenvalues of the stationary covariance $\boldsymbol{\Sigma}$ is given by:

$$\forall i \in \{1, 2, \ldots, m\}, \quad \sigma_i^2 = \frac{\eta}{1-\mu} \cdot \frac{\gamma_i}{\lambda_i \cdot \left(2 - \frac{\eta}{1+\mu}\lambda_i\right)}, \tag{14}$$

where $\sigma_i^2$ stands for the $i$-th eigenvalue of $\boldsymbol{\Sigma}$. Note that the stability condition of Equation 14 requires that $0 < \lambda_i < \frac{2(1+\mu)}{\eta}$ for any $i$. Otherwise, the stationary covariance $\boldsymbol{\Sigma}$ does not exist. Since both the Hessian $\mathbf{H}^*$ and the noise covariance $\mathbf{C}$ implicitly depend on the perturbation strength $\epsilon$, altering the value of $\epsilon$ will correspondingly influence the structure of stationary covariance $\boldsymbol{\Sigma}$.

We can prove the following theorem by applying the stationary mean and covariance formulations in the above lemmas to the PAC-Bayesian robust generalization bound in Theorem 3.7.

**Theorem 4.5** (Robust Generalization under Stationary Regime). *Assume the same conditions as used in Theorem 3.7 and Lemma 4.1. Suppose the posterior $\mathcal{Q}$ reaches a stationary state, and $\mathbf{C}$*

*commutes with $\mathbf{H}^*$. Then, for any $\beta > 0$ and $\alpha \in (0, 1)$, with probability at least $1 - \alpha$, we have*

$$\mathbb{E}_{\boldsymbol{w} \sim \mathcal{Q}}[\mathcal{R}_{\mathrm{adv}}(\boldsymbol{w})] \leq \frac{1}{2} \sum_{i=1}^{m} \lambda_i \, \sigma_i^2 + \frac{1}{2\beta} \left( \frac{\sum_{i=1}^{m} \sigma_i^2}{\sigma_{\mathcal{P}}^2} + \frac{\|\boldsymbol{w}^*\|_2^2}{\sigma_{\mathcal{P}}^2} - \sum_{i=1}^{m} \ln \sigma_i^2 \right)$$
$$+ \hat{\mathcal{R}}_{\mathrm{adv}}(\boldsymbol{w}^*, \mathcal{S}) + \frac{1}{2\beta} \big( -m + m \ln \sigma_{\mathcal{P}}^2 \big) + \frac{\beta C^2}{8|\mathcal{S}|} - \frac{1}{\beta} \ln \alpha, \quad (15)$$

*where $\lambda_i, \gamma_i$ and $\sigma_i$ are the $i$-th eigenvalues of $\mathbf{H}^*, \mathbf{C}$ and $\boldsymbol{\Sigma}$, respectively defined in Remark 4.4.*

**Remark 4.6.** Assuming the posterior $\mathcal{Q}$ reaches a stationary state during the optimization of adversarial loss using SGD, Theorem 4.5 suggests that the PAC-Bayesian robust generalization bound depends on analytical terms related to the Hessian and noise covariance eigenvalues $(\lambda_i, \gamma_i)$, along with learning parameters such as learning rate $\eta$ and momentum $\mu$. As we will illustrate in Section 5, the dominant terms in Equation 15 will be the multiplicative term $\sum_i \lambda_i \sigma_i^2$ and the log-determinant term $-\sum_i \ln \sigma_i^2$, since the (top) eigenvalues of the Hessian matrix $\mathbf{H}^*$ can be order-wise larger than those of $\mathbf{C}$, and the value of the posterior covariance eigenvalues $\sigma_i$ defined by Equation 14 are usually very small, especially after the learning rate step decay during adversarial training.

## 4.2 INITIAL PHASE OF NON-STATIONARY TRANSITION

While the stationary posterior distribution derived in Section 4.1 explains a lot about the SGD dynamics and leads to grounded interpretations, one might also be interested in how the key theoretical quantities evolve if the stationary assumption breaks. This is particularly important for explaining the robust overfitting phenomenon (Rice et al., 2020)–it remains an open question why the test robust error drops immediately after the first learning rate decay step but steadily increases afterward.

The following theorem, whose formal version and proof are in Appendix B.3, shows how the posterior covariance $\boldsymbol{\Sigma}_{\mathcal{Q}}$ behaves in the early non-stationary transient phase. We consider a scenario where the SGD-induced dynamical system (Equation 9) reaches a stationary state with learning rate $\eta_1$ until time step $t$, whereas at step $t + 1$, the learning rate is reduced to $\eta_2$ for future SGD steps.

**Theorem 4.7** (Robust Generalization after Learning Rate Decay: Informal). *Assume the same conditions as used in Theorem 3.7 and Lemma 4.1. Suppose $\mathcal{Q}$ reaches a stationary state for SGD with $(\eta_1, \mu)$ at time step $t$ and $\mathbf{C}$ commutes with $\mathbf{H}^*$. After reducing the learning rate from $\eta_1$ to $\eta_2$ and running $k$ steps of SGD, the $i$-th eigenvalue of the posterior covariance can be approximated as:*

$$\sigma_i^2(t+k) \approx \frac{\eta_1 \gamma_i}{\lambda_i(1-\mu)} e^{-\rho_i k} + \frac{\eta_2^2 \gamma_i}{\lambda_i(1-\mu)} \big( 1 - e^{-\rho_i k} \big), \quad (16)$$

*where $\lambda_i, \gamma_i$ are the $i$-th Hessian and noise covariance eigenvalues at time $t$, respectively and $\rho_i > 0$ is the decaying factor depending on $(\eta_2, \mu, \lambda_i)$. For clarity, we refer to the first term in equation 16 as the propagation term, since it transports the covariance from the previous equilibrium, and to the second term as the injected term, since it reflects the newly injected gradient noise after the learning-rate change. In addition, with probability at least $1 - \alpha$, we have*

$$\mathbb{E}_{\boldsymbol{w} \sim \mathcal{Q}}[\mathcal{R}_{\mathrm{adv}}(\boldsymbol{w})] \leq \frac{1}{2} \sum_{i=1}^{m} \lambda_i \, \sigma_i^2(t+k) + \frac{1}{2\beta} \left( \frac{\sum_{i=1}^{m} \sigma_i^2(t+k)}{\sigma_{\mathcal{P}}^2} + \frac{\|\boldsymbol{w}^*\|_2^2}{\sigma_{\mathcal{P}}^2} - \sum_{i=1}^{m} \ln \sigma_i^2(t+k) \right)$$
$$+ \hat{\mathcal{R}}_{\mathrm{adv}}(\boldsymbol{w}^*, \mathcal{S}) + \frac{1}{2\beta} \big( -m + m \ln \sigma_{\mathcal{P}}^2 \big) + \frac{\beta C^2}{8|\mathcal{S}|} - \frac{1}{\beta} \ln \alpha. \quad (17)$$

**Remark 4.8.** The decaying factor $\rho_i$ is formally defined in Equation 32 (Appendix B.3). Theorem 4.7 characterizes how the covariance of the posterior $\mathcal{Q}$ changes in the early transition phase to a non-stationary state due to learning rate drop, under the condition that $\mathbf{C}$ and $\mathbf{H}^*$ share the same eigenspace. Similar to our analysis in Remark 4.6, one can expect that $\sum_i \lambda_i \sigma_i^2(t+k)$ and $-\sum_i \ln \sigma_i^2(t+k)$ remain the dominating factors. Note that $\eta_1$ controls the initial equilibrium, $\eta_2$ the final equilibrium, $\lambda_i$ the curvature sensitivity, $\gamma_i$ the noise level, and $k$ the interpolation horizon. In the early or transient phase (small $k$), propagation terms dominate, especially when momentum $\mu$ or step size $\eta_2$ are not small. In contrast, in the late phase (large $k$), the injected term becomes dominant, and $\sigma_i^2(t+k)$ converges to the new stationary covariance determined by $(\eta_2, \mu, \lambda_i)$.

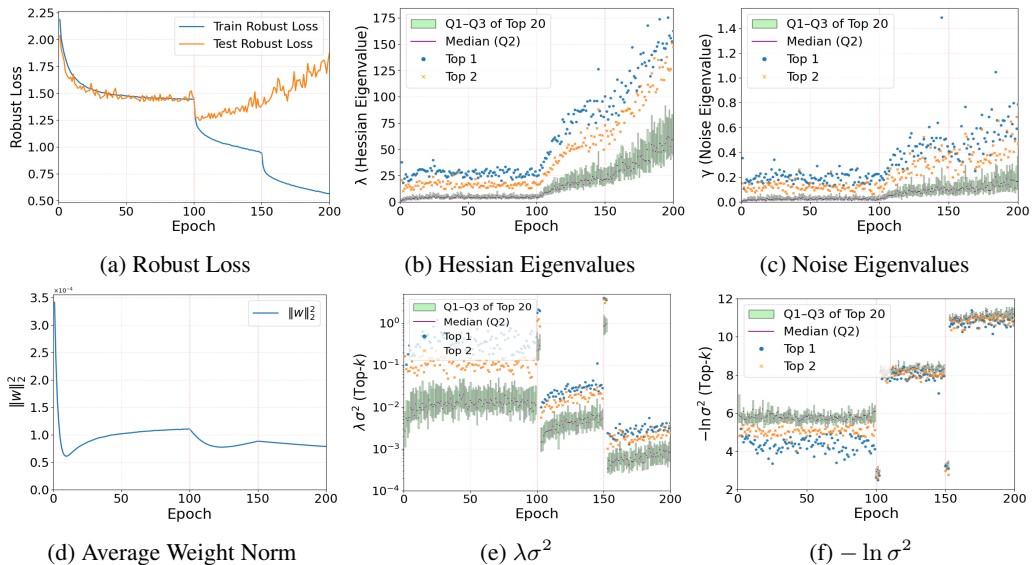

Figure 1: Curves of Hessian and posterior parameters derived from our generalization bounds under standard AT on CIFAR-10. Vertical dashed lines mark learning rate decays at epochs 100 and 150. From epoch 150 to 200, the train robust accuracy increases from 61.81% to 75.96%, while the test robust accuracy decreases from 50.27% to 46.71%, indicating aggravated robust overfitting.

**Remark 4.9.** Our closed-form derivations of the stationary and non-stationary covariance behavior (Theorems 4.5 and 4.7) build on the assumption that $\mathbf{C}$ commutes with $\mathbf{H}^*$. Such a commutative assumption has also been adopted in several previous works for simplifying the analysis of the SGD dynamics (Liu et al., 2021; Ziyin et al., 2021; Suri et al., 2024). One can prove that if $\mathbf{H}^*$ and $\mathbf{C}$ are commutative, they share the same set of eigenbases (see Appendix B.4 for rigorous proof). When the models are optimized through stochastic gradient descent, prior theoretical literature has found evidence that the eigenspace between the loss Hessian and the gradient noise covariance is largely aligned with each other (Ziyin et al., 2021; 2024; Arous et al., 2024; Ziyin et al., 2025). Nevertheless, there are no theoretical guarantees that these two matrices will always be aligned. If the commutative assumption is violated, our robust generalization bounds may become less accurate.

## 5 EXPERIMENTS

In this section, we aim to answer three central questions through empirical evaluations: (i) how Hessian and posterior structure evolve under standard adversarial training (Section 5.1), (ii) how robustness-enhancing methods, such as adversarial weight perturbation (AWP), differ in their behavior (Section 5.2), and (iii) whether the commutativity and alignment assumptions hold (Section 5.3). We approximate the leading Hessian eigenvalues $\{\lambda_i\}$ and the gradient noise eigenvalues $\{\gamma_i\}$ using the procedure described in Appendix E. To be more specific, we compute the top-$k$ Hessian and top-$k$ noise covariance eigenvalues as follows:

$$\forall i \in \{1, 2, \ldots, k\}, \quad \lambda_i = \frac{\boldsymbol{v}_i^\top \mathbf{H} \boldsymbol{v}_i}{\boldsymbol{v}_i^\top \boldsymbol{v}_i}, \quad \gamma_i = \left[ \mathrm{Cov}(\mathbf{V}^\top \boldsymbol{g}_b) \right]_{ii}, \tag{18}$$

where $\boldsymbol{v}_i$ is the $i$-th eigenvector of Hessian $\mathbf{H}$ at the evaluated epoch, $\boldsymbol{g}_b$ stands for the per-batch gradient, and $\mathbf{V} = [\boldsymbol{v}_1, \boldsymbol{v}_2, \ldots, \boldsymbol{v}_k]$ is the eigenspace spanned by the top $k$ Hessian eigenvectors. Detailed settings and additional experiments are provided in Appendices D and F, respectively.

### 5.1 STANDARD ADVERSARIAL TRAINING

We begin by examining the Hessian and posterior covariance structure induced by standard adversarial training (AT) (Madry et al., 2017). Figure 1 summarizes the evolution of robust loss, spectral quantities, and the key PAC-Bayesian terms throughout training. Figures 1b and 1c demonstrate the evolution dynamics of the top-20 Hessian eigenvalues $\{\lambda_i\}$ and the top-20 noise covariance

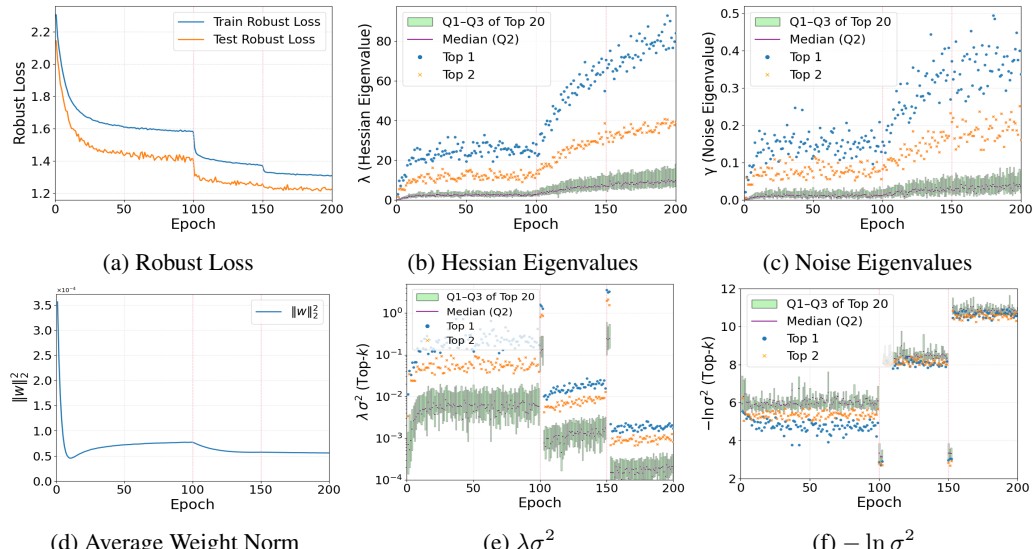

(a) Robust Loss  (b) Hessian Eigenvalues  (c) Noise Eigenvalues

(d) Average Weight Norm  (e) $\lambda\sigma^2$  (f) $-\ln\sigma^2$

Figure 2: Learning curves of Hessian and posterior parameters under AWP on CIFAR-10. From epoch 150 to 200, the train robust accuracy increases from $48.01\%$ to $50.00\%$, while the test robust accuracy increases from $54.03\%$ to $55.49\%$, confirming AWP's strong robust generalization ability.

eigenvalues $\{\gamma_i\}$, respectively. We report their top-2, median, first and third quantile statistics for each training epoch. We observe that Hessian eigenvalues $\{\lambda_i\}$ increase sharply after each learning rate decay at epochs 100 and 150, leading to an increasingly heavy-tailed spectrum, whereas the noise covariance $\{\gamma_i\}$ also increases after learning rate drops, but their growth is much smaller. In addition, Figures 1e and 1f report the behavior of the two dominating factors derived in Section 4: $\{\lambda_i\sigma_i^2\}$ and $\{-\ln\sigma_i^2\}$. While $\lambda\sigma^2$ decreases after each learning rate decay, the increase in $-\ln\sigma^2$ is much larger in magnitude. Consequently, the overall robust generalization bound increases, coinciding with the onset of robust overfitting, depicted in Figure 1a. We also plot the learning curve for the average weight norm $\frac{1}{m}\|\boldsymbol{w}\|_2^2$ in Figure 1d, showing their stability throughout training.

It is worth noting that immediately after the learning rate drops (epochs 100–102 and 150–152), the training dynamics enter a non-stationary transient regime while the iterates remain within the same basin. In this phase, the bound is governed by Theorem 4.7 rather than the stationary form of Theorem 4.5. Consistent with this prediction, we observe that the bound exhibits a temporary downward trend before rising again. This "drop-then-rise" behavior perfectly mirrors the empirical pattern in robust test loss, which decreases briefly after each decay before increasing, thereby reinforcing our theoretical interpretation of robust overfitting.

To study the generalizability of our findings, we choose $\epsilon$ from $\{0, 2, 4, 12, 16\}/255$ with batch size 128, as well as varying the batch size from $\{64, 256\}$ with $\epsilon = 8/255$. Figures 5 to 10 summarize the results: increasing $\epsilon$ mainly affects the curvature spectrum, whereas changing the batch size primarily modulates the noise eigenvalues. Moreover, we conduct additional experiments on CIFAR-100 (Figure 11) and SVHN (Figure 12), as well as with a different architecture, WideResNet-34-10 (Figure 13). In all cases, we consistently observe robust overfitting together with the same qualitative trends: $\lambda$ and $\gamma$ both increase after learning rate drop, posterior covariances $\sigma^2$ shrink, and the dominating components $\lambda\sigma^2$ and $-\ln\sigma^2$ evolve in a manner that explains the emergence of overfitting.

## 5.2 ADVERSARIAL WEIGHT PERTURBATION

We study the dynamics of the key quantities under AWP on CIFAR-10. Figure 2 shows that AWP is highly effective: robust overfitting does not occur, and the test robust loss remains stable throughout training. Compared to standard AT, both the Hessian eigenvalues $\{\lambda_i\}$ and the noise eigenvalues aligned with Hessian directions $\{\gamma_i\}$ are substantially smaller in magnitude. In addition, they are more evenly spread across directions, with the top values showing larger separation, but all remaining concentrated at low levels. This indicates that curvature and gradient-noise variances are both suppressed, leading to posterior variances $\{\sigma_i^2\}$ that are larger than in AT. Consequently, the two dominating terms $\lambda\sigma^2$ and $-\ln\sigma^2$ are significantly reduced, yielding a smaller overall bound. We

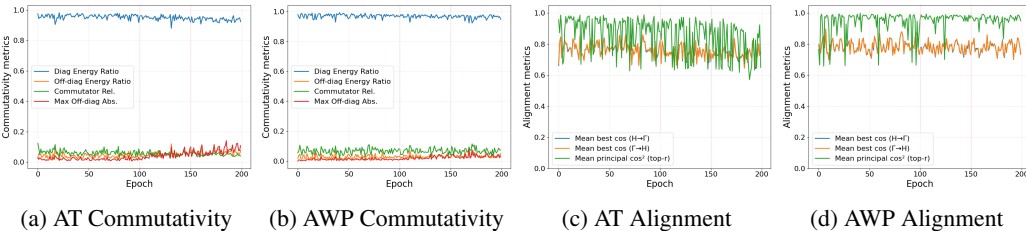

(a) AT Commutativity     (b) AWP Commutativity     (c) AT Alignment     (d) AWP Alignment

Figure 3: Comparison of commutativity and alignment properties under AT and AWP.

observe similar trends for semi-supervised adversarial training algorithms (Figure 14), resulting in consistently smaller bounds.

To facilitate a clearer comparison, we report the numerical differences between AT and AWP in terms of the derived bounds (Tables 1 and 2). Due to space limits, we provide the result tables in Appendix F. Both the aggregated quantity $\sum_i \lambda_i \sigma_i^2$ and $-\sum_i \ln \sigma_i^2$ and the growth from Top-10 to Top-20 are consistently smaller under AWP than under AT. This indicates that AWP's bounds are substantially tighter, perfectly aligning with the empirical finding that AWP significantly out-performs AT, thereby validating our theoretical claims. Moreover, by comparing Table 1 (stationary regime) and Table 2 (initial phase of non-stationary transition), we notice that at the onset of learning rate decay, the bounds decrease for both AT and AWP. This explains why training and test robust losses drop at the beginning of decay. However, once the training re-enters the stationary regime, the bound under AT increases markedly, whereas the bound under AWP remains relatively stable. According to Theorems 4.5 and 4.7, these spectral terms inherently track the evolution of the model's robust generalization capability under adversarial training, explaining both the initial robustness improvement after learning-rate decay and the later robust overfitting phenomenon (Figure 1a).

### 5.3 COMMUTATIVITY AND ALIGNMENT ASSUMPTIONS

We examine the commutativity and alignment assumptions underlying our theoretical analyses. Figure 3 reports our results under standard AT and AWP. Figures 3a and 3b report the metrics evaluating the degree of commutativity, including the ratio of diagonal to off-diagonal energy, the relative norm of the commutator, and the maximum absolute off-diagonal entry. These quantities remain small throughout training, indicating that Hessian and noise covariance are highly commutative. In addition, Figures 3c and 3d measure alignment using cosine similarities between the top eigenvectors. Under AT, we observe consistently strong alignment with some fluctuations, while AWP yields even higher and more stable alignment across epochs. These results confirm that both commutativity and alignment assumptions largely hold empirically, and further highlight that AWP not only suppresses curvature and noise magnitudes but also improves the structural alignment between them.

If the commutativity assumption breaks, then the two matrices cannot be simultaneously diagonalized, and the spectral quantities $\{\lambda_i, \gamma_i\}$ would no longer represent matched curvature–noise pairs. As a consequence, the posterior covariance could exhibit uncontrolled cross-terms, making the PAC-Bayesian bound less interpretable and potentially much looser. Similarly, if alignment were absent, the principal directions of stochastic gradient noise would not coincide with those of curvature. This mismatch would spread noise across directions with different curvatures, leading to inefficient exploration, less reliable stationary approximations, and weaker predictive power of our framework.

## 6 CONCLUSION

We developed a PAC-Bayesian framework that explicitly links the posterior distribution of model parameters to the robust generalization capabilities of adversarially trained models. Using the framework, we connect optimization dynamics and robust generalization by deriving closed-form posterior covariances for two representative training regimes and integrating them into a compact bound. Across diverse empirical configurations, we validate the usefulness of our theoretical results, highlighting the posterior geometry as a unifying principle for understanding and improving adversarially robust generalization. Promising future directions include extending our theoretical analysis to more general settings, such as adaptive learning rate optimizers or non-quadratic loss landscapes, and studying whether our insights can be leveraged to design better robust learning methods.

REPRODUCIBILITY STATEMENT

We have made every effort to ensure the reproducibility of our results. All of our theoretical results are stated with precise assumptions in the main paper, and their proofs are detailed in Appendices A-B. Experimental settings, including datasets, model architectures, training procedures, and spectral estimation details, are provided in Section D. Our code and implementations are available at this anonymized url, which contains scripts to reproduce all the figures and tables reported in the paper.

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

## A  PROOFS OF MAIN THEORETICAL RESULTS IN SECTION 3

### A.1  PROOF OF LEMMA 3.2

To prove Lemma 3.2, we first state a general PAC-Bayes inequality, also known as Catoni's bound (Catoni, 2003).

**Lemma A.1** (PAC-Bayes Bound (Catoni's bound)). *Let $\alpha \in (0,1)$, $\beta > 0$, and $\mathcal{D}$ be any distribution over $\mathcal{X} \times \mathcal{Y}$. Let $\mathcal{W}$ be a parameter space and $\mathcal{P}$ be a data-independent prior on $\mathcal{W}$. Consider any measurable loss $\ell : \mathcal{W} \times \mathcal{X} \times \mathcal{Y} \to [0,C]$ bounded by $C > 0$. Given an i.i.d. sample set $\mathcal{S} = \{(\boldsymbol{x}_i, y_i)\}_{i=1}^{|\mathcal{S}|}$ drawn from $\mathcal{D}$, for any posterior $\mathcal{Q}$ on $\mathcal{W}$, with probability at least $1 - \alpha$ over the draw of $\mathcal{S}$,*

$$\mathbb{E}_{\boldsymbol{w} \sim \mathcal{Q}} \mathbb{E}_{(\boldsymbol{x},y) \sim \mathcal{D}} \big[ \ell(\boldsymbol{w}; \boldsymbol{x}, y) \big] \leq \mathbb{E}_{\boldsymbol{w} \sim \mathcal{Q}} \Big[ \tfrac{1}{|\mathcal{S}|} \sum_{(\boldsymbol{x},y) \in \mathcal{S}} \ell(\boldsymbol{w}; \boldsymbol{x}, y) \Big] + \frac{\beta C^2}{8|\mathcal{S}|} + \frac{\mathrm{KL}(\mathcal{Q} \| \mathcal{P}) + \ln \frac{1}{\alpha}}{\beta}.$$

**Proof of Lemma 3.2.** We instantiate Lemma A.1 with the adversarial loss

$$\tilde{\ell}(\boldsymbol{w}; \boldsymbol{x}, y) := \ell_{\mathrm{adv}}(\boldsymbol{w}, \boldsymbol{x}, y) = \max_{\boldsymbol{\delta} \in B_\varepsilon(0)} \ell(\boldsymbol{w}; \boldsymbol{x} + \boldsymbol{\delta}, y),$$

where the base loss $\ell$ is bounded by $C$ and the perturbation set $B_\varepsilon(0)$ is fixed. Since $\ell \in [0,C]$, the maximization preserves boundedness, so $\tilde{\ell} \in [0,C]$. Measurability follows directly from that of $\ell$ and the continuity of $(\boldsymbol{x}, \boldsymbol{\delta}) \mapsto \ell(\boldsymbol{w}; \boldsymbol{x} + \boldsymbol{\delta}, y)$.

Applying Lemma A.1 to $\tilde{\ell}$ yields

$$\mathbb{E}_{\boldsymbol{w} \sim \mathcal{Q}} \mathbb{E}_{(\boldsymbol{x},y) \sim \mathcal{D}} \big[ \ell_{\mathrm{adv}}(\boldsymbol{w}, \boldsymbol{x}, y) \big] \leq \mathbb{E}_{\boldsymbol{w} \sim \mathcal{Q}} \Big[ \tfrac{1}{|\mathcal{S}|} \sum_{(\boldsymbol{x},y) \in \mathcal{S}} \ell_{\mathrm{adv}}(\boldsymbol{w}, \boldsymbol{x}, y) \Big] + \frac{\beta C^2}{8|\mathcal{S}|} + \frac{\mathrm{KL}(\mathcal{Q} \| \mathcal{P}) + \ln \frac{1}{\alpha}}{\beta}.$$

The left-hand side is exactly $\mathbb{E}_{\boldsymbol{w} \sim \mathcal{Q}}[\mathcal{R}_{\mathrm{adv}}(\boldsymbol{w})]$, and the empirical term is $\mathbb{E}_{\boldsymbol{w} \sim \mathcal{Q}} \big[ \tfrac{1}{|\mathcal{S}|} \sum_{(\boldsymbol{x},y) \in \mathcal{S}} \ell_{\mathrm{adv}}(\boldsymbol{w}, \boldsymbol{x}, y) \big]$. Rearranging the remaining terms gives inequality equation 2, which completes the proof. $\square$

### A.2  PROOF OF LEMMA 3.4

**Proof of Lemma 3.4.** By definition,

$$\mathrm{KL}(\mathcal{Q} \| \mathcal{P}) = \int q(\boldsymbol{w}) \ln \frac{q(\boldsymbol{w})}{p(\boldsymbol{w})} \, \mathrm{d}\boldsymbol{w} = \mathbb{E}_{\boldsymbol{w} \sim \mathcal{Q}}[\ln q(\boldsymbol{w}) - \ln p(\boldsymbol{w})].$$

The density functions of $\mathcal{P}$ and $\mathcal{Q}$ are

$$p(\boldsymbol{w}) = \frac{1}{(2\pi)^{m/2} \sigma_{\mathcal{P}}^m} \exp\left( -\frac{1}{2\sigma_{\mathcal{P}}^2} \boldsymbol{w}^\top \boldsymbol{w} \right),$$

$$q(\boldsymbol{w}) = \frac{1}{(2\pi)^{m/2} \det(\boldsymbol{\Sigma}_{\mathcal{Q}})^{1/2}} \exp\big( -\tfrac{1}{2} (\boldsymbol{w} - \boldsymbol{\mu}_{\mathcal{Q}})^\top \boldsymbol{\Sigma}_{\mathcal{Q}}^{-1} (\boldsymbol{w} - \boldsymbol{\mu}_{\mathcal{Q}}) \big).$$

Taking logs, we obtain

$$\ln p(\boldsymbol{w}) = -\tfrac{m}{2} \ln(2\pi) - m \ln \sigma_{\mathcal{P}} - \tfrac{1}{2\sigma_{\mathcal{P}}^2} \boldsymbol{w}^\top \boldsymbol{w},$$

$$\ln q(\boldsymbol{w}) = -\tfrac{m}{2} \ln(2\pi) - \tfrac{1}{2} \ln \det(\boldsymbol{\Sigma}_{\mathcal{Q}}) - \tfrac{1}{2} (\boldsymbol{w} - \boldsymbol{\mu}_{\mathcal{Q}})^\top \boldsymbol{\Sigma}_{\mathcal{Q}}^{-1} (\boldsymbol{w} - \boldsymbol{\mu}_{\mathcal{Q}}).$$

Hence,

$$\mathrm{KL}(\mathcal{Q} \| \mathcal{P}) = \mathbb{E}_{\boldsymbol{w} \sim \mathcal{Q}} \Big[ -\tfrac{1}{2} \ln \det(\boldsymbol{\Sigma}_{\mathcal{Q}}) - \tfrac{1}{2} (\boldsymbol{w} - \boldsymbol{\mu}_{\mathcal{Q}})^\top \boldsymbol{\Sigma}_{\mathcal{Q}}^{-1} (\boldsymbol{w} - \boldsymbol{\mu}_{\mathcal{Q}}) + m \ln \sigma_{\mathcal{P}} + \tfrac{1}{2\sigma_{\mathcal{P}}^2} \boldsymbol{w}^\top \boldsymbol{w} \Big].$$

Since $w \sim \mathcal{N}(\boldsymbol{\mu}_{\mathcal{Q}}, \boldsymbol{\Sigma}_{\mathcal{Q}})$, the following standard identities hold.

First, by the definition of covariance,

$$\mathbb{E}_{\mathcal{Q}}\big[(\boldsymbol{w} - \boldsymbol{\mu}_{\mathcal{Q}})(\boldsymbol{w} - \boldsymbol{\mu}_{\mathcal{Q}})^{\top}\big] = \boldsymbol{\Sigma}_{\mathcal{Q}}.$$

Multiplying both sides by $\boldsymbol{\Sigma}_{\mathcal{Q}}^{-1}$ and taking the trace yields

$$\mathbb{E}_{\mathcal{Q}}\big[(\boldsymbol{w} - \boldsymbol{\mu}_{\mathcal{Q}})^{\top} \boldsymbol{\Sigma}_{\mathcal{Q}}^{-1}(\boldsymbol{w} - \boldsymbol{\mu}_{\mathcal{Q}})\big] = \mathrm{Tr}\big(\boldsymbol{\Sigma}_{\mathcal{Q}}^{-1}\boldsymbol{\Sigma}_{\mathcal{Q}}\big) = m,$$

since $\mathrm{Tr}(\mathbf{I}_m) = m$.

Second, expanding $\boldsymbol{w}^{\top}\boldsymbol{w}$ around its mean gives

$$\boldsymbol{w}^{\top}\boldsymbol{w} = (\boldsymbol{w} - \boldsymbol{\mu}_{\mathcal{Q}} + \boldsymbol{\mu}_{\mathcal{Q}})^{\top}(\boldsymbol{w} - \boldsymbol{\mu}_{\mathcal{Q}} + \boldsymbol{\mu}_{\mathcal{Q}}) = \|\boldsymbol{w} - \boldsymbol{\mu}_{\mathcal{Q}}\|_2^2 + 2\,\boldsymbol{\mu}_{\mathcal{Q}}^{\top}(\boldsymbol{w} - \boldsymbol{\mu}_{\mathcal{Q}}) + \|\boldsymbol{\mu}_{\mathcal{Q}}\|_2^2.$$

Taking expectations, the cross term vanishes (since $\mathbb{E}[\boldsymbol{w} - \boldsymbol{\mu}_{\mathcal{Q}}] = 0$), so

$$\mathbb{E}_{\mathcal{Q}}\big[\boldsymbol{w}^{\top}\boldsymbol{w}\big] = \mathbb{E}_{\mathcal{Q}}\big[\|\boldsymbol{w} - \boldsymbol{\mu}_{\mathcal{Q}}\|_2^2\big] + \|\boldsymbol{\mu}_{\mathcal{Q}}\|_2^2.$$

By definition of covariance,

$$\mathbb{E}_{\mathcal{Q}}\big[\|\boldsymbol{w} - \boldsymbol{\mu}_{\mathcal{Q}}\|_2^2\big] = \mathrm{Tr}(\boldsymbol{\Sigma}_{\mathcal{Q}})\,.$$

Thus,

$$\mathbb{E}_{\mathcal{Q}}\big[\boldsymbol{w}^{\top}\boldsymbol{w}\big] = \mathrm{Tr}(\boldsymbol{\Sigma}_{\mathcal{Q}}) + \|\boldsymbol{\mu}_{\mathcal{Q}}\|_2^2,$$

where $\|\boldsymbol{\mu}_{\mathcal{Q}}\|_2^2$ denotes the squared Euclidean norm of the posterior mean vector.

Plugging these into the expression above gives

$$\mathrm{KL}(\mathcal{Q}\,\|\,\mathcal{P}) = -\tfrac{1}{2}\ln\det(\boldsymbol{\Sigma}_{\mathcal{Q}}) - \tfrac{1}{2}m + m\ln\sigma_{\mathcal{P}} + \tfrac{1}{2\sigma_{\mathcal{P}}^2}\big(\mathrm{Tr}(\boldsymbol{\Sigma}_{\mathcal{Q}}) + \|\boldsymbol{\mu}_{\mathcal{Q}}\|_2^2\big),$$

which is equivalent to the stated form in Lemma 3.4. $\qquad\square$

### A.3 PROOF OF LEMMA 3.6

**Proof of Lemma 3.6.** Under Assumption 3.5, the empirical adversarial risk admits a quadratic form around the empirical minimizer $\boldsymbol{w}^*$ with Hessian $\mathbf{H}^* \succeq \mathbf{0}$:

$$\hat{\mathcal{R}}_{\mathrm{adv}}(\boldsymbol{w}, \mathcal{S}) = \hat{\mathcal{R}}_{\mathrm{adv}}(\boldsymbol{w}^*, \mathcal{S}) + \tfrac{1}{2}(\boldsymbol{w} - \boldsymbol{w}^*)^{\top}\mathbf{H}^*(\boldsymbol{w} - \boldsymbol{w}^*), \tag{19}$$

where the linear term vanishes because $\nabla_{\boldsymbol{w}}\hat{\mathcal{R}}_{\mathrm{adv}}(\boldsymbol{w}^*, \mathcal{S}) = \mathbf{0}$.

Taking expectation over $\boldsymbol{w} \sim \mathcal{Q} = \mathcal{N}(\boldsymbol{\mu}_{\mathcal{Q}}, \boldsymbol{\Sigma}_{\mathcal{Q}})$ and using equation 19 gives

$$\mathbb{E}_{\boldsymbol{w}\sim\mathcal{Q}}\Big[\hat{\mathcal{R}}_{\mathrm{adv}}(\boldsymbol{w}, \mathcal{S})\Big] = \hat{\mathcal{R}}_{\mathrm{adv}}(\boldsymbol{w}^*, \mathcal{S}) + \tfrac{1}{2}\mathbb{E}_{\boldsymbol{w}\sim\mathcal{Q}}\big[(\boldsymbol{w} - \boldsymbol{w}^*)^{\top}\mathbf{H}^*(\boldsymbol{w} - \boldsymbol{w}^*)\big].$$

Write the second term using the second-moment identity

$$\mathbb{E}_{\boldsymbol{w}\sim\mathcal{Q}}\big[(\boldsymbol{w} - \boldsymbol{w}^*)(\boldsymbol{w} - \boldsymbol{w}^*)^{\top}\big] = \boldsymbol{\Sigma}_{\mathcal{Q}} + (\boldsymbol{\mu}_{\mathcal{Q}} - \boldsymbol{w}^*)(\boldsymbol{\mu}_{\mathcal{Q}} - \boldsymbol{w}^*)^{\top},$$

which holds for any Gaussian (and more generally any distribution with mean $\boldsymbol{\mu}_{\mathcal{Q}}$ and covariance $\boldsymbol{\Sigma}_{\mathcal{Q}}$). Hence,

$$\begin{aligned}
\mathbb{E}_{\boldsymbol{w}\sim\mathcal{Q}}\big[(\boldsymbol{w} - \boldsymbol{w}^*)^{\top}\mathbf{H}^*(\boldsymbol{w} - \boldsymbol{w}^*)\big] &= \mathrm{Tr}\big(\mathbf{H}^*\,\mathbb{E}\big[(\boldsymbol{w} - \boldsymbol{w}^*)(\boldsymbol{w} - \boldsymbol{w}^*)^{\top}\big]\big) \\
&= \mathrm{Tr}(\mathbf{H}^*\boldsymbol{\Sigma}_{\mathcal{Q}}) + \mathrm{Tr}\big(\mathbf{H}^*(\boldsymbol{\mu}_{\mathcal{Q}} - \boldsymbol{w}^*)(\boldsymbol{\mu}_{\mathcal{Q}} - \boldsymbol{w}^*)^{\top}\big) \\
&= \mathrm{Tr}(\mathbf{H}^*\boldsymbol{\Sigma}_{\mathcal{Q}}) + (\boldsymbol{\mu}_{\mathcal{Q}} - \boldsymbol{w}^*)^{\top}\mathbf{H}^*(\boldsymbol{\mu}_{\mathcal{Q}} - \boldsymbol{w}^*),
\end{aligned}$$

where we used identity $\mathrm{Tr}(\mathbf{A}\mathbf{u}\mathbf{u}^{\top}) = \mathbf{u}^{\top}\mathbf{A}\mathbf{u}$.

Combining the above displays yields

$$\mathbb{E}_{\boldsymbol{w}\sim\mathcal{Q}}\big[\hat{\mathcal{R}}_{\mathrm{adv}}(\boldsymbol{w}, \mathcal{S})\big] = \hat{\mathcal{R}}_{\mathrm{adv}}(\boldsymbol{w}^*, \mathcal{S}) + \tfrac{1}{2}(\boldsymbol{\mu}_{\mathcal{Q}} - \boldsymbol{w}^*)^{\top}\mathbf{H}^*(\boldsymbol{\mu}_{\mathcal{Q}} - \boldsymbol{w}^*) + \tfrac{1}{2}\mathrm{Tr}(\mathbf{H}^*\boldsymbol{\Sigma}_{\mathcal{Q}}),$$

which is exactly equation 5. $\qquad\square$

## A.4 PROOF OF COROLLARY 3.8

**Proof of Corollary 3.8.** We start with the most general PAC-Bayesian bound in Lemma 3.2, since it holds for any posterior distribution $\mathcal{Q}$. For any $\beta > 0$ and any $\alpha \in (0, 1)$, with probability at least $1 - \alpha$ over the finite sample set $\mathcal{S}$, we have

$$\mathbb{E}_{\boldsymbol{w} \sim \mathcal{Q}}[\mathcal{R}_{\mathrm{adv}}(\boldsymbol{w})] \leq \mathbb{E}_{\boldsymbol{w} \sim \mathcal{Q}}\left[\frac{1}{|\mathcal{S}|} \sum_{(\boldsymbol{x}, y) \in \mathcal{S}} \ell_{\mathrm{adv}}(\boldsymbol{w}, \boldsymbol{x}, y)\right] + \frac{1}{\beta} \mathrm{KL}(\mathcal{Q} \,\|\, \mathcal{P}) + \frac{\beta C^2}{8|\mathcal{S}|} - \frac{1}{\beta} \ln \alpha. \quad (20)$$

We now instantiate this bound by choosing the posterior to be a mixture of Gaussian distributions specified in Equation 7 and the prior $\mathcal{P} = \mathcal{N}(0, \sigma_{\mathcal{P}}^2 \mathbf{I})$.

**Step 1: Decomposition of the empirical adversarial loss.** By the definition of expectations under mixture distributions, for any measurable function $f : \mathcal{W} \to \mathbb{R}$,

$$\mathbb{E}_{\boldsymbol{w} \sim \mathcal{Q}}[f(\boldsymbol{w})] = \sum_{\ell=1}^{L} \pi_\ell \, \mathbb{E}_{\boldsymbol{w} \sim \mathcal{Q}_\ell}[f(\boldsymbol{w})]. \quad (21)$$

Applying Equation 21 to $f(\boldsymbol{w}) = \hat{\mathcal{R}}_{\mathrm{adv}}(\boldsymbol{w}, \mathcal{S})$ gives

$$\mathbb{E}_{\boldsymbol{w} \sim \mathcal{Q}}\left[\hat{\mathcal{R}}_{\mathrm{adv}}(\boldsymbol{w}, \mathcal{S})\right] = \sum_{\ell=1}^{L} \pi_\ell \, \mathbb{E}_{\boldsymbol{w} \sim \mathcal{Q}_\ell}\left[\hat{\mathcal{R}}_{\mathrm{adv}}(\boldsymbol{w}, \mathcal{S})\right]. \quad (22)$$

For each Gaussian component $\mathcal{Q}_\ell$, note that we assume $\hat{\mathcal{R}}_{\mathrm{adv}}(\boldsymbol{w}, \mathcal{S})$ can be locally approximated around the basin-specific critical point $\boldsymbol{w}_\ell^*$ by a quadratic form with Hessian $\mathbf{H}_\ell^*$. Thus, applying Lemma 3.6 applies to each component, we obtain

$$\mathbb{E}_{\boldsymbol{w} \sim \mathcal{Q}_\ell}\left[\hat{\mathcal{R}}_{\mathrm{adv}}(\boldsymbol{w}, \mathcal{S})\right] = \hat{\mathcal{R}}_{\mathrm{adv}}(\boldsymbol{w}_\ell^*, \mathcal{S}) + \frac{1}{2}(\boldsymbol{\mu}_\ell - \boldsymbol{w}_\ell^*)^\top \mathbf{H}_\ell^*(\boldsymbol{\mu}_\ell - \boldsymbol{w}_\ell^*) + \frac{1}{2}\mathrm{Tr}(\mathbf{H}_\ell^* \boldsymbol{\Sigma}_\ell). \quad (23)$$

Substituting Equation 23 into Equation 22 yields the mixture-expanded empirical loss:

$$\mathbb{E}_{\boldsymbol{w} \sim \mathcal{Q}}[\hat{\mathcal{R}}_{\mathrm{adv}}(\boldsymbol{w}, \mathcal{S})] = \sum_{\ell=1}^{L} \pi_\ell \left[\hat{\mathcal{R}}_{\mathrm{adv}}(\boldsymbol{w}_\ell^*, \mathcal{S}) + \tfrac{1}{2}(\boldsymbol{\mu}_\ell - \boldsymbol{w}_\ell^*)^\top \mathbf{H}_\ell^*(\boldsymbol{\mu}_\ell - \boldsymbol{w}_\ell^*) + \tfrac{1}{2}\mathrm{Tr}(\mathbf{H}_\ell^* \boldsymbol{\Sigma}_\ell)\right].$$

$$(24)$$

**Step 2: KL divergence upper bound for the mixture posterior.** Since KL divergence is convex in its first argument, the mixture posterior $\mathcal{Q} = \sum_{\ell=1}^{L} \pi_\ell \mathcal{Q}_\ell$ satisfies

$$\mathrm{KL}(\mathcal{Q} \,\|\, \mathcal{P}) = \mathrm{KL}\left(\sum_{\ell=1}^{L} \pi_\ell \mathcal{Q}_\ell \,\Big\|\, \mathcal{P}\right) \leq \sum_{\ell=1}^{L} \pi_\ell \mathrm{KL}(\mathcal{Q}_\ell \,\|\, \mathcal{P}), \quad (25)$$

where the inequality follows from the definition KL divergence and the log sum inequality. For each $\mathcal{Q}_\ell = \mathcal{N}(\boldsymbol{\mu}_\ell, \boldsymbol{\Sigma}_\ell)$, Lemma 3.4 provides the closed-form expression:

$$\mathrm{KL}(\mathcal{Q}_\ell \,\|\, \mathcal{P}) = \frac{\mathrm{Tr}(\boldsymbol{\Sigma}_\ell)}{2\sigma_{\mathcal{P}}^2} + \frac{\|\boldsymbol{\mu}_\ell\|_2^2}{2\sigma_{\mathcal{P}}^2} - \frac{m}{2} + \frac{m}{2} \ln \sigma_{\mathcal{P}}^2 - \frac{1}{2} \ln \det \boldsymbol{\Sigma}_\ell. \quad (26)$$

Multiplying Equation 25 by $1/\beta$ and substituting Equation 26 yields

$$\frac{1}{\beta} \mathrm{KL}(\mathcal{Q} \,\|\, \mathcal{P}) \leq \sum_{\ell=1}^{L} \frac{\pi_\ell}{2\beta} \left(\frac{\mathrm{Tr}(\boldsymbol{\Sigma}_\ell)}{\sigma_{\mathcal{P}}^2} + \frac{\|\boldsymbol{\mu}_\ell\|_2^2}{\sigma_{\mathcal{P}}^2} - m + m \ln \sigma_{\mathcal{P}}^2 - \ln \det \boldsymbol{\Sigma}_\ell\right). \quad (27)$$

Finally, we substitute the empirical-loss expansion (Equation 24) and the KL bound (Equation 27) into Equation 20. Noticing that the remaining terms $\frac{\beta C^2}{8|\mathcal{S}|}$ and $-\frac{1}{\beta} \ln \alpha$ do not depend on $\ell$, we obtain

$$\mathbb{E}_{\boldsymbol{w} \sim \mathcal{Q}}[\mathcal{R}_{\mathrm{adv}}(\boldsymbol{w})] \leq \sum_{\ell=1}^{L} \pi_\ell \left[\hat{\mathcal{R}}_{\mathrm{adv}}(\boldsymbol{w}_\ell^*, \mathcal{S}) + \tfrac{1}{2}(\boldsymbol{\mu}_\ell - \boldsymbol{w}_\ell^*)^\top \mathbf{H}_\ell^*(\boldsymbol{\mu}_\ell - \boldsymbol{w}_\ell^*) + \tfrac{1}{2}\mathrm{Tr}(\mathbf{H}_\ell^* \boldsymbol{\Sigma}_\ell)\right]$$

$$+ \sum_{\ell=1}^{L} \frac{\pi_\ell}{2\beta} \left(\frac{\mathrm{Tr}(\boldsymbol{\Sigma}_\ell)}{\sigma_{\mathcal{P}}^2} + \frac{\|\boldsymbol{\mu}_\ell\|_2^2}{\sigma_{\mathcal{P}}^2} - m + m \ln \sigma_{\mathcal{P}}^2 - \ln \det \boldsymbol{\Sigma}_\ell\right) + \frac{\beta C^2}{8|\mathcal{S}|} - \frac{1}{\beta} \ln \alpha.$$

This expression matches exactly the bound asserted in Corollary 3.8, completing the proof. $\qquad \square$

# B    PROOFS OF MAIN THEORETICAL RESULTS IN SECTION 4

## B.1    PROOF OF LEMMA 4.1

**Proof of Lemma 4.1.** *State-space form.* From the updates in equation 9,

$$\boldsymbol{h}_t = \mu\,\boldsymbol{h}_{t-1} + \mathbf{H}^*(\boldsymbol{w}_{t-1} - \boldsymbol{w}^*) + \boldsymbol{\xi}_{t-1},$$
$$\boldsymbol{w}_t - \boldsymbol{w}^* = (\boldsymbol{w}_{t-1} - \boldsymbol{w}^*) - \eta\boldsymbol{h}_t$$
$$= (\mathbf{I} - \eta\mathbf{H}^*)(\boldsymbol{w}_{t-1} - \boldsymbol{w}^*) - \eta\mu\,\boldsymbol{h}_{t-1} - \eta\,\boldsymbol{\xi}_{t-1}.$$

Stacking the two lines with the joint state $\boldsymbol{u}_t := \begin{bmatrix} \boldsymbol{w}_t - \boldsymbol{w}^* \\ \boldsymbol{h}_t \end{bmatrix} \in \mathbb{R}^{2m}$, we obtain the linear time-varying system

$$\boldsymbol{u}_t \;=\; \mathbf{A}\,\boldsymbol{u}_{t-1} + \mathbf{G}\,\boldsymbol{\xi}_{t-1}, \quad \mathbf{A} = \begin{bmatrix} \mathbf{I} - \eta\mathbf{H}^* & -\eta\mu\,\mathbf{I} \\ \mathbf{H}^* & \mu\,\mathbf{I} \end{bmatrix}, \quad \mathbf{G} = \begin{bmatrix} -\eta\mathbf{I} \\ \mathbf{I} \end{bmatrix},$$

which is exactly equation 10.

*Unrolling the trajectory.* We claim that for every $k \geq 1$,

$$\boldsymbol{u}_{t+k} = \mathbf{A}^k\boldsymbol{u}_t + \sum_{j=0}^{k-1} \mathbf{A}^j\mathbf{G}\,\boldsymbol{\xi}_{t+k-1-j}. \tag{28}$$

For $k = 1$ this is just the one-step recursion. Assume equation 28 holds for some $k$; multiply by $\mathbf{A}$ and add $\mathbf{G}\boldsymbol{\xi}_{t+k}$ to get

$$\boldsymbol{u}_{t+k+1} = \mathbf{A}^{k+1}\boldsymbol{u}_t + \sum_{j=0}^{k-1} \mathbf{A}^{j+1}\mathbf{G}\,\boldsymbol{\xi}_{t+k-1-j} + \mathbf{G}\boldsymbol{\xi}_{t+k} = \mathbf{A}^{k+1}\boldsymbol{u}_t + \sum_{j=0}^{k} \mathbf{A}^j\mathbf{G}\,\boldsymbol{\xi}_{t+k-j},$$

i.e., equation 28 with $k \leftarrow k + 1$.

*Joint covariance.* Let $\mathbf{S}_t := \mathrm{Cov}(\boldsymbol{u}_t) = \mathbb{E}[(\boldsymbol{u}_t - \mathbb{E}\boldsymbol{u}_t)(\boldsymbol{u}_t - \mathbb{E}\boldsymbol{u}_t)^\top]$. Using equation 28, bilinearity of covariance, and $\mathrm{Cov}(MX, NY) = M\,\mathrm{Cov}(X,Y)\,N^\top$ for deterministic $M, N$, we have

$$\mathbf{S}_{t+k} = \mathrm{Cov}(\mathbf{A}^k\boldsymbol{u}_t) + \sum_{j=0}^{k-1} \mathrm{Cov}(\mathbf{A}^j\mathbf{G}\,\boldsymbol{\xi}_{t+k-1-j}) + 2\,\mathrm{Cov}\Big(\mathbf{A}^k\boldsymbol{u}_t,\ \sum_{j=0}^{k-1}\mathbf{A}^j\mathbf{G}\,\boldsymbol{\xi}_{t+k-1-j}\Big)$$

$$+ \sum_{0 \leq j \neq \ell \leq k-1} \mathrm{Cov}\big(\mathbf{A}^j\mathbf{G}\,\boldsymbol{\xi}_{t+k-1-j},\ \mathbf{A}^\ell\mathbf{G}\,\boldsymbol{\xi}_{t+k-1-\ell}\big).$$

Assume the mini-batch noises $\{\boldsymbol{\xi}_t\}_{t \geq 0}$ are zero-mean with finite second moments, $\mathbf{C}_t := \mathrm{Cov}(\boldsymbol{\xi}_t) \in \mathbb{R}^{m \times m}$, independent across time, and for each $t$, $\boldsymbol{\xi}_t$ is independent of the past $\sigma$-algebra $\mathcal{F}_t := \sigma(\boldsymbol{u}_0, \boldsymbol{\xi}_0, \ldots, \boldsymbol{\xi}_{t-1})$. Then $\mathrm{Cov}(\boldsymbol{u}_t, \boldsymbol{\xi}_{t+r}) = \mathbf{0}$ for all $r \geq 0$, which nullifies the cross term. For $j \neq \ell$, independence implies $\mathrm{Cov}(\boldsymbol{\xi}_{t+k-1-j}, \boldsymbol{\xi}_{t+k-1-\ell}) = \mathbf{0}$, killing the double sum. Hence

$$\mathbf{S}_{t+k} = \mathbf{A}^k\mathbf{S}_t(\mathbf{A}^k)^\top + \sum_{j=0}^{k-1} \mathbf{A}^j\mathbf{G}\,\mathbf{C}_{t+k-1-j}\,\mathbf{G}^\top(\mathbf{A}^j)^\top. \tag{29}$$

*Parameter covariance via projection.* Let $\mathbf{\Pi} := \begin{bmatrix} \mathbf{I} & \mathbf{0} \end{bmatrix} \in \mathbb{R}^{m \times 2m}$ denote the projection onto the parameter component, so that $\boldsymbol{w}_{t+k} - \boldsymbol{w}^* = \mathbf{\Pi}\,\boldsymbol{u}_{t+k}$. By linearity of covariance under deterministic transforms,

$$\mathbf{\Sigma}_{t+k} = \mathrm{Cov}(\boldsymbol{w}_{t+k} - \boldsymbol{w}^*) = \mathrm{Cov}(\mathbf{\Pi}\boldsymbol{u}_{t+k}) = \mathbf{\Pi}\,\mathbf{S}_{t+k}\,\mathbf{\Pi}^\top.$$

Substituting equation 29 and distributing $\mathbf{\Pi}$ across each term gives

$$\mathbf{\Sigma}_{t+k} = \mathbf{\Pi}\mathbf{A}^k\mathbf{S}_t(\mathbf{A}^k)^\top\mathbf{\Pi}^\top + \sum_{j=0}^{k-1} \big(\mathbf{\Pi}\mathbf{A}^j\mathbf{G}\big)\,\mathbf{C}_{t+k-1-j}\,\big(\mathbf{\Pi}\mathbf{A}^j\mathbf{G}\big)^\top,$$

which is exactly equation 11. $\qquad\square$

### B.2 Proofs of Lemmas in Subsection 4.1

**Proof of Lemma 4.2.** According to Equation 4 and the posterior Gaussian distribution, we have

$$\mathbb{E}_{\boldsymbol{w}\sim\mathcal{Q}}\big[\nabla_{\boldsymbol{w}}\,\hat{\mathcal{R}}_{\mathrm{adv}}(\boldsymbol{w},\mathcal{S})\big] = \mathbb{E}_{\boldsymbol{w}\sim\mathcal{Q}}\big[\mathbf{H}^*(\boldsymbol{w}-\boldsymbol{w}^*)\big] = \mathbf{H}^*(\boldsymbol{\mu}_{\mathcal{Q}}-\boldsymbol{w}^*). \tag{30}$$

Note that we assume the posterior $\mathcal{Q}$ is stationary and is obtained via performing SGD algorithms on $\hat{\mathcal{R}}_{\mathrm{adv}}(\boldsymbol{w},\mathcal{S})$. This immediately implies that the expected adversarial loss gradient has to be a zero vector, namely $\mathbb{E}_{\boldsymbol{w}\sim\mathcal{Q}}\big[\nabla_{\boldsymbol{w}}\,\hat{\mathcal{R}}_{\mathrm{adv}}(\boldsymbol{w},\mathcal{S})\big] = \mathbf{0}$; otherwise, running another step of SGD will break the stationary assumption. Based on Equation 30, we have $\mathbf{H}^*(\boldsymbol{\mu}_{\mathcal{Q}}-\boldsymbol{w}^*) = \mathbf{0}$. Since $\mathbf{H}^*$ is the Hessian of a local optimum, it is therefore a positive definite matrix, which further implies $\boldsymbol{\mu}_{\mathcal{Q}} = \boldsymbol{w}^*$. $\qquad\square$

**Proof of Lemma 4.3.** *Joint Lyapunov equation and projection.* From Lemma 4.1, the stacked state $\boldsymbol{u}_t = \begin{bmatrix}\boldsymbol{w}_t-\boldsymbol{w}^*\\ \boldsymbol{h}_t\end{bmatrix} \in \mathbb{R}^{2m}$ obeys the linear recursion

$$\boldsymbol{u}_t = \mathbf{A}\,\boldsymbol{u}_{t-1} + \mathbf{G}\,\boldsymbol{\xi}_{t-1}, \qquad \mathbf{A} = \begin{bmatrix}\mathbf{I}-\eta\mathbf{H}^* & -\eta\mu\,\mathbf{I}\\ \mathbf{H}^* & \mu\,\mathbf{I}\end{bmatrix}, \quad \mathbf{G} = \begin{bmatrix}-\eta\mathbf{I}\\ \mathbf{I}\end{bmatrix},$$

with zero-mean, temporally independent noise $\{\boldsymbol{\xi}_t\}$ of covariance $\mathbf{C}_t = \mathrm{Cov}(\boldsymbol{\xi}_t)$. Iterating yields $\boldsymbol{u}_t = \mathbf{A}^t\boldsymbol{u}_0 + \sum_{j=0}^{t-1}\mathbf{A}^j\mathbf{G}\,\boldsymbol{\xi}_{t-1-j}$. Taking covariances, using independence across time and independence from the past, gives

$$\boldsymbol{\Sigma}_t^{\mathrm{joint}} = \mathbf{A}^t\boldsymbol{\Sigma}_0^{\mathrm{joint}}(\mathbf{A}^t)^\top + \sum_{j=0}^{t-1}\mathbf{A}^j\mathbf{G}\,\mathbf{C}_{t-1-j}\,\mathbf{G}^\top(\mathbf{A}^j)^\top.$$

Assuming the limits $\boldsymbol{\Sigma}_{\mathrm{joint}} = \lim_{t\to\infty}\boldsymbol{\Sigma}_t^{\mathrm{joint}}$ and $\mathbf{C} = \lim_{t\to\infty}\mathbf{C}_t$ exist and are finite, and the system is mean-square stable (e.g., $\rho(\mathbf{A}) < 1$), we obtain the unique solution of the discrete Lyapunov equation

$$\boldsymbol{\Sigma}_{\mathrm{joint}} = \mathbf{A}\,\boldsymbol{\Sigma}_{\mathrm{joint}}\,\mathbf{A}^\top + \mathbf{G}\,\mathbf{C}\,\mathbf{G}^\top.$$

Let $\boldsymbol{\Pi} = [\,\mathbf{I}\ \mathbf{0}\,] \in \mathbb{R}^{m\times 2m}$ be the projection onto the parameter component; then $\boldsymbol{\Sigma} = \mathrm{Cov}(\boldsymbol{w}_t) = \boldsymbol{\Pi}\,\boldsymbol{\Sigma}_{\mathrm{joint}}\,\boldsymbol{\Pi}^\top$, which proves equation 12.

*Closed form under commutativity.* Assume $\mathbf{H}^*$ and $\mathbf{C}$ are real symmetric and commute. Then there exists an orthogonal $\mathbf{U}$ such that $\mathbf{H}^* = \mathbf{U}\boldsymbol{\Lambda}\mathbf{U}^\top$, $\mathbf{C} = \mathbf{U}\boldsymbol{\Gamma}\mathbf{U}^\top$ with $\boldsymbol{\Lambda} = \mathrm{diag}(\lambda_1,\ldots,\lambda_m)$ and $\boldsymbol{\Gamma} = \mathrm{diag}(\gamma_1,\ldots,\gamma_m)$. Define the block-orthogonal transform $\mathbf{T} := \mathrm{diag}(\mathbf{U},\mathbf{U})$ and rotated state/noise $\boldsymbol{z}_t := \mathbf{T}^\top\boldsymbol{u}_t$, $\boldsymbol{\zeta}_t := \mathbf{U}^\top\boldsymbol{\xi}_t$. In this eigenbasis the dynamics decouple:

$$\boldsymbol{z}_t = \widetilde{\mathbf{A}}\,\boldsymbol{z}_{t-1} + \widetilde{\mathbf{G}}\,\boldsymbol{\zeta}_{t-1}, \quad \widetilde{\mathbf{A}} = \begin{bmatrix}\mathbf{I}-\eta\boldsymbol{\Lambda} & -\eta\mu\,\mathbf{I}\\ \boldsymbol{\Lambda} & \mu\,\mathbf{I}\end{bmatrix}, \quad \widetilde{\mathbf{G}} = \begin{bmatrix}-\eta\mathbf{I}\\ \mathbf{I}\end{bmatrix}, \ \mathrm{Cov}(\boldsymbol{\zeta}_t) = \boldsymbol{\Gamma}.$$

Hence each eigendirection $i$ follows a $2\times 2$ "heavy-ball" system with curvature $\lambda = \lambda_i$ and noise variance $\gamma = \gamma_i$:

$$\mathbf{A}(\lambda) = \begin{bmatrix}1-\eta\lambda & -\eta\mu\\ \lambda & \mu\end{bmatrix}, \quad \mathbf{G} = \begin{bmatrix}-\eta\\ 1\end{bmatrix}, \quad \mathbf{Q}(\gamma) = \mathbf{G}\,\gamma\,\mathbf{G}^\top = \begin{bmatrix}\gamma\eta^2 & -\gamma\eta\\ -\gamma\eta & \gamma\end{bmatrix}.$$

Let $\mathbf{S} = \begin{bmatrix}x & y\\ y & z\end{bmatrix}$ be the stationary joint covariance in this mode, solving $\mathbf{S} = \mathbf{A}(\lambda)\,\mathbf{S}\,\mathbf{A}(\lambda)^\top + \mathbf{Q}(\gamma)$. Solving the resulting linear system in $x,y,z$ (unique under stability) gives the parameter variance

$$x = \frac{\gamma\,\eta(1+\mu)}{\lambda(1-\mu)\big(2(1+\mu)-\eta\lambda\big)}.$$

Therefore, in the eigenbasis the stationary parameter covariance is diagonal with entries

$$\frac{\eta}{1-\mu}\cdot\frac{\gamma_i}{\lambda_i\big(2-\frac{\eta}{1+\mu}\lambda_i\big)} = \Big(\lambda_i\big(2-\tfrac{\eta}{1+\mu}\lambda_i\big)\Big)^{-1}\frac{\eta}{1-\mu}\,\gamma_i,$$

and conjugating back by $\mathbf{U}$ yields

$$\boldsymbol{\Sigma} = \left[ \mathbf{H}^* \left( 2\mathbf{I} - \frac{\eta}{1+\mu}\mathbf{H}^* \right) \right]^{-1} \frac{\eta}{1-\mu}\,\mathbf{C},$$

which is exactly equation 13.

*Stability condition.* For the scalar mode, the characteristic polynomial of $\mathbf{A}(\lambda)$ is $t^2 - (1 - \eta\lambda + \mu)t + \mu$. All roots lie in the open unit disk iff $|\mu| < 1$ and $0 < \eta\lambda < 2(1+\mu)$ (e.g., Jury criterion). Since $\mu \in [0,1)$, this equivalently requires $0 < \frac{\eta}{1+\mu}\lambda_i < 2$ for every $i$. Under this condition, the stationary covariance exists and is finite. $\qquad\square$

### B.3 FORMAL STATEMENT & DETAILED DERIVATIONS OF THEOREM 4.7

We first present a formal version of the PAC-Bayes bound after a learning-rate change, and from this we derive the informal approximation in Theorem 4.7. The full proof of this formal statement is deferred to Appendix C, while here we focus on how it leads to the informal result.

**Theorem B.1** (Robust Generalization after Learning Rate Decay, Formal). *Assume that $\mathbf{H}^*$ and $\mathbf{C}$ are real symmetric matrices which commute, and hence are simultaneously diagonalizable by an orthogonal matrix. In particular, there exists $\mathbf{U}$ such that*

$$\mathbf{H}^* = \mathbf{U}\boldsymbol{\Lambda}\mathbf{U}^\top, \qquad \mathbf{C} = \mathbf{U}\boldsymbol{\Gamma}\mathbf{U}^\top,$$

*with $\boldsymbol{\Lambda} = \mathrm{diag}(\lambda_1, \ldots, \lambda_m)$ and $\boldsymbol{\Gamma} = \mathrm{diag}(\gamma_1, \ldots, \gamma_m)$. Suppose the post-switch noise is time-invariant. Let $\sigma_i^2(t+k)$ denote the modal variances given explicitly by*

$$\sigma_i^2(t+k) = p_{i,k}^2\, x_i^{(1)} + 2\, p_{i,k}\, q_{i,k}\, y_i^{(1)} + q_{i,k}^2\, z_i^{(1)}$$

$$+ \frac{\eta_2^2\, \gamma_i}{\left( r_{i,+}^{(2)} - r_{i,-}^{(2)} \right)^2} \left[ \frac{(r_{i,+}^{(2)})^2\left( 1 - (r_{i,+}^{(2)})^{2k} \right)}{1 - (r_{i,+}^{(2)})^2} + \frac{(r_{i,-}^{(2)})^2\left( 1 - (r_{i,-}^{(2)})^{2k} \right)}{1 - (r_{i,-}^{(2)})^2} - \frac{2\,\mu\,(1 - \mu^k)}{1 - \mu} \right],$$

*where*

$$r_{i,\pm}^{(2)} = \frac{1 - \eta_2\lambda_i + \mu \pm \sqrt{(1 - \eta_2\lambda_i + \mu)^2 - 4\mu}}{2},$$

$$p_{i,k} = \frac{(r_{i,+}^{(2)} - \mu)\,(r_{i,+}^{(2)})^k + (\mu - r_{i,-}^{(2)})\,(r_{i,-}^{(2)})^k}{r_{i,+}^{(2)} - r_{i,-}^{(2)}}, \qquad q_{i,k} = -\frac{\eta_2\mu\left( (r_{i,+}^{(2)})^k - (r_{i,-}^{(2)})^k \right)}{r_{i,+}^{(2)} - r_{i,-}^{(2)}},$$

*and the pre-switch stationary joint entries $(x_i^{(1)}, y_i^{(1)}, z_i^{(1)})$ under $(\eta_1, \mu)$ are*

$$x_i^{(1)} = \frac{\eta_1\gamma_i(1 + \mu)}{\lambda_i(1 - \mu)\left[ 2(1 + \mu) - \eta_1\lambda_i \right]},$$

$$y_i^{(1)} = -\frac{\eta_1\gamma_i}{(1 - \mu)\left[ 2(1 + \mu) - \eta_1\lambda_i \right]}, \qquad z_i^{(1)} = \frac{2\,\gamma_i}{(1 - \mu)\left[ 2(1 + \mu) - \eta_1\lambda_i \right]}.$$

*Then, for any $\beta > 0$ and $\alpha \in (0,1)$, with probability at least $1 - \alpha$,*

$$\mathbb{E}_{\boldsymbol{w}\sim\mathcal{Q}}[\mathcal{R}_{\mathrm{adv}}(\boldsymbol{w})] \leq \frac{1}{2}\sum_{i=1}^m \lambda_i\, \sigma_i^2(t+k)$$

$$+ \frac{1}{2\beta}\left( \frac{\sum_{i=1}^m \sigma_i^2(t+k)}{\sigma_\mathcal{P}^2} + \frac{\|\boldsymbol{w}^*\|_2^2}{\sigma_\mathcal{P}^2} - m + m\ln\sigma_\mathcal{P}^2 - \sum_{i=1}^m \ln\sigma_i^2(t+k) \right)$$

$$+ \hat{\mathcal{R}}_{\mathrm{adv}}(\boldsymbol{w}^*, \mathcal{S}) + \frac{\beta C^2}{8|\mathcal{S}|} - \frac{1}{\beta}\ln\alpha. \qquad (31)$$

*Stability requires $0 < \frac{\eta_\ell}{1+\mu}\lambda_i < 2$ for both $\ell \in \{1,2\}$ and all $i$.*

**Proof of Theorem 4.7.** From Theorem B.1, the robust generalization bound involves the modal variances $\sigma_i^2(t+k)$, whose exact closed-form expression is

$$\sigma_i^2(t+k) = p_{i,k}^2\, x_i^{(1)} + 2\, p_{i,k}\, q_{i,k}\, y_i^{(1)} + q_{i,k}^2\, z_i^{(1)}$$
$$+ \frac{\eta_2^2\, \gamma_i}{\left(r_{i,+}^{(2)} - r_{i,-}^{(2)}\right)^2} \left[ \frac{(r_{i,+}^{(2)})^2\left(1 - (r_{i,+}^{(2)})^{2k}\right)}{1 - (r_{i,+}^{(2)})^2} + \frac{(r_{i,-}^{(2)})^2\left(1 - (r_{i,-}^{(2)})^{2k}\right)}{1 - (r_{i,-}^{(2)})^2} - \frac{2\,\mu\,(1 - \mu^k)}{1 - \mu} \right].$$

Although algebraically explicit, this formula is rather involved. To obtain a more interpretable approximation, we analyze its asymptotic structure.

The expression shows that $\sigma_i^2(t+k)$ evolves from its pre-switch stationary level $\sigma_i^2(t)$, attained under $(\eta_1, \mu)$, towards the post-switch stationary level $\sigma_{i,\star}^2(\eta_2)$ corresponding to $(\eta_2, \mu)$. Since $\mathbf{H}^*$ and $\mathbf{C}$ commute, they are simultaneously diagonalizable. In the joint eigenbasis, the global recursion

$$\boldsymbol{u}_t = \mathbf{A}_2 \boldsymbol{u}_{t-1} + \mathbf{G}_2 \boldsymbol{\xi}_{t-1}$$

decouples across eigendirections. Along the $i$-th eigenvector, the dynamics reduce to a $2 \times 2$ system for the state $[\, w_t^{(i)} - w_i^*,\, h_t^{(i)}\,]^\top$, with transition matrix

$$\mathbf{A}_i(\eta_2) = \begin{bmatrix} 1 - \eta_2\lambda_i + \mu & -\mu \\ 1 & 0 \end{bmatrix}.$$

The eigenvalues of $\mathbf{A}_i(\eta_2)$ are

$$r_{i,\pm}^{(2)} = \frac{1 - \eta_2\lambda_i + \mu \pm \sqrt{(1 - \eta_2\lambda_i + \mu)^2 - 4\mu}}{2}.$$

In the stable regime $0 < \frac{\eta_2}{1+\mu}\lambda_i < 2$, the characteristic polynomial $r^2 - (1 - \eta_2\lambda_i + \mu)r + \mu = 0$ has roots strictly inside the unit disk, i.e. $|r_{i,\pm}^{(2)}| < 1$. This stability condition ensures that the trajectory remains bounded. Moreover, the deviation from the stationary covariance can be expressed as a linear combination of $(r_{i,+}^{(2)})^k$ and $(r_{i,-}^{(2)})^k$, so the error term decays geometrically at rate $\max\{|r_{i,+}^{(2)}|, |r_{i,-}^{(2)}|\} < 1$. Defining

$$e^{-\rho_i} := \max\{|r_{i,+}^{(2)}|, |r_{i,-}^{(2)}|\},$$

which is equivalently written as

$$\rho_i = -\ln\left( \max\{|r_{i,+}^{(2)}|, |r_{i,-}^{(2)}|\} \right) > 0, \tag{32}$$

we obtain the exponential interpolation

$$\sigma_i^2(t+k) \approx \sigma_i^2(t)\, e^{-\rho_i k} + \sigma_{i,\star}^2(\eta_2)\, (1 - e^{-\rho_i k}).$$

The two endpoints admit clean leading-order approximations that are consistent with the exact stationary solutions in the small-step regime. Before the switch, the stationary variance scales as

$$\sigma_i^2(t) \approx \frac{\eta_1 \gamma_i}{\lambda_i(1 - \mu)},$$

which is linear in $\eta_1$. After the switch, the stationary solution under $(\eta_2, \mu)$ scales as

$$\sigma_{i,\star}^2(\eta_2) \approx \frac{\eta_2^2 \gamma_i}{\lambda_i(1 - \mu)},$$

which is quadratic in $\eta_2$ because the effective noise accumulation under momentum involves the squared step size.

Substituting these two approximations into the interpolation formula yields

$$\sigma_i^2(t+k) \approx \frac{\eta_1 \gamma_i}{\lambda_i(1 - \mu)}\, e^{-\rho_i k} + \frac{\eta_2^2 \gamma_i}{\lambda_i(1 - \mu)}\left(1 - e^{-\rho_i k}\right),$$

which is exactly the expression stated in equation 16. Finally, substituting this $\sigma_i^2(t+k)$ into the bound of Theorem B.1 gives equation 17, completing the proof. $\qquad\square$

### B.4    PROOF OF THE STATEMENT IN REMARK 4.9

**Proof of Remark 4.9.** Assume $\mathbf{H}, \mathbf{C} \in \mathbb{R}^{m \times m}$ are real symmetric and commute, i.e. $\mathbf{HC} = \mathbf{CH}$. By the spectral theorem, there exists an orthonormal decomposition of $\mathbb{R}^m$ into the eigenspaces of $\mathbf{H}$:

$$\mathbb{R}^m = \bigoplus_{\lambda} E_{\lambda}, \qquad E_{\lambda} := \ker(\mathbf{H} - \lambda \mathbf{I}).$$

We first show that each $E_{\lambda}$ is invariant under $\mathbf{C}$. If $v \in E_{\lambda}$, then $\mathbf{H}v = \lambda v$. Using commutativity,

$$\mathbf{H}(\mathbf{C}v) = \mathbf{C}(\mathbf{H}v) = \mathbf{C}(\lambda v) = \lambda(\mathbf{C}v),$$

so $\mathbf{C}v \in E_{\lambda}$. Thus $\mathbf{C}(E_{\lambda}) \subseteq E_{\lambda}$ for every eigenvalue $\lambda$ of $\mathbf{H}$.

For each $\lambda$, the restriction $\mathbf{C}|_{E_{\lambda}}$ is symmetric with respect to the Euclidean inner product and hence admits an orthonormal eigenbasis of $E_{\lambda}$. Collecting these bases across all $\lambda$ yields an orthonormal basis of $\mathbb{R}^m$ consisting of vectors that lie in some $E_{\lambda}$ and are simultaneously eigenvectors of $\mathbf{C}$. Each such vector is therefore an eigenvector of both $\mathbf{H}$ and $\mathbf{C}$.

Let $\mathbf{U}$ be the orthogonal matrix with these basis vectors as columns. In this basis both $\mathbf{H}$ and $\mathbf{C}$ are diagonal:

$$\mathbf{U}^\top \mathbf{H} \mathbf{U} = \mathrm{diag}(\lambda_1, \ldots, \lambda_m), \qquad \mathbf{U}^\top \mathbf{C} \mathbf{U} = \mathrm{diag}(\gamma_1, \ldots, \gamma_m).$$

Hence $\mathbf{H}$ and $\mathbf{C}$ are simultaneously diagonalizable and share an orthonormal set of eigenvectors.  $\square$

## C    PROOF OF THEOREM B.1

**Proof of Theorem B.1.** *Decoupling under commutativity.* Since $\mathbf{H}^*$ and $\mathbf{C}$ are real symmetric and commute, there exists an orthogonal matrix $\mathbf{U}$ such that

$$\mathbf{H}^* = \mathbf{U}\mathbf{\Lambda}\mathbf{U}^\top, \qquad \mathbf{C} = \mathbf{U}\mathbf{\Gamma}\mathbf{U}^\top,$$

with $\mathbf{\Lambda} = \mathrm{diag}(\lambda_1, \ldots, \lambda_m)$ and $\mathbf{\Gamma} = \mathrm{diag}(\gamma_1, \ldots, \gamma_m)$. Transforming to this joint eigenbasis decouples the global $2m$-dimensional recursion into $m$ independent $2 \times 2$ linear systems, one for each mode $i$.

*Unrolling the trajectory.* Along the $i$-th eigendirection, the state is $\mathbf{x}_s^{(i)} = [\, w_s^{(i)} - w_i^*, \, h_s^{(i)} \,]^\top$ and evolves as

$$\mathbf{x}_s^{(i)} = \mathbf{A}_i(\eta_2)\, \mathbf{x}_{s-1}^{(i)} + \boldsymbol{\xi}_{s-1}^{(i)}, \qquad \mathbf{A}_i(\eta_2) = \begin{bmatrix} 1 - \eta_2 \lambda_i + \mu & -\mu \\ 1 & 0 \end{bmatrix}, \quad \boldsymbol{\xi}_{s-1}^{(i)} = \begin{bmatrix} -\eta_2 \zeta_{s-1}^{(i)} \\ 0 \end{bmatrix},$$

with noise variance $\mathbb{E}[(\zeta_{s-1}^{(i)})^2] = \gamma_i$. By iteration,

$$\mathbf{\Sigma}_{t+k}^{(i)} = \mathbf{A}_i(\eta_2)^k\, \mathbf{\Sigma}_t^{(i)} \big(\mathbf{A}_i(\eta_2)^k\big)^\top + \sum_{j=0}^{k-1} \mathbf{A}_i(\eta_2)^j\, \mathbf{Q}_i \big(\mathbf{A}_i(\eta_2)^j\big)^\top, \qquad \mathbf{Q}_i = \eta_2^2 \gamma_i\, \mathbf{e}_1 \mathbf{e}_1^\top.$$

*Pre-switch stationary initialization.* At time $t$, the covariance is stationary under $(\eta_1, \mu)$; the unique solution to the Lyapunov equation gives

$$x_i^{(1)} = \frac{\eta_1 \gamma_i (1 + \mu)}{\lambda_i (1 - \mu)\,[\,2(1 + \mu) - \eta_1 \lambda_i\,]},$$

$$y_i^{(1)} = -\frac{\eta_1 \gamma_i}{(1 - \mu)\,[\,2(1 + \mu) - \eta_1 \lambda_i\,]},$$

$$z_i^{(1)} = \frac{2\,\gamma_i}{(1 - \mu)\,[\,2(1 + \mu) - \eta_1 \lambda_i\,]}.$$

Thus $\mathbf{\Sigma}_t^{(i)} = S_i^{(1)} = \begin{bmatrix} x_i^{(1)} & y_i^{(1)} \\ y_i^{(1)} & z_i^{(1)} \end{bmatrix}$.

*Closed form of the modal variance.* Diagonalizing $\mathbf{A}_i(\eta_2)$ gives roots

$$r_{i,\pm}^{(2)} = \frac{1 - \eta_2 \lambda_i + \mu \pm \sqrt{(1 - \eta_2 \lambda_i + \mu)^2 - 4\mu}}{2},$$

and the first-row coefficients

$$p_{i,k} = \frac{(r_{i,+}^{(2)} - \mu)(r_{i,+}^{(2)})^k + (\mu - r_{i,-}^{(2)})(r_{i,-}^{(2)})^k}{r_{i,+}^{(2)} - r_{i,-}^{(2)}}, \qquad q_{i,k} = -\frac{\eta_2 \mu \left((r_{i,+}^{(2)})^k - (r_{i,-}^{(2)})^k\right)}{r_{i,+}^{(2)} - r_{i,-}^{(2)}}.$$

Similarly, define $v_{i,j} = \frac{(r_{i,-}^{(2)})^{j+1} - (r_{i,+}^{(2)})^{j+1}}{r_{i,+}^{(2)} - r_{i,-}^{(2)}}$. Then the variance of parameter $w^{(i)}$ is

$$\sigma_i^2(t+k) = p_{i,k}^2 \, x_i^{(1)} + 2\, p_{i,k}\, q_{i,k}\, y_i^{(1)} + q_{i,k}^2\, z_i^{(1)} + \eta_2^2 \gamma_i \sum_{j=0}^{k-1} v_{i,j}^2$$

$$= p_{i,k}^2 \, x_i^{(1)} + 2\, p_{i,k}\, q_{i,k}\, y_i^{(1)} + q_{i,k}^2\, z_i^{(1)}$$

$$+ \frac{\eta_2^2 \gamma_i}{(r_{i,+}^{(2)} - r_{i,-}^{(2)})^2} \left[ \frac{(r_{i,+}^{(2)})^2(1 - (r_{i,+}^{(2)})^{2k})}{1 - (r_{i,+}^{(2)})^2} + \frac{(r_{i,-}^{(2)})^2(1 - (r_{i,-}^{(2)})^{2k})}{1 - (r_{i,-}^{(2)})^2} - \frac{2\mu(1 - \mu^k)}{1 - \mu} \right],$$

which matches the statement of Theorem B.1.

*Substitution into the PAC-Bayes framework.* The quadratic expansion of the adversarial loss and the Catoni PAC-Bayes bound with Gaussian KL divergence were already established in Theorem 3.7. Plugging in the modal variances $\{\sigma_i^2(t+k)\}_{i=1}^m$ derived above into that general bound yields inequality equation 31.

*Stability.* Finally, the condition $0 < \frac{\eta_\ell}{1+\mu} \lambda_i < 2$ for both $\ell \in \{1, 2\}$ and all $i$ is precisely the Jury stability criterion for the heavy-ball characteristic polynomial, which guarantees $|r_{i,\pm}^{(2)}| < 1$ and hence convergence of the geometric series in the covariance expression. $\qquad \square$

# D  EXPERIMENT SETTINGS

We use the CIFAR-10 dataset (Krizhevsky et al., 2009) with standard train splits. Inputs are scaled to $[0, 1]$ and normalized channel-wise. Data augmentation includes random cropping with 4-pixel padding and random horizontal flipping. In additional experiments, we also evaluate on CIFAR-100 (Krizhevsky et al., 2009) and SVHN(Netzer et al., 2011) to verify the generality of our findings.

We consider PreActResNet-18 (He et al., 2016) as the backbone model. Training uses momentum SGD ($\mu = 0.9$) with weight decay $5 \times 10^{-4}$. The initial learning rate is $0.1$, decayed to $0.01$ at epoch 100 and to $0.001$ at epoch 150, following a piecewise schedule. Training runs for 200 epochs with batch size 128. In supplementary experiments, we additionally use WideResNet-34-10 (Zagoruyko & Komodakis, 2016) as an alternative architecture.

For robustness, we adopt PGD adversarial training (Madry et al., 2017) as the default baseline. We generate $\ell_\infty$ adversarial perturbations with $\epsilon = 8/255$, step size $2/255$, 10 attack iterations, and 1 random restart. For computational efficiency, we approximate curvature and gradient-noise statistics using a small number of randomly sampled mini-batches. We average the loss over $m = 128$ batches when estimating the top-$k$ Hessian eigenpairs (with $k = 20$), perform 30 power iterations for each eigenpair, and compute gradient covariances from 128 batches to obtain noise statistics. This stochastic approximation is sufficient to capture the dominant spectral structure while keeping runtime feasible.

# E  SPECTRAL ESTIMATION DETAILS

To obtain the Hessian spectrum, we approximate the Hessian on a vector $v$ via the identity

$$\mathbf{H}\boldsymbol{v} = \nabla_{\boldsymbol{w}}\Big(\nabla_{\boldsymbol{w}}\mathcal{L}(\boldsymbol{w})^\top \boldsymbol{v}\Big),$$

which can be computed efficiently by automatic differentiation without explicitly forming $\mathbf{H}$. We then apply power iteration with Gram–Schmidt orthogonalization to extract the top-$k$ eigenvectors $\{\boldsymbol{v}_i\}$, and estimate their associated eigenvalues using the Rayleigh quotient,

$$\lambda_i = \frac{\boldsymbol{v}_i^\top (\mathbf{H}\boldsymbol{v}_i)}{\boldsymbol{v}_i^\top \boldsymbol{v}_i}.$$

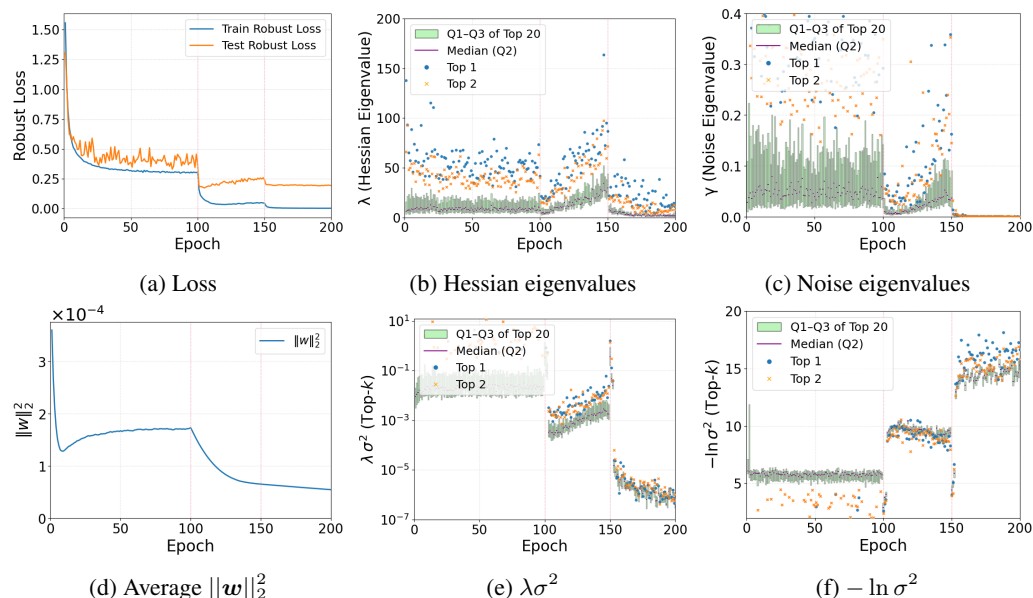

Figure 4: Additional results for standard training (i.e., $\epsilon = 0$) batch size 128 on CIFAR-10. From epoch 150 to 200, the train robust accuracy increases from 98.47% to 99.98%, and the test robust accuracy increases from 93.09% to 95.13%.

This procedure is consistent with established practice for large-scale curvature estimation (Dangel et al., 2019; Yao et al., 2020). For the posterior structure, we consider the stochastic gradients at the mini-batch level, $g_b = \nabla_w \mathcal{L}_b(w)$. Projecting these gradients onto the subspace spanned by the leading Hessian eigenvectors $\mathbf{V} = [v_1, \ldots, v_k]$ yields the projected quantities $p_b = \mathbf{V}^\top g_b \in \mathbb{R}^k$. Their covariance matrix is $\Gamma = \mathrm{Cov}[p_b]$, whose diagonal entries $\gamma_i = \Gamma_{ii}$ quantify the variance of stochastic gradients along the principal curvature directions. By construction, this definition ensures that $\lambda_i$ characterizes curvature while $\gamma_i$ represents the corresponding noise magnitude in the same eigendirections. This approach follows recent empirical observations that gradient-noise covariance tends to align with the Hessian eigenspectrum in neural networks (Jastrzebski et al., 2018; Ziyin et al., 2025), allowing for a direct analysis of curvature–noise interactions.

# F ADDITIONAL EXPERIMENTS AND ANALYSES

## F.1 GENERALIZABILITY STUDY

Figure 4 presents additional results for standard training on CIFAR-10. Here, both the Hessian eigenvalues $\{\lambda_i\}$ and noise covariance eigenvalues $\{\gamma_i\}$ remain much smaller and evolve more smoothly, without the sharp curvature escalation like adversarial training. For completeness, we also vary the perturbation strength $\epsilon$ to evaluate their impact on the Hessian and noise eigenvalues. Figures 5-8 present the results for $\epsilon \in \{2/255, 4/255, 12/255, 16/255\}$. We also run experiments with varying batch sizes from $\{64, 256\}$, with the corresponding results shown in Figures 9 and 10.

In addition, we conduct experiments to test the generalizability of our findings across other image benchmarks, including CIFAR-100 (Figure 11) and SVHN (Figure 12), as well as different learning algorithms, such as adversarial training on a larger WideResNet-34-10 architecture (Figure 13) and semi-supervised adversarial training (Figure 14).

Overall, the ablation studies across perturbation radii, batch sizes, datasets, and architectures reveal a highly consistent picture. Increasing the perturbation strength $\epsilon$ primarily amplifies the curvature of the adversarial loss landscape, leading to larger Hessian eigenvalues with only mild changes in gradient noise levels. In contrast, varying the batch size mainly rescales the noise eigenvalues $\{\gamma_i\}$ while leaving the curvature spectrum largely unchanged. These orthogonal effects precisely match the roles of curvature and noise predicted by our PAC-Bayesian analysis.

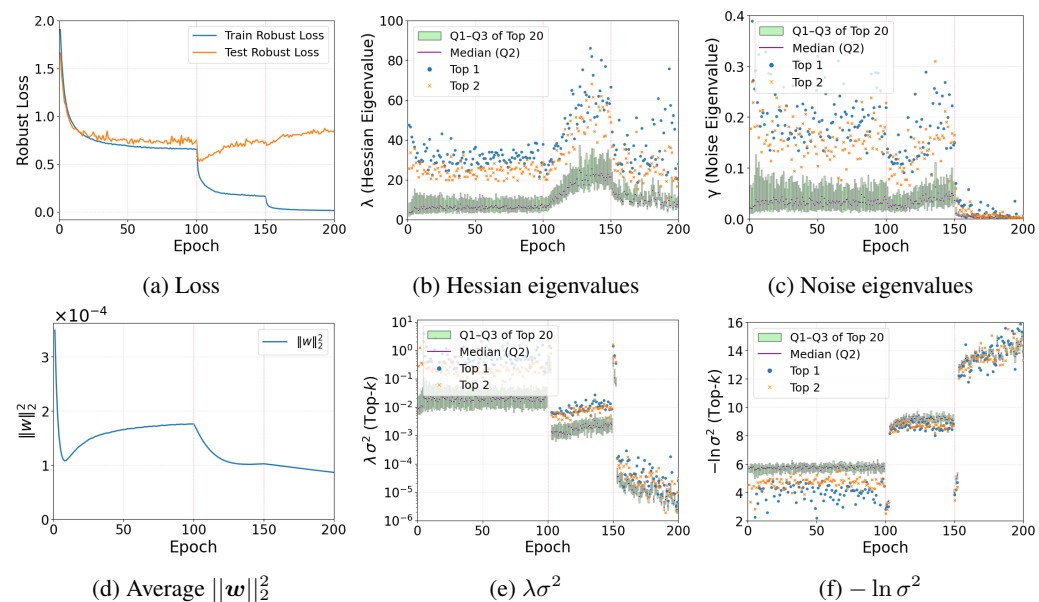

Figure 5: Additional results for adversarial training with batch size 128, $\epsilon = 2/255$ and PGD step size $1/255$ on CIFAR-10. From epoch 150 to 200, the train robust accuracy increases from $93.73\%$ to $99.48\%$, and the test robust accuracy increases from $78.17\%$ to $80.07\%$.

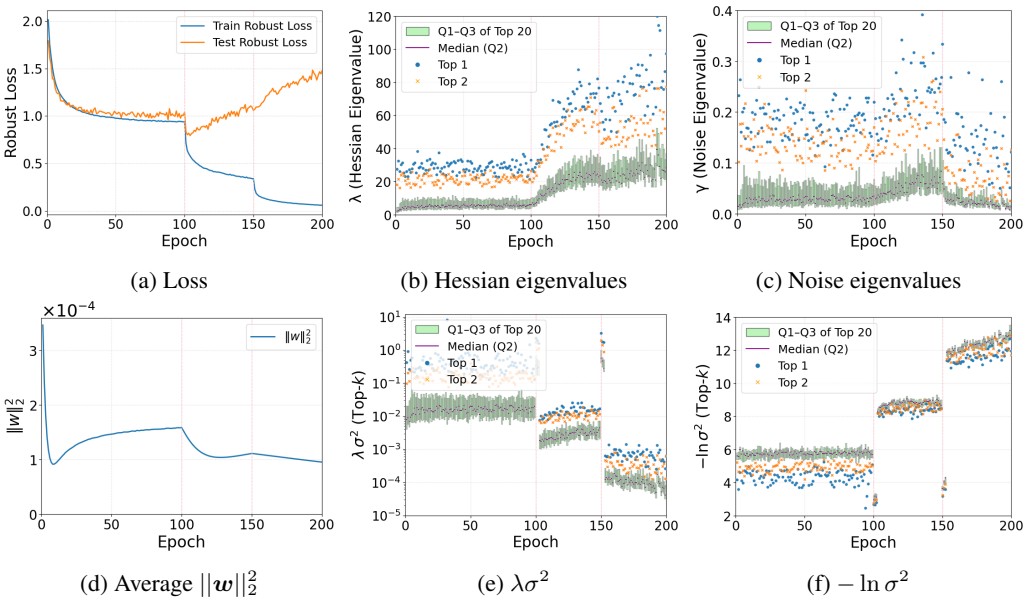

Figure 6: Additional results for adversarial training with batch size 128, $\epsilon = 4/255$ and PGD step size $1/255$ on CIFAR-10. From epoch 150 to 200, the train robust accuracy increases from $85.82\%$ to $97.82\%$, and the test robust accuracy increases from $66.16\%$ to $66.99\%$.

Moreover, the same qualitative patterns persist on CIFAR-100, SVHN, and WideResNet-34-10: learning-rate drops trigger sharp curvature escalation, shrink posterior variances, and inflate the dominant spectral terms in our bound, thereby reproducing the characteristic onset of robust over-fitting. Taken together, the results demonstrate that the coupled evolution of curvature and posterior geometry—rather than dataset- or architecture-specific artifacts—is a universal mechanism governing adversarially robust generalization.

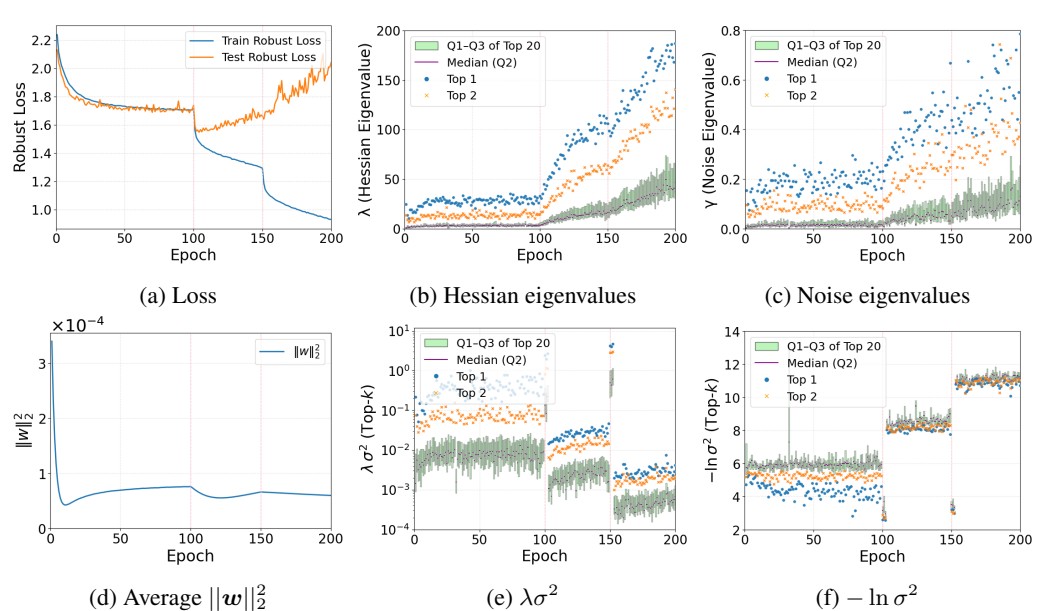

Figure 7: Additional results for adversarial training with batch size 128, $\epsilon = 12/255$ and PGD step size $3/255$ on CIFAR-10. From epoch 150 to 200, the train robust accuracy increases from 48.38% to 60.65%, while the test robust accuracy decreases from 39.75% to 35.94%.

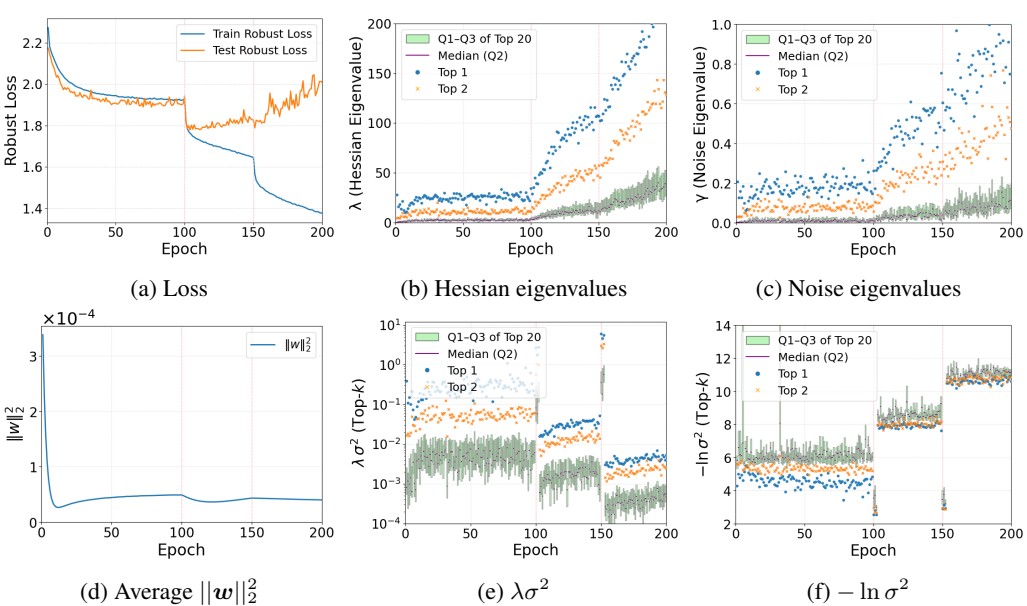

Figure 8: Additional results for adversarial training with batch size 128, $\epsilon = 16/255$ and PGD step size $4/255$ on CIFAR-10. From epoch 150 to 200, the train robust accuracy increases from 36.83% to 44.98%, while the test robust accuracy decreases from 34.01% to 32.37%.

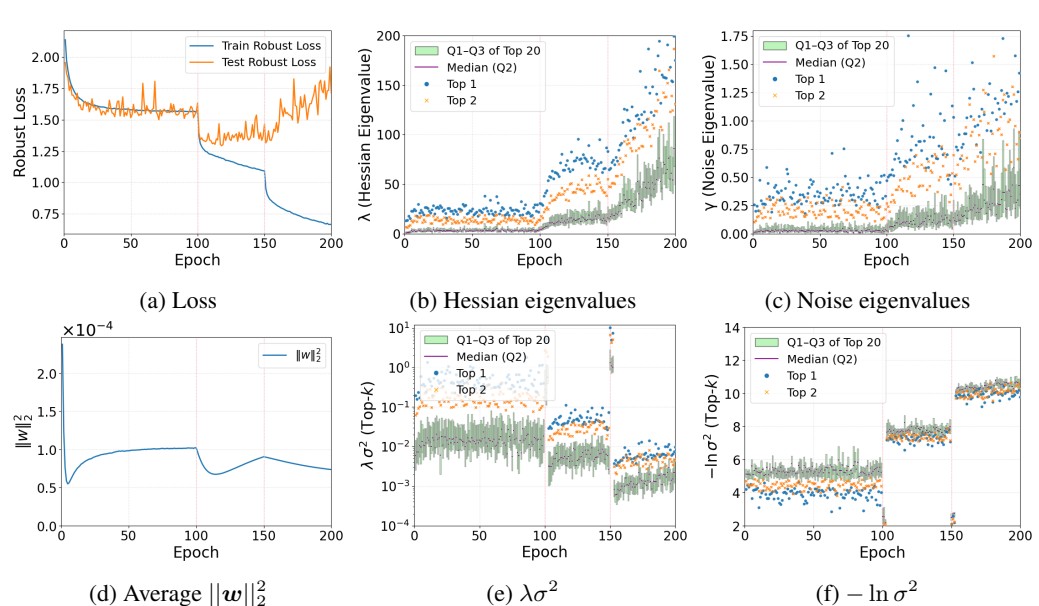

(a) Loss     (b) Hessian eigenvalues     (c) Noise eigenvalues

(d) Average $||\boldsymbol{w}||_2^2$     (e) $\lambda\sigma^2$     (f) $-\ln\sigma^2$

Figure 9: Additional results for adversarial training with batch size $64$ and $\epsilon = 8/255$ on CIFAR-10. From epoch $150$ to $200$, the train robust accuracy increases from $56.61\%$ to $71.72\%$, while the test robust accuracy decreases from $47.34\%$ to $46.36\%$.

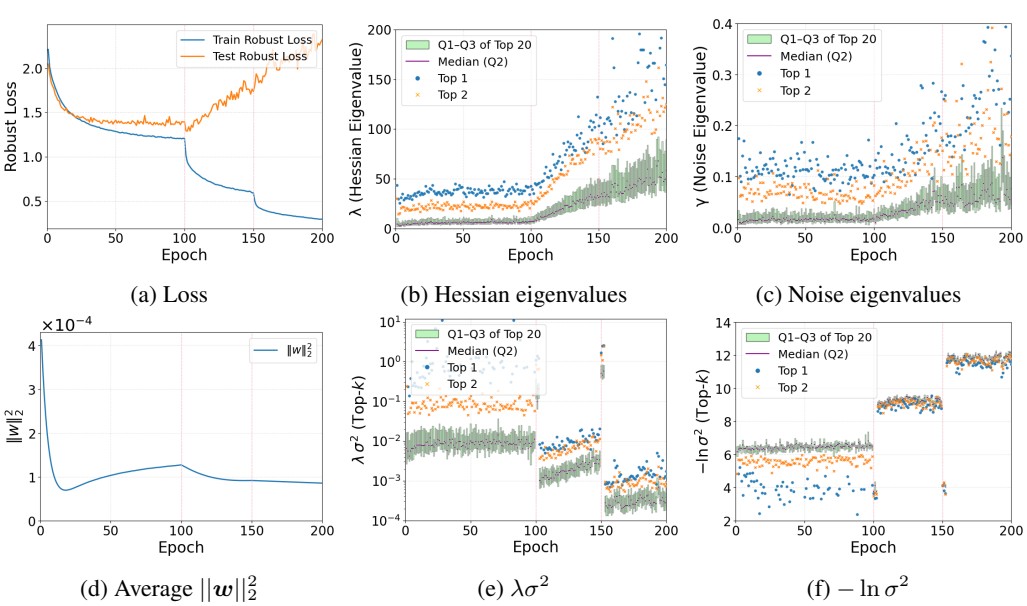

(a) Loss     (b) Hessian eigenvalues     (c) Noise eigenvalues

(d) Average $||\boldsymbol{w}||_2^2$     (e) $\lambda\sigma^2$     (f) $-\ln\sigma^2$

Figure 10: Additional results for adversarial training with batch size $256$ and $\epsilon = 8/255$ on CIFAR-10. From epoch $150$ to $200$, the train robust accuracy increases from $73.87\%$ to $86.71\%$, while the test robust accuracy decreases from $46.91\%$ to $43.41\%$.

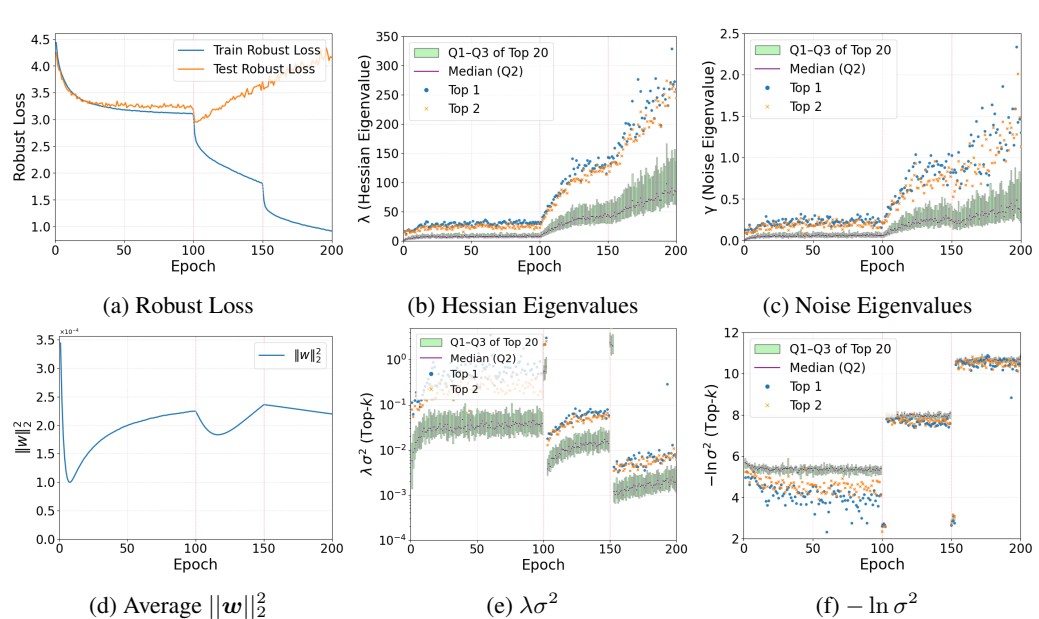

Figure 11: Additional results for adversarial training with batch size 128 and $\epsilon = 8/255$ on CIFAR-100. From epoch 150 to 200, the train robust accuracy increases from 48.54% to 72.90%, while the test robust accuracy slightly decreases from 21.86% to 21.32%.

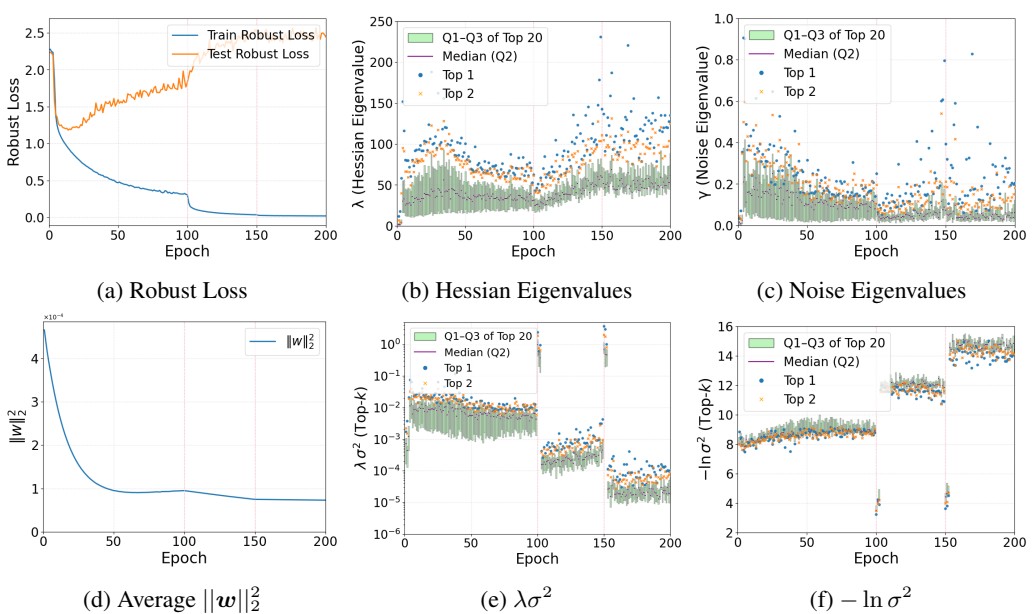

Figure 12: Additional results for adversarial training with batch size 128 and $\epsilon = 8/255$ on SVHN. From epoch 150 to 200, the train robust accuracy increases from 98.59% to 99.28%, while the test robust accuracy slightly increases from 54.06% to 54.31%.

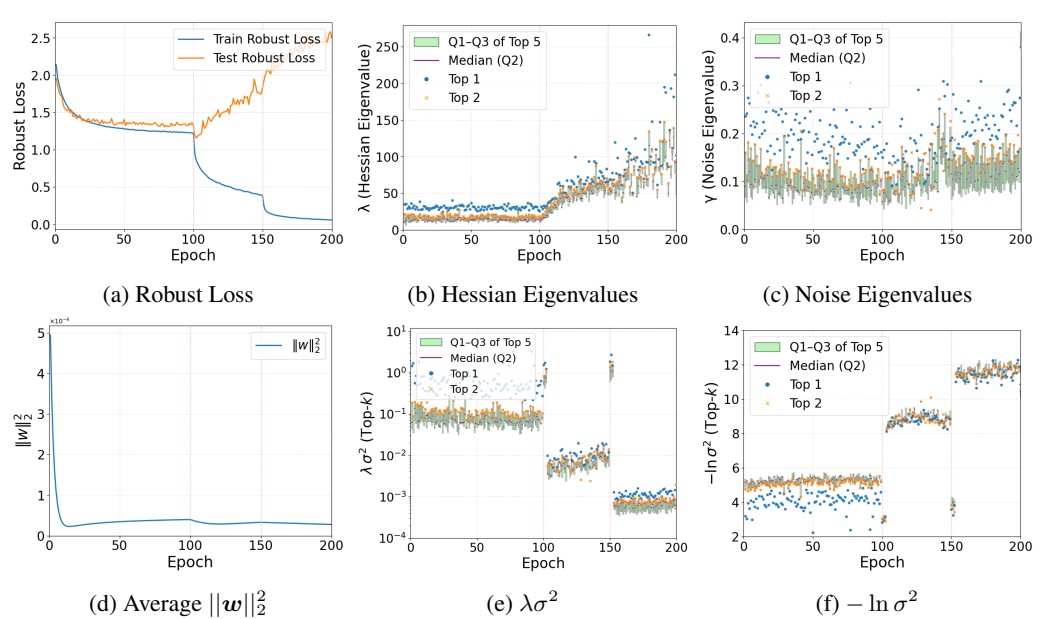

Figure 13: Additional results for adversarial training on CIFAR-10 with WideResNet-34-10. From epoch 150 to 200, the train robust accuracy increases from 82.91% to 97.71%, while the test robust accuracy stays around 48.28% to 48.37%.

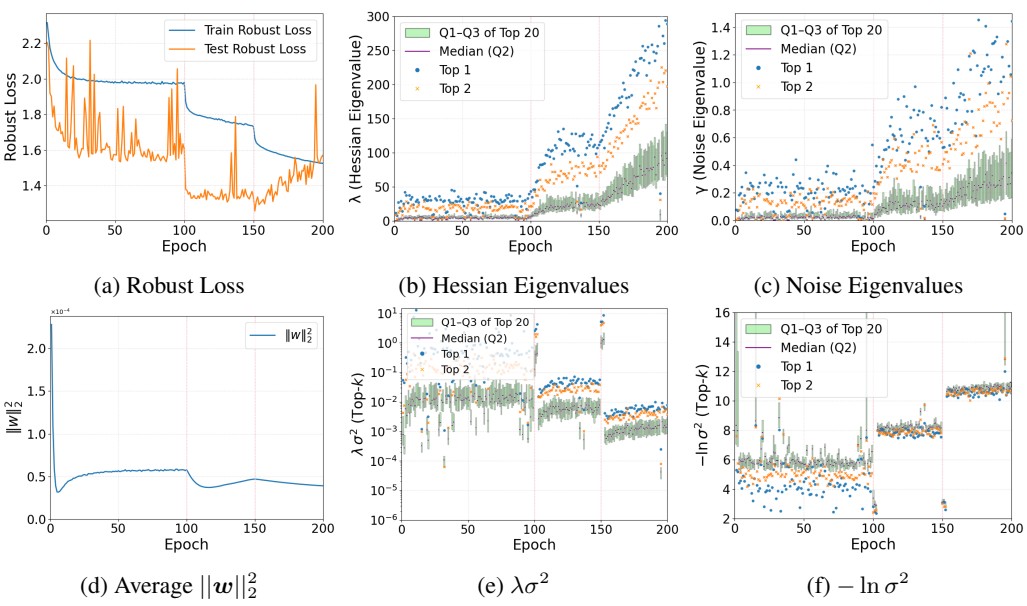

Figure 14: Results for semi-supervised adversarial training (Carmon et al., 2019) with batch size 128 and $\epsilon = 8/255$ on CIFAR-10. From epoch 150 to 200, the train robust accuracy increases from 32.25% to 39.42%, while the test robust accuracy decreases from 50.83% to 48.39%.

Table 1: Evaluation of AT and AWP at Stationary Regime.

| Method | Epoch | LR | Robust Loss | | Top-10 | | Top-20 | |
|--------|-------|-----|------|------|------|------|------|------|
| | | | Train | Test | $\sum_i \lambda_i\sigma_i^2$ | $-\sum_i \ln\sigma_i^2$ | $\sum_i \lambda_i\sigma_i^2$ | $-\sum_i \ln\sigma_i^2$ |
| AT | 98 | 0.1000 | 1.4417 | 1.5115 | 0.7392 | 55.2797 | 0.7869 | 119.6521 |
| | 99 | 0.1000 | 1.4448 | 1.5202 | 0.7691 | 54.5511 | 0.8319 | 116.6751 |
| | 148 | 0.0100 | 0.9568 | 1.3784 | 0.1224 | 81.9588 | 0.1564 | 166.0552 |
| | 149 | 0.0100 | 0.9450 | 1.3684 | 0.1138 | 81.7163 | 0.1462 | 166.8927 |
| | 198 | 0.0010 | 0.5651 | 1.6828 | 0.0149 | 110.5211 | 0.0187 | 226.3246 |
| | 199 | 0.0010 | 0.5647 | 1.7991 | 0.0176 | 109.8221 | 0.0228 | 224.3322 |
| AWP | 98 | 0.1000 | 1.5845 | 1.4217 | 0.3878 | 52.3761 | 0.4245 | 111.8577 |
| | 99 | 0.1000 | 1.5827 | 1.4138 | 0.3528 | 54.0705 | 0.3734 | 116.7480 |
| | 148 | 0.0100 | 1.3735 | 1.2468 | 0.0598 | 79.1922 | 0.0678 | 165.7136 |
| | 149 | 0.0100 | 1.3711 | 1.2463 | 0.0578 | 79.0380 | 0.0670 | 165.8011 |
| | 198 | 0.0010 | 1.3076 | 1.2237 | 0.0060 | 104.0535 | 0.0071 | 211.4283 |
| | 199 | 0.0010 | 1.3061 | 1.2191 | 0.0069 | 102.8904 | 0.0082 | 210.2709 |

Table 2: Evaluation of AT and AWP at Initial Phase of Non-Stationary Transition.

| Method | Epoch | LR | Robust Loss | | Top-10 | | Top-20 | |
|--------|-------|-----|------|------|------|------|------|------|
| | | | Train | Test | $\sum_i \lambda_i\sigma_i^2$ | $-\sum_i \ln\sigma_i^2$ | $\sum_i \lambda_i\sigma_i^2$ | $-\sum_i \ln\sigma_i^2$ |
| AT | 100 | 0.0100 | 1.4447 | 1.4568 | 7.1910 | 26.5496 | 8.4420 | 57.9231 |
| | 101 | 0.0100 | 1.2653 | 1.2717 | 6.6961 | 28.1169 | 8.0763 | 60.1948 |
| | 102 | 0.0100 | 1.2016 | 1.2732 | 7.5940 | 27.9818 | 9.8374 | 57.4486 |
| | 150 | 0.0010 | 0.9466 | 1.3881 | 19.7690 | 31.1946 | 25.2198 | 65.3582 |
| | 151 | 0.0010 | 0.8242 | 1.4036 | 19.3195 | 31.2056 | 25.1003 | 65.2913 |
| | 152 | 0.0010 | 0.7798 | 1.4614 | 18.1461 | 32.4100 | 24.8218 | 66.7165 |
| AWP | 100 | 0.0100 | 1.5810 | 1.4095 | 4.6366 | 26.3079 | 5.0578 | 86.3721 |
| | 101 | 0.0100 | 1.4798 | 1.3057 | 4.7988 | 26.4234 | 5.6639 | 57.1908 |
| | 102 | 0.0100 | 1.4496 | 1.3153 | 4.5107 | 26.3610 | 5.1733 | 59.2366 |
| | 150 | 0.0010 | 1.3709 | 1.2578 | 10.5189 | 27.6671 | 11.5429 | 88.3216 |
| | 151 | 0.0010 | 1.3397 | 1.2394 | 9.8622 | 28.2500 | 11.1231 | 62.2065 |
| | 152 | 0.0010 | 1.3303 | 1.2350 | 9.6732 | 28.1511 | 11.0591 | 62.7482 |

## F.2 VERIFYING THE SPECTRAL STRUCTURE OF OUR BOUNDS

We first empirically validate the two theoretical regimes considered in Section 4. Table 1 corresponds to the stationary setting of Theorem 4.5, where the posterior distribution $\mathcal{Q}$ has reached equilibrium during adversarial training under a fixed learning rate. Under this regime, the PAC-Bayesian robust generalization bound takes the following form if only keeping the dominating terms:

$$\frac{1}{2}\sum_i \lambda_i\sigma_i^2 \;+\; \frac{1}{2\beta}\Big(-\sum_i \ln\sigma_i^2\Big),$$

with the curvature–variance term unscaled and the log-determinant term attenuated by $1/\beta$, where $\beta$ is often set to the order of the square root of the sample set size ($\sqrt{|\mathcal{S}|}$) to balance the related terms in the PAC-Bayesian bound. The data in Table 1 exhibit exactly this behaviour: within each stationary plateau, both spectral terms evolve smoothly, and their relative magnitudes match the theoretical prediction that $\sum_i \lambda_i\sigma_i^2$ dominates unless posterior variance has collapsed substantially.

Table 2 characterizes the non-stationary transition behavior of adversarial training algorithms governed by Theorem 4.7. Immediately after a learning-rate decay, the stationary covariance condition no longer applies, and each $\sigma_i^2$ becomes a mixture of a decaying propagation term and an injected noise term evaluated at the smaller step size. This produces the predicted sharp, non-monotonic shifts: an abrupt decrease in $-\sum_i \ln\sigma_i^2$ and a slight increase in $\sum_i \lambda_i\sigma_i^2$. These discontinuities

appear only at decay points and are absent in stationary phases, providing direct experimental confirmation of the transient covariance dynamics.

### F.3 CONNECTING ROBUST OVERFITTING TO LEARNING RATE DECAY

Having established that the empirical dynamics match both stationary and transient theory, we now connect these dynamics to the emergence of robust overfitting. In stationary phases, curvature and covariance evolve gradually: $\sum_i \lambda_i \sigma_i^2$ contracts as the optimizer enters narrower regions of the landscape, and $-\sum_i \ln \sigma_i^2$ grows as the posterior becomes more concentrated. Decreasing the learning rate will disrupt this balance. The transient mixture of propagation and injected components induces an immediate reduction in the overall covariance scale, driving a sharp decrease in $-\sum_i \ln \sigma_i^2$ that outweighs the mild increase in $\sum_i \lambda_i \sigma_i^2$, thereby producing the initial drop in robust test loss.

As optimization continues with the smaller learning rate, adversarial curvature grows, while the posterior variance continues to shrink. Eventually, $-\sum_i \ln \sigma_i^2$ increases rapidly enough to dominate the shrinking curvature–variance term, reversing the direction of the bound and producing the subsequent rise in robust test loss. From a geometric perspective, these spectral dynamics describe a transition from a broad, weakly curved basin to an increasingly sharp and anisotropic one. The contraction of $\sum_i \lambda_i \sigma_i^2$ reflects the narrowing of the basin, while the growth of $-\sum_i \ln \sigma_i^2$ signals collapse into a low-dimensional subspace. This geometric tightening underlies the full "drop–then–rise" pattern characterizing robust overfitting.

In contrast, adversarial weight perturbation (AWP) fundamentally alters this geometry. By suppressing curvature amplification and preventing posterior collapse, it keeps both spectral terms within a moderate range. Consequently, the curvature–variance term remains dominant across training, and the combined spectral expression decreases monotonically, explaining why AWP avoids the overfitting dynamics observed in standard adversarial training.

### F.4 FURTHER DISCUSSION WITH VARYING $\epsilon$ AND BATCH SIZE.

Although our primary analysis focuses on learning-rate decay, the same spectral mechanism also predicts how adversarial radius $\epsilon$ and batch size influence robust generalization. Increasing $\epsilon$ steepens the adversarial loss landscape, enlarging the dominant curvature directions and accelerating posterior contraction. This amplifies the growth of $-\sum_i \ln \sigma_i^2$ and reduces $\sum_i \lambda_i \sigma_i^2$ more aggressively, pushing the optimizer into the log-determinant–dominated regime earlier in training. The resulting degradation in robust test loss mirrors the rise phase of the learning-rate–induced pattern.

Batch size acts through the gradient-noise spectrum. Larger batches reduce stochastic variability, leading to smaller stationary variances $\sigma_i^2$ via Equation 14 and hastening the onset of the collapse regime in which the log-determinant term dominates. Smaller batches inject more noise, maintain larger posterior variances, and thereby delay entry into this regime. Across these ablations, the qualitative evolution of $\sum_i \lambda_i \sigma_i^2$ and $-\sum_i \ln \sigma_i^2$ consistently mirrors the spectral behaviour observed under learning-rate decay. These results provide additional evidence that robust overfitting arises precisely when adversarial curvature intensifies while posterior covariance collapses, regardless of which hyperparameter induces these geometric shifts.

## G LLM USAGE

Large Language Models (LLMs) were employed solely for proofreading purposes in the preparation of this manuscript. Specifically, the LLM was used to detect and correct typographical errors and minor grammatical issues. Its role was restricted to ensuring the textual accuracy and consistency of the manuscript.

It is important to emphasize that the LLM was not involved in the conception of ideas, theoretical development, data analysis, or interpretation of results. All scientific content, including methodology, experiments, and conclusions, was entirely the responsibility of the authors. The use of the LLM was strictly limited to linguistic refinement at the level of typo correction and proofreading.

The authors take full responsibility for the content of the manuscript, including any sections proofread by the LLM. The application of the LLM adhered to ethical guidelines and did not contribute to plagiarism, scientific misconduct, or the generation of original scientific content.

