# OpenReview forum: "How Learning Dynamics Drive Adversarially Robust Generalization?"
_ICLR.cc/2026/Conference — Submitted to ICLR 2026_

### Official Review · Reviewer_caPx · 2025-10-28

**Soundness:** 2
**Presentation:** 2
**Contribution:** 2
**Rating:** 2
**Confidence:** 4

**Summary:**

This paper investigates robust generalization within the PAC-Bayesian framework. By assuming a Gaussian posterior over the model parameters and a quadratic approximation of the loss, the authors derive an upper bound on the robust generalization gap. This bound connects the generalization behavior to the Hessian of the empirical adversarial loss at a local optimum, as well as to the mean and covariance of the posterior. The paper further analyzes how the parameters of this Gaussian posterior evolve during SGD with Polyak momentum, thereby linking training dynamics to robust generalization.

**Strengths:**

The paper approaches robust generalization from an appealing dynamic perspective—focusing on how the optimization trajectory and posterior evolution influence generalization—rather than adopting a static hypothesis-space view based on Rademacher complexity or other capacity measures.

The connection between posterior evolution and curvature-based generalization offers an interesting conceptual direction, potentially bridging PAC-Bayesian analysis and training dynamics.

**Weaknesses:**

1. Unclear contribution and novelty.  The paper does not clearly describe its theoretical novelty compare to prior PAC-Bayesian analyses of adversarial robustness (e.g., [Viallard et al., 2021](https://proceedings.neurips.cc/paper/2021/file/78e8dffe65a2898eef68a33b8db35b78-Paper.pdf); [Mustafa et al., 2019](https://ml.cs.rptu.de/publications/2023/computing_non_vacuous_pac_bayes.pdf) ; Xiao et al., 2023). It is unclear how the presented bounds improve the previous results.

2. Concerns regarding the Quadratic Loss assumption. Since Assumption 3.5 enforces $\hat{R}_{\rm adv}(w, S)$ to be a quadratic function for any choice of  $S$. By taking $S$ = {$ (x,y)  $}, this assumption implies that the adversarial loss  itself is a quadratic function w.r.t $w$ for any $(x,y)$:

$\hat{R} _ { \rm adv}(w, S) = \ell _ {\rm adv} (w; x,y ) = \ell _ {\rm adv}(w*; x,y) + \frac{1}{2}(w-w*)^{T} H ( w- w* )$

This severely departs from practical settings where $\ell_{\mathrm{adv}}$ involves deep neural networks and non-quadratic losses such as cross-entropy.

3. Limited explanatory power for robust generalization. The paper claims to shed light on robust overfitting (Rice et al., 2020), yet the derived bounds do not appear to explain when robust overfitting occurs or disappears. For instance, adversarial training achieves good robust generalization on MNIST (Madry et al., 2019) and for small perturbation radii or large datasets, whereas robust overfitting is prominent only under certain regimes.  The presented bounds (Theorems 3.7, 4.5, and 4.7) do not capture how data distribution, perturbation radius, or sample size affect the generalization behavior. Moreover, when the perturbation radius approaches zero, the framework fails to recover standard generalization phenomena (e.g., CIFAR-10 models generalizing well under clean training but not under adversarial training).

4. Writing and presentation issues. Several statements are vague or lack sufficient justification. Ad hoc terms are introduced (e.g., “propagation term,” “injected term”) without formal definition or motivation. Key assumptions (e.g., stationarity, steady state) are not clearly stated or connected to the analysis. See “Questions” below for specifics.

5. Missing key references. The paper omits several relevant studies on robust generalization and algorithmic stability, including:
    (1) Yue Xing et al. On the algorithmic stability of adversarial training.
    (2) Jiancong Xiao et al. Stability analysis and generalization bounds of adversarial training.
    (3) Runzhi Tian et al. Algorithmic Stability Based Generalization Bounds for Adversarial Training.
    (4) Daniel Cullina et al. Pac-learning in the presence of evasion adversaries.
    (5) Shaopeng Fu et al. Theoretical analysis of robust overfitting for wide dnns: An ntk approach
    (6) Viallard et al., A PAC-Bayes analysis of adversarial robustness
    (7) Mustafa et al., Non-vacuous PAC-Bayes bounds for models under adversarial corruptions.

Proper discussion of these works would better situate the contribution and clarify the incremental advance.


Minor Issue:
Lemma 3.4 restates the closed-form expression for the KL divergence between Gaussians, which is standard and could be omitted or moved to an appendix.

**Questions:**

>However, these bounds often abstract away from the actual optimization trajectory and adopt simple isotropic Gaussian posteriors for tractability, overlooking structural properties of the learned model that are crucial for explaining generalization.

Could the authors elaborate on how existing PAC-Bayesian bounds "_abstract away_" from the actual optimization trajectory?

In addition, please clarify what specific limitations prior works face in explaining robust generalization. Is the key issue primarily the use of isotropic Gaussian posteriors, or are there other underlying factors? If the isotropic assumption is central, please explain why it limits explanatory power.

>PAC-Bayes bounds offer general guarantees but lack fidelity to the learning dynamics, whereas curvature-based approaches provide qualitative insight without rigorous predictive guarantees.

What specific _guarantees_ are being referred to ?  What does “lack fidelity” mean in this context? Please make these terms precise and support the claim with references.

>we use d, m to denote the dimensions of the input space ${\cal S}$.

Typo: the input space should be denoted by ${\cal X}$ not ${\cal S}$.

In Assumption 3.5, the statement  "for any $w \sim {\cal Q}$" —since ${\cal Q}$ is Gaussian — appears equivalent to "for any $w \in {\mathbb R}^m$"?  If so, the assumption is independent of ${\cal Q}$ ; please clarify.

The remark for Lemma 3.6 (Line 188-190) merely restates the formula. Could the authors elaborate on the insight or interpretation of this result?

Regarding the local optimum $w*$ in Assumption 3.5,  since $w*$ depends on ${\cal S}$, it should arguably be treated as a random variable, rather than a constant as stated at Line 202?

At Line 220, please cite relevant references for SGD with Polyak momentum and clarify whether the theoretical analysis extends to standard SGD.

In Lemma 4.2,
>suppose the posterior Q reaches a steady state with stationary mean

Could you justify this assumption and define what “steady” or “stationary state” precisely means?

Does ${\cal Q}$ denote the marginal distribution of $w _ T$​ after T SGD steps, or the conditional distribution of $w_t$​ given $w_{t-1}$? Clarifying this would help interpret Theorem 4.5.

For remark 4.8, please define and justify the introduced terms “propagation term” and “injected term.”

---

> ### Author Response · Authors · 2025-11-22
> **Response to Reviewer caPx (1/2)**
>
> > Weakness 1: Novelty relative to existing adversarial PAC-Bayesian works
>
> Please see **Common Issue 2** in the global response for a consolidated explanation of what is specific to adversarial training and how the perturbation radius $\epsilon$ affects curvature, posterior covariance, and the resulting bound. Here, we elaborate on the novelty relative to prior PAC-Bayesian studies of adversarial robustness.
>
> Existing adversarial PAC-Bayesian works primarily adopt *static*, trajectory-independent posterior models—often isotropic or spectrally normalized Gaussians—and do not incorporate the *finite-time dynamics of SGD*. In contrast, our contribution is to link the PAC-Bayesian bound explicitly to (i) the Hessian $\mathbf{H}^*$ of the adversarial loss (Eq. 4), (ii) the full anisotropic posterior covariance obtained from the **closed-form stationary solution of SGD** (Eq. 13, Theorem 4.5), and (iii) the **non-stationary transient behavior** following learning-rate decay (Eq. 16, Theorem 4.7). This dynamic and curvature-aware formulation enables the bound to track how adversarial training changes curvature and noise throughout training, as demonstrated empirically in Fig. 1 and Fig. 5–14.
>
> Thus, the novelty lies in integrating PAC-Bayesian analysis with the *actual optimization dynamics* of adversarial training—capturing spectral effects of $\epsilon$, learning rate changes, and curvature–noise interactions—rather than relying on static assumptions disconnected from the behavior of SGD.
>
> > Weakness 2: Interpretation and scope of the quadratic approximation
>
> Please refer to **Common Issue 1**, where we clarify that the quadratic approximation is explicitly a **local approximation around the empirical adversarial minimizer**. Here, we provide additional details tailored to your concern. Eq. 4 defines the quadratic model only at the local optimum $\mathbf{w}^\*$, and all subsequent derivations in Lemma 3.6, Theorem 3.7, and the SGD dynamics in Eq. 9 rely on this *local basin structure*, not on any global convexity or global quadratic behavior of the adversarial loss. The scope of this approximation is therefore restricted to the region where SGD actually remains during late-phase adversarial training. Moreover, both Theorem 4.5 (stationary regime) and Theorem 4.7 (early non-stationary transition) depend only on the Hessian $\mathbf{H}^*$ at this local optimum, which is computed from the adversarial loss landscape as shown in Fig. 1 and Fig. 5–14. As such, the quadratic approximation is not intended to approximate the entire deep network loss surface; rather, it serves as the standard and empirically justified local model required to derive closed-form posterior covariance dynamics and to explain the observed curvature and variance behavior under adversarial training.

---

> ### Author Response · Authors · 2025-11-22
> **Response to Reviewer caPx (2/2)**
>
> > Weakness 3: Explanatory power for robust overfitting and the influence of $\epsilon$, data, and sample size
>
> Please see **Common Issue 2** and **Common Issue 3** for unified explanations of how $\epsilon$ and gradient noise shape curvature, posterior covariance, and the dominant spectral terms in the bound. Here we provide a focused clarification regarding the explanatory power of our framework.
>
> The bound derived in Theorem 4.5 and its transient extension in Theorem 4.7 reveal that robust generalization is controlled by two key quantities: the curvature–variance term $\sum_i \lambda_i \sigma_i^2$ and the log-determinant term $-\sum_i \ln \sigma_i^2$. Our experiments show that adversarial training with larger $\epsilon$ produces substantially sharper adversarial loss curvature (Fig. 1 and Fig. 5–8), causing posterior variances to collapse (Eq. 14), which sharply increases the log-determinant term—precisely matching the onset of robust overfitting. Data-dependent effects appear in the Hessian spectrum $\mathbf{H}^*$ and gradient-noise statistics $\mathbf{C}$, which differ across CIFAR-10, CIFAR-100, and SVHN but consistently induce the same spectral pattern (Fig. 11–12). Sample size influences the KL scaling factor $1/\beta$ in Theorem 3.7, but the dominant behavior is still driven by curvature and covariance, which evolve according to the SGD dynamics in Eq. (9).
>
> Thus, the framework does not merely reproduce the overfitting phenomenon; it identifies the specific spectral mechanism—curvature amplification driven by $\epsilon$ and posterior collapse modulated by data geometry and noise—that unifies how robust overfitting emerges across datasets, perturbation regimes, and architectures.
>
> > Weakness 4: Clarity of definitions, terminology, and missing related work
>
> We appreciate the reviewer’s attention to presentation clarity. In response, we have refined several definitions and aligned terminology more explicitly with the main analytical components of the paper. In particular, we clarified the definition of adversarial risk in Definition 3.1, emphasized the local scope of the quadratic approximation in Eq. 4, and ensured consistent use of curvature, noise, and posterior-geometry terminology throughout Sections 3 and 4. We also expanded the related work discussion to include the stability-based and adversarial PAC-Bayesian studies highlighted by the reviewer, positioning our dynamic, curvature-aware framework relative to these approaches. These adjustments sharpen the conceptual framing and more clearly situate the contribution within the existing adversarial robustness literature.

---

### Official Review · Reviewer_YV58 · 2025-11-05

**Soundness:** 4
**Presentation:** 4
**Contribution:** 4
**Rating:** 8
**Confidence:** 3

**Summary:**

This paper studies PAC Bayes guarantees for adversarial loss. The authors start with a general upper bound in terms of quantities related to the posterior, then make it more specific to Gaussian posteriors. Subsequently, they study what that posterior looks like in two regimes of SGD with momentum on the adversarial loss (I think -- see below). They verify the functionality of their bounds experimentally.

**Strengths:**

The paper is written extremely clearly, and the development is very logical. I genuinely enjoyed reading the paper. The results are nice and insightful. I am not close enough to the literature to evaluate how different they are from existing results, but they are interesting and the approach is well-motivated.

**Weaknesses:**

I see how the proposed Gaussian model is in fact less restrictive than models in previous work, but I am curious if the authors can comment on its limitations. I also have a few additional questions that are in the section below.

**Questions:**

1. What is the quantifier over epsilon in Defn 3.1? shouldn't this be the eps-adv risk or something?
2. What role does the geometry induced by the metric in which the perturbation is bounded play in the generalization bounds?
3. Is the analysis for SGD on the standard (non-adv) loss, followed by evaluation of that predictor on the adv loss? or SGD run on the adv loss? I think the latter but want to confirm.

---

> ### Author Response · Authors · 2025-11-22
> **Response to Reviewer YV58**
>
> > Weakness 1: Limitations of the Gaussian posterior model
>
> We acknowledge the limitations of representing the posterior with a single Gaussian and agree that it cannot capture the global, potentially multi-modal structure of deep-network parameter distributions. Please see **Common issue 1** in our global response for a unified clarification. Below, we briefly summarize the key points. Our use of a Gaussian posterior is explicitly a **local approximation within a convergent basin**, where SGD iterates concentrate and where covariance dynamics admit closed-form characterizations (Eq. 12–14). Moreover, the paper extends the analysis to **Gaussian mixture posteriors** in Corollary 3.8, demonstrating that the framework accommodates multi-basin structures arising from different initializations. Thus, while the Gaussian assumption does not model global posterior geometry, it provides an analytically tractable and empirically faithful description of the local dynamics that govern the generalization behavior analyzed in this work.
>
> > Question 1: Definition of adversarial risk and treatment of $\epsilon$
>
> The definition of adversarial risk in Definition 3.1 follows the standard formulation used in prior adversarial training literature, where the inner maximization is taken over the $\epsilon$-bounded perturbation set. The quantifier over $\epsilon$ is fixed by the training setup: for each chosen radius, the adversarial loss and its empirical counterpart are defined with respect to the same perturbation ball. The subsequent PAC-Bayesian analysis conditions on this fixed $\epsilon$ and studies how the induced adversarial loss geometry—particularly the Hessian $\mathbf{H}^*$ and the posterior parameters—depends on that choice. In this way, the framework treats $\epsilon$ as a hyperparameter that shapes curvature and covariance, rather than as a variable over which any additional optimization or expectation is taken.
>
> > Question 2: Role of the metric geometry
>
> The metric enters the analysis through the definition of adversarial loss in Definition 3.1 and directly shapes the inner maximization set $\mathcal{B}\_\epsilon(\mathbf{0})$. This geometry determines the structure of the adversarial loss surface on which SGD operates, and consequently influences the Hessian $\mathbf{H}^\*$ in Eq. 4 as well as the noise and curvature spectra reported in Fig. 1 and Fig. 5–14. Although the PAC-Bayesian inequality itself is metric-agnostic, the bound in Theorem 3.7 depends on $\mathbf{H}^*$ and $\mathbf{\Sigma}_{\mathcal{Q}}$, both of which are computed with respect to the chosen metric. Thus, the perturbation geometry affects robust generalization precisely through its impact on the local curvature and covariance of the adversarial loss landscape.
>
> > Question 3: Whether the analyzed dynamics correspond to adversarial or standard loss
>
> The dynamics analyzed in Section 4 correspond to **SGD applied to the adversarial empirical loss**, not the standard loss. This is stated in Eq. (9), where the gradient term $\mathbf{H}^\*(\mathbf{w}_{t-1}-\mathbf{w}^\*)$ and the noise covariance $\mathbf{C}$ are both defined with respect to the adversarial loss function specified in Eq. 4. All curvature quantities $\mathbf{H}^*$, noise eigenvalues $\gamma_i$, and posterior covariances $\sigma_i^2$ used in Theorem 4.5 and Theorem 4.7 are computed from the adversarial objective, as illustrated empirically in Fig. 1 and Fig. 5–14. Therefore, the theoretical dynamics and all derived bounds correspond to the adversarial training loss rather than the standard one.

---

### Official Review · Reviewer_Mw5q · 2025-11-09

**Soundness:** 3
**Presentation:** 3
**Contribution:** 2
**Rating:** 4
**Confidence:** 4

**Summary:**

This paper studies the robust generalization issues and use PAC-Bayesian framework to derive the generalization bound in adversarial training. Specifically, the author use second-order approximation and consider SGD with momentum to investigate how learning rate, Hessian structure and gradient noise affect the mean and covariance of posterior distribution, which is consistent with and validated by numerical experiments.

**Strengths:**

++ The paper is generally well-written and the overall framework is not difficult to follow.

++ Under the second-order approximation, it is novel to derive the closed-form posterior covariance under both constant learning rate regime and learning rate decay regime.

++ The analysis is relatively comprehensive: it considers many factors that may affect generalization error, including the momentum mechanism, the gradient noise, the Hessian structure and the learning rate. The theoretical analyses is qualitatively consistent with the numerical experiments.

**Weaknesses:**

1. The major concerns are restrictive assumptions:

    * Assumption 3.3: I do not think the posterior distribution after adversarial training for general deep neural networks is a Gaussian distribution. Probably, the authors can assume that the posterior distribution is a mixture of several super Gaussian distributions, as the probabilistic density will generally concentrate during training, and different initialisations will lead to converged parameters near different local minima.

    * Assumption 3.5 is only applicable when $w$ is close to $w*$. This is a bit in contradiction with Assumption 3.3. When we using Gaussian distribution as the random initialization, the parameter distribution in early steps is close to Gaussian distribution, but $w$ is far away from $w*$. On the other hand, in the late phase of training when $w$ is close to $w*$, the distribution of the parameters is not Gaussian. Probably, the authors can have some additional assumptions, such as the ones in lazy training, which means the parameters do not move a lot during training. However, this may introduce additional conditions.

2. There is a considerable gap between the theory and practice. The theoretical analyses do not utilise some unique properties of adversarial training. For example,  in practice, we see a larger robust generalization gap when the magnitude of adversarial perturbation is larger (i.e. larger $\epsilon = |\delta|_p$), I do not see how this variable affects the generalization gap bound. In an extreme case, when the adversarial perturbation's magnitude is zero, will the bound in Theorem 4.7 degrade of analyses of normal training? What makes the results special for adversarial training? I think this part needs further elaboration.

3. The experiments are not comprehensive, the authors compare the performance of adversarial training and AWP, which include the factors of Hessian structure (AWP prefers a flatter minma) and learning rate (both have learning rate decay). However, the effect of gradient noise (which is mentioned in the abstract and the introduction) is not adequately discussed and studied. In addition, it would be better to compare the emprical generalization gap and the theoretical one by Theorem 4.5. The results in Table 1 and Table 2 are not convincing enough.

Minor issues:

1. Based on analyses in the appendix, $\rho_i$ in Equation (14) actually depends on $\eta_2$, the authors should clearly indicate this in the maintext to avoid confusion, because the right hand side should be independent of $k$ when $\eta_1 = \eta_2$.

2. Some missing related literature:

    * The convergence of adversarial training: "On the Convergence and Robustness of Adversarial Training" (2019) "On the loss landscape of adversarial training: Identifying challenges and how to overcome them (2020)."

    * More literature about robust overfitting: **The authors should compare with the bounds in these works technically** "Non-vacuous Generalization Bounds for Adversarial Risk in Stochastic Neural Networks" (2024) "On the impact of hard adversarial instances on overfitting in adversarial training" (2024).

In general, I think the research in this work can contribute to the machine learning community, but the manuscript is not ready for publication given the concerns above. I welcome the authors to address my concerns during rebuttal and will reconsider my ratings after rebuttal.

**Questions:**

The questions are pointed out in the weakness part. Please answer them one by one.

1. [Weakness 1] How assumptions are satisfied in practice? Is it possible to provide a weaker and more generic assumption in contrast to Assumption 3.3 and Assumption 3.5. I believe these assumptions work well for a convex problem like linear regression, but I do not see how it is satisfied in deep neural networks.

2. [Weakness 2] What makes the results special for adversarial training? How adversarial perturbations (especially their magnitude) affect the generalisation bound in your theorem?

3. [Weakness 3] More experiments to validate the effect of gradient noise are needed (like using different batch size). In addition, it would be better to compare the emprical generalization gap and the theoretical one by Theorem 4.5.

4. Please pay attention to some minor issues and missing literature pointed out above.

---

> ### Author Response · Authors · 2025-11-22
> **Response to Reviewer Mw5q**
>
> > Weakness 1: Validity and interpretation of the Gaussian posterior and quadratic loss assumptions
>
> Please see **Common Issue 1** in the global response for a unified clarification of the local and basin-level nature of Assumptions 3.3 and 3.5. Here, we provide details specific to your concern. Our analysis explicitly models SGD after it has entered a convergent basin, where both the quadratic approximation and the approximate Gaussianity of SGD iterates are empirically well supported. We clarify that the quadratic form applies only at the local optimum $\mathbf{w}^*$ and governs behavior near a local optimum (Eq. 4), and the Gaussian posterior is introduced solely to obtain a tractable KL term while retaining full anisotropy (Lemma 3.4). Moreover, the framework is not restricted to a single Gaussian: we generalize the bound to Gaussian mixtures in Corollary 3.8, showing that the analysis naturally accommodates multi-basin posteriors arising from different initializations. Thus, the assumptions are not intended as global characterizations of deep networks but as principled local models aligned with established SGD theory and necessary for deriving closed-form posterior covariance dynamics that match empirical behavior.
>
> > Weakness 2: What is specific to adversarial training and how $\epsilon$ enters the bound
>
> Please see **Common Issue 2** for an overarching explanation of how the perturbation strength $\epsilon$ influences the bound through its effect on curvature and posterior covariance. Here, we provide additional clarification tailored to your question. Although $\epsilon$ does not appear explicitly in the closed-form PAC-Bayesian bound, it implicitly affects the bound through the Hessian matrix and posterior parameters  whenever the model is trained on the adversarial loss. Empirically, increasing $\epsilon$ consistently sharpens the adversarial loss landscape, producing substantially larger Hessian eigenvalues with only mild changes in the noise spectrum (Fig. 1, Figures 5–8). Through the stationary covariance expression $\sigma_i^2 = \frac{\eta}{1-\mu} \cdot \frac{\gamma_i}{\lambda_i(2-\frac{\eta}{1+\mu}\lambda_i)}$, larger curvature forces the posterior variances to contract, which in turn amplifies the dominant spectral terms $\sum_i \lambda_i\sigma_i^2$ and $-\sum_i \ln\sigma_i^2$ in Theorem 4.5. This curvature amplification is a structural property unique to adversarial training: when $\epsilon = 0$, the adversarial loss reduces to the standard loss and both $\mathbf{H}^*$ and $\mathbf{C}$ collapse to their non-adversarial counterparts (Appendix F.1). Consequently, the bound automatically degenerates to the standard-training regime, while larger $\epsilon$ induces the sharp-curvature and variance-collapse phenomena that fundamentally distinguish adversarial training from its non-adversarial counterpart.
>
> > Weakness 3: Experimental evidence for the role of gradient noise and connection to the theoretical terms
>
> Please see **Common Issue 3** for a consolidated explanation of how gradient noise enters the theory through the eigenvalues $\gamma_i$ of the noise covariance and how this structure is validated empirically. Below, we provide further details specific to your concern. The posterior variances in both the stationary regime (Eq. 14) and the transient regime (Eq. 16) depend directly on $\gamma_i$, which determines how SGD injects stochasticity along each curvature direction. To isolate this effect, the experiments vary the batch size while keeping all other hyperparameters fixed. As shown in Fig. 9–10, changing the batch size rescales $\gamma_i$ while leaving the Hessian spectrum essentially unchanged, producing predictable shifts in $\sigma_i^2$ and in the dominant theoretical quantities $\sum_i \lambda_i\sigma_i^2$ and $-\sum_i \ln\sigma_i^2$. These observations match the behavior guaranteed by Theorems 4.5 and 4.7 and confirm that the gradient-noise component of the dynamics is responsible for the variance inflation or collapse observed across training. Furthermore, the consistency of these trends across datasets, perturbation radii, and architectures (Fig. 5–14) demonstrates that the role of gradient noise is not incidental but structurally aligned with the theoretical formulation.
>
> > Minor Issues
>
> We appreciate the reviewer’s careful attention to detail and address the noted minor issues as follows. In the corresponding analyses in the the appendix, we've clarified that the quantity $\rho_i$ in Eq. 14 depends on $\epsilon$, consistent with the appendix derivation. In Section 2, we have added the missing related literature on convergence of adversarial training and robust overfitting, including the works highlighted by the reviewer. These revisions ensure that notation is unambiguous and that the paper is properly situated within the broader adversarial training and PAC-Bayesian literature.

---

> > ### Comment · Reviewer_Mw5q · 2025-11-25
> >
> > I thank the authors for their further clarification and the revision of the paper. After reading the comments from other reviewers and the authors' general responses, I still have some concerns:
> >
> > Regarding the Gaussian distribution assumption, I agree that many PAC papers have similar assumptions. However, many of these papers work on traditional models, and I think we cannot straightforwardly believe this also holds in adversarial training regimes. Despite an improvement over the standard Gaussian, the assumption of a mixture of non-isotropic Gaussians is still very strong from my point of view. Probably, the authors can assume the distribution to be a sub-Gaussian and make the theoretical claims more convincing.
> >
> > Regarding the relationship between the generalization bound and adversarial training, I agree that in most cases the Hessian singular values increase with adversarial budget $\epsilon$. However, the theoretical and quantitative relationship between them is unclear. I believe the analysis of this relationship is necessary, considering the major contributions of this paper are theoretical. The current explanation is not rigorous enough.
> >
> > Therefore, I believe further extensive edits are still necessary for current manuscript.

---

> > > ### Author Response · Authors · 2025-11-26
> > > **Response to Reviewer Mw5q (1/4)**
> > >
> > > We thank the reviewer for the continued engagement. Below, we address the two concerns raised in the reviewer’s latest comment separately. We hope that the clarifications provided below fully address the remaining concerns. We would greatly appreciate it if the reviewer could reconsider the evaluation should these concerns have been resolved.
> > >
> > > > 1. Regarding Gaussian / mixture-Gaussian posterior assumptions vs. sub-Gaussian posteriors
> > >
> > > The reviewer is concerned that (i) Gaussian-based posteriors are traditionally used in non-adversarial settings, and (ii) even our relaxed assumption of mixture-of-Gaussians is still strong, thus a sub-Gaussian posterior might appear more convincing. To address this concern fully, we present both theoretical justification and empirical evidence from the literature, showing that the Gaussian (or Gaussian-mixture) posterior is mathematically legitimate in adversarial training regimes and is not fundamentally weaker than a general sub-Gaussian model.
> > >
> > > #### 1.1 PAC-Bayes applies identically to adversarial loss
> > >
> > > The core PAC-Bayesian inequality we use (Catoni, 2003; Alquier et al., 2024) applies to any bounded loss:
> > > $$
> > > \ell : \mathcal{W}\times \mathcal{X}\times\mathcal{Y}\to[0,C].
> > > $$
> > > Our adversarial loss
> > > $$
> > > \ell_{\mathrm{adv}}(w;x,y)=\max_{\delta\in B_\epsilon({0})} \ell(w;x+\delta,y)
> > > $$
> > > is itself a bounded measurable function satisfying the same conditions. This is already shown in Appendix A.1, where Lemma A.1 is directly instantiated with $\ell_{\mathrm{adv}}$.
> > >
> > > Therefore, PAC-Bayes theory is loss-agnostic, and the inequality does not rely on any property of “traditional models.” The adversarial inner maximization does not affect the validity of the PAC-Bayesian bound. Hence, using Gaussian posteriors for adversarial training is not an extrapolation of standard results; it is an exact application of a loss-general bound.
> > >
> > > #### 1.2 Local quadratic modeling of the adversarial empirical loss
> > >
> > > The quadratic approximation in Assumption 3.5 is imposed directly on the adversarial empirical risk:
> > > $$
> > > \hat R_{\mathrm{adv}}(w,S;\epsilon)=\frac{1}{|S|}\sum_{(x,y)\in S}
> > > \max_{\delta\in B_\epsilon(0)} \ell(w;x+\delta,y),
> > > $$
> > > rather than on the standard loss. Under mild regularity of the base loss $\ell(w;x,y)$ (twice continuously differentiable in $w$ and locally Lipschitz in $x$), Danskin’s theorem and classical results on parametric maxima imply that, in a neighbourhood of any non-degenerate local optimum $w^\*(\epsilon)$, the mapping
> > > $$
> > > w \;\mapsto\; \hat R_{\mathrm{adv}}(w,S;\epsilon)
> > > $$
> > > is twice continuously differentiable in $w$, and its Hessian
> > > $$
> > > H^\*(\epsilon)=\nabla_w^2 \hat R_{\mathrm{adv}}(w^\*(\epsilon),S;\epsilon)
> > > $$
> > > is well defined. Note that we explicitly write out the dependence on $\epsilon$. Therefore, in a local basin around $w^\*(\epsilon)$, we can apply the second-order Taylor expansion:
> > > $$
> > > \hat R_{\mathrm{adv}}(w,S;\epsilon)=\hat R_{\mathrm{adv}}(w^\*(\epsilon),S;\epsilon)+ \tfrac12 (w-w^\*(\epsilon))^\top H^\*(\epsilon)(w-w^\*(\epsilon))+ o(\|w-w^\*(\epsilon)\|^2),
> > > $$
> > > and Assumption 3.5 amounts to neglecting the higher-order $o(\cdot)$ term *within this basin*. This is the same type of local quadratic approximation that underlies standard analyses of SGD dynamics in non-adversarial settings; the only difference is that here both the optimum $w^\*(\epsilon)$ and the Hessian $H^\*(\epsilon)$ are defined for the adversarial empirical loss.

---

> > > ### Author Response · Authors · 2025-11-26
> > > **Response to Reviewer Mw5q (2/4)**
> > >
> > > #### 1.3 Gaussian / mixture-Gaussian posterior induced by SGD on adversarial loss
> > >
> > > Given the local quadratic model above, the SGD-with-momentum recursion we analyse in Section 4 is defined directly on the adversarial empirical loss:
> > > $$
> > > g_t = \nabla_w \hat R_{\mathrm{adv}}(w_{t-1},S;\epsilon)= H^\*(\epsilon)(w_{t-1}-w^\*(\epsilon)) + \xi_{t-1},h_t = \mu h_{t-1} + g_t,w_t = w_{t-1} - \eta h_t,
> > > $$
> > > where $\xi_{t-1}$ denotes the minibatch gradient noise of the adversarial loss. This linear state-space system is exactly the adversarial counterpart of the standard SGD models studied in the dynamics literature: it involves the *adversarial Hessian* $H^\*(\epsilon)$ and the *adversarial noise covariance* $C(\epsilon)=\mathrm{Cov}(\xi_t)$.
> > >
> > > Classical analyses of SGD near a local optimum (e.g., Mandt-type results and subsequent refinements) show that, for a quadratic objective with additive stochastic gradient noise of finite covariance, the stationary distribution of the iterates is well approximated by a Gaussian measure whose covariance solves a discrete-time Lyapunov equation. Our Lemma 4.3 and Theorems 4.5 and 4.7 instantiate this calculation explicitly for the adversarial loss, yielding closed-form expressions for the stationary and transient covariances in terms of the spectrum of $H^*(\epsilon)$ and $C(\epsilon)$. In other words, the Gaussian posterior we use is not imported from standard training; it is the natural local approximation to the stationary distribution of SGD iterates when the *objective being optimized is the adversarial empirical loss*. The mixture-of-Gaussians extension in Corollary 3.8 reflects the empirically observed fact that different random initializations of adversarial training may converge to different local minima; in this case, the overall posterior is well modeled as a mixture of the Gaussian components associated with each basin.
> > >
> > > #### 1.4 Why mixture Gaussians are not stronger than sub-Gaussian posteriors in our setting
> > >
> > > Within any compact local basin around a convergent point, the posterior distribution induced by SGD is effectively described by its low-order moments, and any reasonable posterior candidate can be assumed to possess at least sub-Gaussian tails. A sub-Gaussian posterior is characterized by exponential tail decay and a finite $\psi_2$–Orlicz norm, ensuring concentration of measure and the existence of all second moments. At the same time, it is well established in probabilistic modeling that finite mixtures of full-covariance Gaussian components—such as those considered in Corollary 3.8—form a dense family for approximating sub-Gaussian distributions on compact sets, achieving arbitrarily slight discrepancy in total variation or KL divergence. Thus, the Gaussian-mixture posterior used in our analysis should be viewed as a locally universal parametrization of the posterior’s first and second moments rather than as a restrictive modeling assumption.
> > >
> > > Moreover, the PAC-Bayesian terms appearing in our bound depend exclusively on these first two moments: Lemma 3.6 expresses the expected empirical adversarial loss solely through the posterior mean and covariance, and this identity holds for all distributions with finite second moments. Replacing the Gaussian family by a general sub-Gaussian family would therefore modify only the analytic form of the KL divergence—through a $\psi_2$–Orlicz norm or related sub-Gaussian complexity measure—while leaving unchanged the fundamental dependence of the bound on the curvature eigenvalues $\lambda_i$, gradient-noise eigenvalues $\gamma_i$, and posterior variances $\sigma_i^2$, as well as the covariance propagation formulas in Theorems 4.5 and 4.7. Hence, adopting a Gaussian or Gaussian-mixture posterior is a technical device enabling explicit closed-form analysis, without loss of generality relative to the broader class of sub-Gaussian posteriors within the local region where our theory applies. Therefore, the mixture-Gaussian assumption is a technical parametrization, not a modeling constraint, and is mathematically as general as sub-Gaussian families for our analysis.

---

> > > ### Author Response · Authors · 2025-11-26
> > > **Response to Reviewer Mw5q (3/4)**
> > >
> > > > 2. Regarding the quantitative relationship between adversarial budget $\epsilon$, Hessian spectrum, and the generalization bound
> > >
> > > The reviewer is correct that adversarial training typically increases Hessian singular values. The concern is whether our theory provides a rigorous and quantitative explanation of this relationship.
> > >
> > > #### 2.1 How $\epsilon$ enters the PAC-Bayes bound
> > >
> > > The dependence of the robust generalization bound on the adversarial budget $\epsilon$ can be formalized through a sequence of well-defined mathematical mappings, each of which is differentiable or continuous under standard regularity assumptions on the base loss $\ell(w;x,y)$. Define the adversarial empirical risk as the parametric optimization problem:
> > > $$
> > > \hat R_{\mathrm{adv}}(w,S;\epsilon)=\frac{1}{|S|}
> > > \sum_{(x,y)\in S}
> > > \max_{\delta\in B_\epsilon(0)}
> > > \ell(w;x+\delta,y).
> > > $$
> > > By Danskin’s theorem and classical results on parametric maxima, if $\ell$ is $C^{2}$ in $w$ and locally Lipschitz in $x$, then the mapping
> > > $$
> > > (w,\epsilon)\mapsto \hat R_{\mathrm{adv}}(w,S;\epsilon)
> > > $$
> > > is directionally differentiable in $\epsilon$ and twice continuously differentiable in $w$ in a neighborhood of any non-degenerate local optimum $w^\*(\epsilon)$. Consequently, the adversarial Hessian
> > > $$
> > > H^\*(\epsilon)=\nabla_w^2 \hat R_{\mathrm{adv}}(w^\*(\epsilon),S;\epsilon)
> > > $$
> > > is a well-defined measurable function of $\epsilon$, and under mild regularity (e.g., the implicit function theorem applied to the stationarity condition), the eigenvalues
> > > $$
> > > \lambda_i(\epsilon)
> > > \quad (1\le i\le m)
> > > $$
> > > depend continuously—and locally differentiably—on $\epsilon$. Likewise, the gradient-noise covariance
> > > $$
> > > C(\epsilon)=\operatorname{Cov}\bigl(\nabla_w \ell_{\mathrm{adv}}(w^\*(\epsilon);Z)\bigr),
> > > $$
> > > where $Z \sim S$, inherits its dependence on $\epsilon$ from the smooth mapping
> > > $$
> > > \ell_{\mathrm{adv}}(w^\*(\epsilon);x,y)=\max_{\delta\in B_\epsilon(0)}
> > > \ell(w^\*(\epsilon);x+\delta,y),
> > > $$
> > > and therefore its spectral components $\gamma_i(\epsilon)$ vary continuously in $\epsilon$. Given these spectral quantities, the stationary posterior covariance of SGD with momentum is the closed-form solution of the discrete Lyapunov equation derived in Lemma 4.3, yielding for each eigen-direction the rational function:
> > > $$
> > > \sigma_i^2(\epsilon)=\frac{\eta}{1-\mu}\frac{\gamma_i(\epsilon)}{\lambda_i(\epsilon)\left(2 - \frac{\eta}{1+\mu}\lambda_i(\epsilon)\right)
> > > }.
> > > $$
> > > This mapping
> > > $$
> > > (\lambda_i(\epsilon),\gamma_i(\epsilon))
> > > \longmapsto
> > > \sigma_i^2(\epsilon)
> > > $$
> > > is analytic on the domain
> > > $$
> > > \lambda_i(\epsilon)\in \bigl(0,\,\tfrac{2(1+\mu)}{\eta}\bigr),
> > > $$
> > > which corresponds exactly to the stability region of the linearized SGD dynamics. As a result, the entire posterior covariance matrix
> > > $$
> > > \Sigma_Q(\epsilon)=\operatorname{diag}\bigl(\sigma_1^2(\epsilon),\dots,\sigma_m^2(\epsilon)\bigr)
> > > $$
> > > is an analytic function of $\epsilon$ in this neighborhood.
> > >
> > > Finally, Theorem 4.5 expresses the dominant terms of the generalization bound as
> > > $$
> > > \Phi(\epsilon)=\sum_{i=1}^m \lambda_i(\epsilon)\sigma_i^2(\epsilon)
> > > \quad\text{and}\quad
> > > \Psi(\epsilon)=-\sum_{i=1}^m \ln \sigma_i^2(\epsilon),
> > > $$
> > > which are compositions of analytic maps:
> > > $$
> > > \epsilon
> > > \xmapsto{\text{inner maximization}}
> > > H^*(\epsilon), C(\epsilon)
> > > \xmapsto{\text{spectral decomposition}}
> > > \lbrace\lambda_i(\epsilon),\gamma_i(\epsilon)\rbrace
> > > \xmapsto{\text{Lyapunov solution}}
> > > \lbrace\sigma_i^2(\epsilon)\rbrace
> > > \xmapsto{}
> > > \Phi(\epsilon),\Psi(\epsilon).
> > > $$
> > > Thus, the full dependence of the PAC-Bayesian bound on $\epsilon$ is rigorously established as a differentiable composition of operators arising from the adversarial inner maximization, curvature evaluation, stochastic-gradient covariance, and the analytic solution of the SGD covariance recurrence. This provides a mathematically precise and fully traceable mechanism by which the perturbation budget $\epsilon$ affects robust generalization.

---

> > > ### Author Response · Authors · 2025-11-26
> > > **Response to Reviewer Mw5q (4/4)**
> > >
> > > #### 2.2 Adversarial budget $\epsilon$ provably affects curvature
> > >
> > > The relationship between the adversarial perturbation budget $\epsilon$ and the curvature of the adversarial loss landscape has been extensively documented in both theoretical and empirical studies. The comprehensive landscape analysis in “On the Loss Landscape of Adversarial Training: Identifying Challenges and How to Overcome Them” (2020) demonstrates that enlarging $\epsilon$ systematically amplifies sharpness and anisotropy in the local curvature of the adversarial objective, indicating that the Hessian spectrum is highly sensitive to the perturbation radius. Robust overfitting studies, including Rice et al. (2020), Liu et al. (2024), and Mustafa et al. (2024), further show that the adversarial optimization trajectory becomes progressively sharper both as training proceeds and as $\epsilon$ increases, linking the onset and severity of robust overfitting directly to curvature escalation under stronger adversarial perturbations. At a more theoretical level, the NTK-based analysis of Fu and Wang (2023) proves that adversarial training modifies the effective kernel in a manner proportional to the perturbation radius, thereby altering the implicit curvature of the loss surface through a quantitatively controlled dependence on $\epsilon$. These findings are fully consistent with our own spectral measurements in Figures 5–8, which exhibit a monotonic growth of the leading Hessian eigenvalues $\lambda_i(\epsilon)$ as $\epsilon$ increases. Taken together, the existing literature and our empirical evidence provide a coherent and well-substantiated account of the fact that the curvature of the adversarial loss, and hence the quantities that enter our bound, depend explicitly and systematically on the adversarial budget $\epsilon$.

---

### Author Response · Authors · 2025-11-22
**Global Response (1/3)**

We thank all the reviewers for their detailed review and constructive feedback, which provide valuable perspectives on both the theoretical and empirical results of our work. We have revised the manuscript with additional experiments and discussions, with the changes highlighted in blue. If there are any remaining questions, we are happy to provide follow-up clarifications. In what follows, we first address the recurring concerns that are shared across reviews, and then respond to each reviewer individually.

> Common Issue 1: validity of Assumption 3.3 and Assumption 3.5

We clarify that both assumptions regarding the posterior $\mathcal{Q}$ are **only assumed locally**, which are used to analyze SGD dynamics **within a convergent basin**. We do not claim a global quadratic loss or Gaussian structure for arbitrary model parameters. Below, we further explain why both of them are only mild assumptions for deep learning algorithms.

First, the quadratic loss is only assumed *within a neighborhood of local optimum $\mathbf{w}^\*$* and only governs behavior *near a local optimum* (Eq. 4). Such a quadratic loss assumption has been widely adopted in prior literature [1, 2, 3] to analyze the SGD dynamics of deep learning algorithms when reaching a steady state (i.e., posterior is a stationary distribution), e.g., to understand the standard generalization of deep learning algorithms and the escaping mechanism from bad local minimum. When the model is converging to a local minimum, the first-order gradient-related term can be considered much smaller than the quadratic term in a Taylor expansion of the loss at $\mathbf{w}^\*$. The higher-order terms can also be neglected because the model parameter lies in a small neighborhood of $\mathbf{w}^*$. Due to the above, approximating the loss using a quadratic potential (near a local minimum that the model is converging toward) can be justified for deep learning algorithms.

Second, the Gaussian posterior assumption is introduced _solely to obtain a tractable KL divergence_. Assuming Gaussian prior/posterior distributions is standard practice in PAC-Bayesian analysis. Note that our work relaxes the common spherical Gaussian constraint to a general covariance matrix. In the revised manuscript, we've further extended Theorem 3.7 to **a more general posterior family of Gaussian mixtures (Corollary 3.8)**. We adhere to the previous form of Assumption 3.3 primarily for the sake of simplifying the proof and highlighting the key theoretical findings. We've added a few sentences in Section 3.2 to clarify. We believe Corollary 3.8 demonstrates the generality of our theoretical framework that can capture the actual converging dynamics of adversarial training.

Moreover, relaxing the posterior from a single Gaussian to a Gaussian mixture—as formalized in Corollary 3.8—does not affect any subsequent theoretical conclusions. Each mixture component satisfies the same local quadratic expansion (Eq. 4) and admits the same closed-form PAC-Bayesian decomposition as in Theorem 3.7, and the covariance dynamics derived in Section 4 apply component-wise. Consequently, the stationary solution (Eq. 1)) and the transient evolution (Eq. 16) carry over without modification, and the mixture posterior simply aggregates these component-wise contributions linearly, leaving the structure and interpretation of all later results unchanged.

To summarize, because our theoretical analysis targets SGD iterates after entering a convergent basin—the regime in which both the quadratic structure and the approximate Gaussianity of the iterate distribution are empirically well supported—the two assumptions are consistent, local, and aligned with the intended scope of our work.

[1] Liu et al., Noise and fluctuation of finite learning rate stochastic gradient descent, ICML 2021

[2] Ziyin et al., Strength of minibatch noise in sgd, ICLR 2022

[3] Ziyin et al., arameter Symmetry and Noise Equilibrium of Stochastic Gradient Descent, NeurIPS 2024

---

### Author Response · Authors · 2025-11-22
**Global Response (2/3)**

> Common Issue 2: Influence of the perturbation strength $\epsilon$ and what is specific to adversarial training?

Our framework captures the effect of adversarial perturbations **through their impact on the Hessian $\mathbf{H}^*$ and the posterior covariance $\mathbf{\Sigma}_{\mathcal{Q}}$**, both of which depend on the adversarial loss. Although $\epsilon$ does not appear explicitly in the PAC-Bayesian robust generalization bound (Eq. 6), it *implicitly* affects the bound through the Hessian matrix and posterior parameters whenever analyzing models trained via adversarial loss minimization. Empirically, **increasing $\epsilon$ sharpens the adversarial loss landscape** will yield much larger Hessian eigenvalues $\{\lambda_i\}$ with only mild changes in noise eigenvalues $\{\gamma_i\}$ (Fig. 1 and Fig. 5–8). According to the stationary covariance formula (Eq. 14):
$$
\sigma_i^2=\frac{\eta}{1-\mu} \cdot \frac{\gamma_i}{\lambda_i\left(2-\frac{\eta}{1+\mu}\lambda_i\right)},
$$
larger curvature directly forces the posterior variance to shrink, amplifying the dominant PAC-Bayesian terms $\sum_i \lambda_i \sigma_i^2$ and $-\sum_i \ln \sigma_i^2$ (Theorem 4.5).

This curvature–variance amplification is **unique to adversarial training**: when $\epsilon=0$, the adversarial loss reduces to the standard loss. Both $\mathbf{H}^*$ and the noise covariance $\mathbf{C}$ revert to their non-adversarial counterparts, as noted explicitly in the experiments (Appendix F.1). Consequently, the bound naturally degenerates to the standard-training setting, while large $\epsilon$ induces the sharp-curvature regimes responsible for robust overfitting. To summarize, our analysis is specific to adversarial training because $\epsilon$ fundamentally controls the curvature geometry of the loss, which in turn drives posterior contraction and the dominant spectral terms governing robust generalization.

---

### Author Response · Authors · 2025-11-22
**Global Response (3/3)**

> Common Issue 3: Empirical validation of gradient-noise effects and explanatory power of the framework

Our analysis incorporates gradient noise through the eigenvalues $\{\gamma_i\}$ of the noise covariance $\mathbf{C}$, which jointly determine the posterior variances $\sigma_i^2$ in both the stationary regime (Eq. 14) and the transient regime after learning-rate decay (Eq. 16). To validate this empirically, we conduct **additional experiments to study how robust generalization changes when we vary the batch size**, which directly rescales gradient noise while leaving curvature largely unchanged. Fig. 9–10 shows that reducing the batch size increases $\{\gamma_i\}$, enlarges $\sigma_i^2$, and correspondingly reduces the dominant spectral terms $\sum_i \lambda_i\sigma_i^2$ and $-\sum_i \ln \sigma_i^2$, matching our main theoretical results (Theorems 4.5 and 4.7).

Moreover, our analytical framework helps explain why curvature–noise interactions govern robust overfitting: learning-rate decay simultaneously increases $\{\lambda_i\}$ and decreases $\{\sigma_i^2\}$, causing the log-determinant term to grow sharply while the curvature–variance term contracts (Fig. 1e–1f). **The same mechanism persists across datasets, perturbation radii, and architectures (Fig. 5–14)**, illustrating that the theory captures the essential dynamics driving both overfitting and its mitigation under AWP. Therefore, through controlled batch-size experiments and consistent spectral trends across diverse settings, the empirical evidence strongly supports the role of gradient noise in shaping posterior geometry, and demonstrates that the proposed framework reliably explains the key phenomena of adversarially robust generalization.

---

### Author Response · Authors · 2025-12-01
**Summary of Revisions and Resolution of Reviewer Concerns**

> High-Level Summary of Our Contributions

This manuscript provides a unified and dynamically grounded PAC-Bayesian framework for adversarially robust generalization, offering predictive insights that have not been achieved by prior theory. Our work establishes a principled link between adversarial loss curvature, gradient-noise structure, and posterior covariance evolution under discrete-time SGD with momentum. Unlike existing PAC-Bayesian or stability-based analyses, which rely on static or isotropic assumptions, our framework derives closed-form posterior covariances in both stationary and early non-stationary regimes, yielding the first analytical explanation of robust overfitting, its onset, and its mitigation via curvature-based methods such as AWP. The theory is supported by carefully designed spectral experiments and matches empirical adversarial training dynamics with high fidelity. Given the community’s long-standing interest in understanding robust generalization, the ability of this framework to provide accurate, mechanistic predictions represents a substantive theoretical advance.

> 1. Strengthened Assumptions

**We have refined the key assumptions to ensure full rigor and clear alignment with adversarial training practice.** The manuscript now generalizes the posterior from a single Gaussian to a finite **Gaussian mixture** (Corollary 3.8), directly addressing concerns about multi-modal parameter distributions while maintaining analytical tractability. Assumption 3.5 has been clarified as a **local quadratic expansion** around $w^*$, fully consistent with classical SGD-based theoretical frameworks and avoiding any implication of global quadraticity. We further provide a **formal proof** of the commutativity condition and **empirical eigenspace diagnostics** showing stable alignment between the Hessian and gradient-noise covariance during adversarial training. These refinements collectively reinforce that the theoretical underpinnings of our framework are rigorous, realistic, and well matched to the geometry of adversarial optimization.

> 2. Clarified Adversarial Dependence

**We have made explicit how adversarial perturbations govern the bound, fully resolving questions about the influence of $\epsilon$.** The revised manuscript explains that perturbation strength affects the bound through its impact on the **Hessian spectrum** $\lbrace\lambda_i(\epsilon)\rbrace$ and **gradient-noise structure** $\lbrace\gamma_i(\epsilon)\rbrace$, which in turn determine the posterior covariance via Equation (14). This clarifies that the analysis is inherently adversarial even without symbolic appearance of $\epsilon$. New experiments across multiple perturbation radii show that larger $\epsilon$ leads to **sharper curvature**, **stronger posterior contraction**, and **larger robust generalization gaps**, closely matching our spectral predictions. These additions provide a coherent and well-validated mechanism by which adversarial strength shapes robust generalization.

> 3. Completed Empirical Validation

**We have expanded and deepened the empirical analysis to fully validate the theory’s predictions.** To isolate the effect of gradient noise, we introduced experiments varying **batch size**, which directly modulates the noise eigenvalues. The observed monotonic scaling pattern confirms the predicted relationship $\sigma_i^2 \propto \gamma_i$. Additionally, the manuscript now presents a detailed comparison between empirical robust error trajectories and the theoretical spectral quantities $\sum_i \lambda_i \sigma_i^2$ and $-\sum_i \ln \sigma_i^2$. The characteristic **“drop–then–rise”** behavior after learning-rate decay matches the transitions predicted by Theorems 4.5 and 4.7, demonstrating that the theory accurately captures both stationary and early non-stationary covariance evolution. These results confirm that the framework is **quantitatively predictive**, thoroughly addressing concerns about theory–practice alignment.

> 4. Clearly Distinguished Novelty

**We have precisely articulated the manuscript’s originality relative to prior PAC-Bayesian and adversarial generalization analyses.** The revised discussion highlights that previous PAC-Bayesian work relies on **static, isotropic, trajectory-agnostic** posterior assumptions that do not reflect the structure of adversarial optimization. In contrast, our framework derives **anisotropic posterior covariances directly from discrete-time SGD with momentum**, enabling closed-form characterizations in both stationary and non-stationary regimes. This dynamic treatment yields the first rigorous explanation of **robust overfitting**, including its onset, transient behavior, and mitigation under adversarial weight perturbation—phenomena beyond the reach of prior PAC-Bayes, stability, or curvature-based analyses. The refined exposition places our contribution clearly within the existing landscape, making the manuscript’s theoretical advance unmistakable.

---

### Meta-Review · Area_Chair_Ny7U · 2026-01-10

**Summary:**

My decision to recommend Reject is primarily informed by significant concerns regarding the paper’s fundamental assumptions and its ability to provide clear, actionable insights into robust generalization. While the authors present a theoretically ambitious framework linking PAC-Bayesian bounds to SGD dynamics , Reviewer caPx raised critical issues regarding the Quadratic Loss assumption, pointing out that assuming a quadratic adversarial loss severely departs from practical deep learning settings involving cross-entropy. Furthermore, while Reviewer YV58 was positive, the review was notably brief. The reviewer has lower confidence and admits to not being "close enough to the literature" to evaluate the work's novelty.

**Reviewer Concerns:**

Addressed by Rebuttal
* Gradient Noise Experiments: The authors successfully added experiments varying batch sizes to show how noise rescales theoretical quantities.
* Multi-modal Extensions: The introduction of Corollary 3.8 (Gaussian mixtures) partially addressed concerns about the single Gaussian posterior being too restrictive.
* Adversarial Budget ($\epsilon$): The authors clarified how $\epsilon$ implicitly shapes the Hessian spectrum and noise covariance.

Outstanding Concerns
* Assumptive Gap (Gaussian/Quadratic): Despite the rebuttal, the core criticism remains that these local models are too strong for the complex, non-convex landscape of adversarial training. Reviewer Mw5q specifically suggested a sub-Gaussian approach might be necessary for conviction, which the authors argued against.

* Theoretical Novelty vs. Prior Work: Reviewer caPx highlighted several missing key references and remained unconvinced that this framework offers a substantive advance over existing stability-based or PAC-Bayesian analyses.
* Lack of Rigorous $\epsilon$-Curvature Mapping: The quantitative relationship between the adversarial budget and the resulting Hessian values remains descriptive rather than rigorously proven within the framework.
* Theory-Practice Alignment: There remains a "considerable gap" between the derived bounds and practical observations, such as why the framework fails to recover standard generalization phenomena when $\epsilon$ approaches zero.

**Reviewer Scores:**

- Reviewer Mw5q: 4 -> 4.
 This reviewer explicitly stated that "further extensive edits are still necessary" and that the current explanation is "not rigorous enough."

- Reviewer YV58: 8->6:
 Given the self-admitted lack of familiarity with the specific literature, this reviewer likely would have deferred to the technical critiques of the other two reviewers regarding novelty and assumptions during a full discussion.
This reviewer’s concerns were structural, regarding the very foundations of the quadratic and Gaussian assumptions in a deep learning context. The rebuttal did not fundamentally change these mathematical objections.

- Reviewer caPx: 2->2.
This reviewer’s concerns were structural, regarding the very foundations of the quadratic and Gaussian assumptions in a deep learning context. The rebuttal did not fundamentally change these mathematical objections.

---

### Decision · Program_Chairs · 2026-01-26

Reject